# A CMDP-within-online framework for Meta-Safe Reinforcement Learning

**Vanshaj Khattar**
Virginia Tech
Blacksburg, VA 24061
vanshajk@vt.edu

**Yuhao Ding**
UC Berkeley
Berkeley, CA 94709
yuhao_ding@berkeley.edu

**Bilgehan Sel**
Virginia Tech
Blacksburg, VA 24061
bsel@vt.edu

**Javad Lavaei**
UC Berkeley
Berkeley, CA 94709
lavaei@berkeley.edu

**Ming Jin** [*]
Virginia Tech
Blacksburg, VA 24061
jinming@vt.edu

## Abstract

Meta-reinforcement learning has widely been used as a learning-to-learn framework to solve unseen tasks with limited experience. However, the aspect of constraint violations has not been adequately addressed in the existing works, making their application restricted in real-world settings. In this paper, we study the problem of meta-safe reinforcement learning (Meta-SRL) through the CMDP-within-online framework to establish the *first provable guarantees* in this important setting. We obtain task-averaged regret bounds for the reward maximization (optimality gap) and constraint violations using gradient-based meta-learning and show that the task-averaged optimality gap and constraint satisfaction improve with task-similarity in a static environment or task-relatedness in a dynamic environment. Several technical challenges arise when making this framework practical. To this end, we propose a meta-algorithm that performs inexact online learning on the upper bounds of within-task optimality gap and constraint violations estimated by off-policy stationary distribution corrections. Furthermore, we enable the learning rates to be adapted for every task and extend our approach to settings with a competing dynamically changing oracle. Finally, experiments are conducted to demonstrate the effectiveness of our approach.

## 1 Introduction

The field of meta-reinforcement learning (meta-RL) has recently evolved as one of the promising directions that enables reinforcement learning (RL) agents to learn quickly in dynamically changing environments (Finn et al., 2017; Mitchell et al., 2021; Zintgraf et al., 2021). Many real-world applications, nevertheless, have safety constraints that should rarely be violated, which existing works do not fully address. Safe RL problems are often modeled as constrained Markov decision processes (CMDPs), where the agent aims to maximize the value function while satisfying given constraints on the trajectory (Altman, 1999). However, unlike meta-learning, CMDP algorithms are not designed to generalize efficiently over unseen tasks (Paternain et al., 2022; Ding et al., 2021a; Ding & Lavaei, 2022) In this paper, we study how meta-learning can be principally designed to help safe RL algorithms adapt quickly while satisfying safety constraints.

There are several unique challenges involved in meta-learning for the CMDP settings. First, multiple losses are incurred at each time step, i.e., reward and constraints, which are typically nonconvex and coupled through dynamics. Hence, adapting existing theories developed for stylized settings such as online convex optimization (Hazan et al., 2016) is not straightforward. Second, it is unrealistic to assume the computation of a globally optimal policy for CMDPs (unlike online learning (Hazan et al., 2016)). Thus, classical online learning algorithms that assume exact or unbiased estimator of the loss

---

[*]Corresponding author

function do not apply (Khodak et al., 2019). Overall, there is an interplay among nonconvexity, the stochastic nature of the optimization problem, as well as algorithm and generalization considerations, posing significant complexity to leverage inter-task dependency (Denevi et al., 2019).

To this end, we propose a provably low-regret online learning framework that extends the current meta-learning algorithms to safe RL settings. Our main contributions are as follows:

1. **Inexact CMDP-within-online framework:** We propose a novel CMDP-within-online framework where the within-task is CMDP, and the meta-learner aims to learn the meta-initialization and learning rate. In our framework, the meta-learner only requires the inexact optimal policies for each within-task CMDP and the approximate state visitation distributions estimated using collected offline trajectories to construct the upper bounds on the suboptimality gap and constraint violations. An upper bound on these estimation errors is established in Theorem 3.1.

2. **Task-averaged regret in terms of empirical task-similarity:** We show that the task-averaged regrets for optimality gap (TAOG) and constraint violations (TACV) (Def. 1) diminish with respect to both the number of steps in the within-task algorithm $M$ and the number of tasks $T$. Specifically, task-averaged regret of $\mathcal{O}\left(\frac{1}{\sqrt{M}}\sqrt{\frac{\mathcal{E}_T}{\sqrt{T}} + \hat{D}^{*2}}\right)$ holds, where $\mathcal{E}_T$ is the total inexactness in online learning and $\hat{D}^*$ is the empirical task-similarity (Theorem 3.2).

3. **Adapting to a dynamic environment:** We adapt the learning rates for each task to environments that entail dynamically changing meta-initialization policies. An improved rate of $\mathcal{O}\left(\frac{1}{M^{3/4}\sqrt{T}}\left(\mathcal{E}_T + \sqrt{\frac{\mathcal{E}_T}{T} + \hat{V}_\psi^2}\right)\right)$ for TAOG and TACV are shown, where $\hat{V}_\psi$ is the empirical task-relatedness with respect to a sequence of changing comparator policies $\{\psi_t^*\}_{t=1}^T$ (Corollary 1).

Incorporating all these components makes our Meta-safe RL (Meta-SRL) approach highly practical and theoretically appealing for potential adaption to different RL settings. Furthermore, we remark on some key technical contributions that support the above developments, which may be of independent interest: *1)* We study the *optimization landscape* of CMDP (Theorem 3.1) that is algorithmic-agnostic, which differs from the existing work of (Mei et al., 2020)[Lemmas 3 and 15] that is restricted to the setting of policy gradient. This is achieved by developing new techniques based on tame geometry and subgradient flow systems; *2)* we provide static and dynamic regret bounds for *inexact online gradient descent* (see Appendix E), which we leverage to obtain our final theoretical results in Theorems 3.2, 3.3, and Corollary 1. Due to the space restrictions, the related work can be found in Appendix A.

## 2 CMDP-WITHIN-ONLINE FRAMEWORK

In this section, we introduce the CMDP-within-online framework for the Meta-SRL problems. In this framework, a within-task algorithm (such as CRPO (Xu et al., 2021)) for some CMDP task $t \in [T]$ is encapsulated in an online learning algorithm (meta-learning algorithm), which decides upon a sequence of initialization policy $\phi_t$ and learning rate $\alpha_t > 0$ for each within-task algorithm. The goal of the meta-learning algorithm is to minimize some notion of task-averaged performance regret to facilitate provably efficient adaptation to a new task.

### 2.1 CMDP AND THE PRIMAL APPROACH

**Model.** For each task $t \in [T]$, a CMDP $\mathcal{M}_t$ is defined by the state space $\mathcal{S}$, the action space $\mathcal{A}$, discount factor $\gamma$, initial state distribution over the state-space $\rho_t$, the transition kernel $P_t(s'|s, a) : \mathcal{S} \times \mathcal{A} \to \mathcal{S}$, reward functions $c_{t,0} : \mathcal{S} \times \mathcal{A} \to [0, 1]$ and cost functions $c_{t,i} : \mathcal{S} \times \mathcal{A} \to [0, 1]$ for $i = 1, ..., p$. The actions are chosen according to a stochastic policy $\pi_t : \mathcal{S} \to \Delta(\mathcal{A})$ where $\Delta(\mathcal{A})$ is the simplex over the action space. We use $\Delta(\mathcal{A})^{|\mathcal{S}|}$ to denote the simplex over all states $\mathcal{S}$. The initial policy for task $t$ is denoted as $\pi_{t,0}$. The discounted state visitation distribution of a policy $\pi$ is defined as $\nu_{t,s_0}^\pi(s) := (1-\gamma)\sum_{m=0}^\infty \gamma^m P_t\left(s_m = s \mid \pi, s_0\right)$ and we write $\nu_t^*(s) := \mathbb{E}_{s_0 \sim \rho_t}\left[\nu_{t,s_0}^\pi(s)\right]$ as the visitation distribution when the initial state follows $\rho_t$ at task $t$. We denote $\pi_t^*$ as an optimal policy for task $t$ and $\nu_t^*(s) := \mathbb{E}_{s_0 \sim \rho_t}\left[\nu_{t,s_0}^{\pi_t^*}(s)\right]$ is the corresponding state visitation distribution induced by policy $\pi_t^*$ when the initial state $s_0$ is sampled from initial state distribution $\rho_t$ at task $t$.

**Policy parametrization.** We consider the softmax parametrization. For any $\theta \in \mathbb{R}^{|\mathcal{S}| \times |\mathcal{A}|}$, the corresponding softmax policy $\pi_\theta$ is defined as $\pi_\theta(a \mid s) := \frac{\exp(\theta(s,a))}{\sum_{a' \in \mathcal{A}} \exp(\theta(s,a'))}, \forall(s,a) \in \mathcal{S} \times \mathcal{A}$. We neglect the dependence on $\theta$ to alleviate the notational burden.

**Value function.** For task $t$ and a policy $\pi$, we define the state-value function as $V_{t,\pi}^i(s) = \mathbb{E}_t \left[ \sum_{m=0}^\infty \gamma^m c_{t,i}(s_m, a_m, s_{m+1}) \mid s_0 = s, \pi \right]$ and the action-value function as $Q_{t,\pi}^i(s,a) = \mathbb{E}_t \left[ \sum_{m=0}^\infty \gamma^m c_{t,i}(s_m, a_m, s_{m+1}) \mid s_0 = s, a_0 = a, \pi \right]$, where $m$ denotes the time steps. Furthermore, the expected total reward/cost functions are $J_{t,i}(\pi) = \mathbb{E}_{\rho_t} \left[ V_{t,\pi}^i(s) \right] = \mathbb{E}_{\rho_t \cdot \pi} \left[ Q_{t,\pi}^i(s,a) \right]$.

**CMDP.** In each task $t$, the goal of the agent is to solve the following CMDP problem

$$\max_\pi J_{t,0}(\pi) \quad \text{s.t.} \quad J_{t,i}(\pi) \leq d_{t,i}, \quad \forall i = 1, ..., p, \tag{1}$$

where $d_{t,i}$ is a fixed limit on the expected total cost $J_{t,i}$ for task $t$ and constraint $i$ (among a total of $p$ constraints). We denote the optimal solution of 1 for the task $t$ as $\pi_t^*$ which can be non-unique.

**Primal approach.** In this work, we focus on the primal approach, CRPO (Xu et al., 2021), as an exemplary algorithm with guarantees for a single-task CMDP. CRPO is a primal-based online CMDP algorithm, which performs policy optimization (natural gradient ascent on the reward) when constraints are not violated, or constraint minimization (natural gradient descent on the constraint function) for one of the violated constraints. Specifically, with softmax parametrization and carefully chosen parameters, the suboptimality gap and constraint violation for task $t$ are bounded as follows (if the exact action-value function $\{Q_{t,\pi}^i\}_{i=0}^p$ are available for all $\pi$)[1]:

$$R_0 = J_{t,0}(\pi_t^*) - \mathbb{E}[J_{t,0}(\hat{\pi}_t)] \leq \frac{2}{\alpha_t M} \mathbb{E}_{s \sim \nu_t^*}[D_{KL}(\pi_t^* | \pi_{t,0})] + \frac{4\alpha_t c_{max}^2 |\mathcal{S}||\mathcal{A}|}{(1-\gamma)^3},$$

$$R_i = \mathbb{E}[J_{t,i}(\hat{\pi}_t)] - d_{t,i} \leq \frac{2}{\alpha_t M} \mathbb{E}_{s \sim \nu_t^*}[D_{KL}(\pi_t^* | \pi_{t,0})] + \frac{4\alpha_t c_{max}^2 |\mathcal{S}||\mathcal{A}|}{(1-\gamma)^3}, \forall i = 1, ..., p. \tag{2}$$

where $\hat{\pi}_t$ is the policy returned by running CRPO for $M$ steps with learning rate $\alpha_t$ in task $t$, $\pi_t^*$ is the optimal policy, $c_{max}$ is the upper bound on reward/cost function, and $D_{KL}(\cdot|\cdot)$ is the KL divergence.

## 2.2 META-SRL PROBLEM SETUP

We now consider the lifelong extension of CMDPs in which safe RL tasks arrive one at a time, and $t = 1, 2, \ldots, T$ denotes the index for a sequence of online learning problems. In each single task $t$, the agent must sequentially optimize the policy $\{\pi_{t,i}\}_{i=0}^M$ so that the corresponding sub-optimality and the constraint violation, given in 2, decays sub-linearly in $M$. Beyond the single task, the meta-learner should aim to optimize the upper bounds in 2 over the initial policy $\pi_{t,0}$ and the learning rate $\alpha_t$ so that the task-averaged sub-optimality and constraint violation are expected to improve as the meta-learner solves more tasks. Therefore, we will aim to minimize the task-averaged sub-optimality gap and the task-averaged constraint violation defined as follows:

**Definition 1.** *The **task-averaged optimality gap (TAOG)** $\bar{R}_0$ and the **task-averaged constraint-violation (TACV)** $\bar{R}_i$ of a safe RL algorithm after $T$ tasks are*

$$\bar{R}_0 = \frac{1}{T} \sum_{t=1}^T \left[ J_{t,0}(\pi_t^*) - \mathbb{E}[J_{t,0}(\hat{\pi}_t)] \right], \quad \bar{R}_i = \frac{1}{T} \sum_{t=1}^T \left[ \mathbb{E}[J_{t,i}(\hat{\pi}_t)] - d_{t,i} \right], \forall i = 1, ..., p, \tag{3}$$

*where $\hat{\pi}_t$ is the policy returned by running some safe RL algorithm for $M$ time-steps at task $t$ where the expectation is taken with respect to the randomness of the algorithm and environment.*

We can observe from (2) that the task-averaged regrets can be upper bounded by terms based on the policy initializations $\{\pi_{t,0}\}_{t=1}^T$ and the learning rates $\{\alpha_t\}_{t=1}^T$. The crux of our idea is to design a meta-algorithm that can sequentially update the initial policy $\pi_{t,0}$ and the learning rate of the CRPO algorithm $\alpha_t$ by performing online learning on the upper bounds, i.e., we consider the right-hand sides of (2) as the individual loss function. This enables us to bound the dynamic regrets (TAOG and TACV), which are measured against a dynamic sequence of optimal policies $\{\pi_t^*\}_{t=1}^T$, *via the static regret, which is measured against a fixed initial policy $\phi$.*

---

[1]This regret is slightly different from (Xu et al., 2021) as we assume an exact critic estimation for simplicity. For more details about the CRPO algorithm, choice of parameters, and the convergence analysis for a single task, we refer the reader to Appendices B and D.

## 2.3 TASK-SIMILARITY

In Meta-SRL, we expect TAOG and TACV to improve with the similarity among the online CMDP tasks. We now discuss the notions of similarity in a static environment; an extension to a dynamic environment is introduced in Sec. 3.2. Given optimal polices $\{\pi_t^*\}_{t=1}^T$, where $\pi_t^* \in \Pi_t^*$ for every $t$, the **task-similarity** can be measured as $D^{*2} = \min_{\phi \in \Delta(\mathcal{A})^{|\mathcal{S}|}} \frac{1}{T} \sum_{t=1}^T \mathbb{E}_{s \sim \nu_t^*}[D_{KL}(\pi_t^*|\phi)]$. If the optimal policy is not unique, we take the worst case for $D^*$, i.e., a set of policies for which $D^{*2}$ is maximum. This notion of task-similarity in the static environment is natural for studying gradient-based meta-learning, as it implies that there exists a meta initialization $\phi$ with respect to which optimal policies for individual tasks are all close together. In particular, when the tasks are all identical, i.e., $\{\pi_t^*\}_{t=1}^T$ are all equal, we have $D^{*2} = 0$. In the practical scenario where only suboptimal policies are accessible, we denote the **empirical task-similarity** as $\hat{D}^{*2} := \min_{\phi \in \Delta(\mathcal{A})^{|\mathcal{S}|}} \frac{1}{T} \sum_{t=1}^T \mathbb{E}_{s \sim \hat{\nu}_t}[D_{KL}(\hat{\pi}_t|\phi)]$, which depends on the suboptimal policies $\{\hat{\pi}_t\}_{t=1}^T$ returned by a within-task algorithm. As it is desirable that the meta-initialization policy $\pi_{t,0}$ has good exploration properties, we need the initial policy to have full support over $\mathcal{S} \times \mathcal{A}$. We introduce the following assumption:

**Assumption 1.** *The meta-initialization policy $\pi_{t,0}$ for any task $t$ lies inside a **shrinkage simplex set**, i.e., $\pi_{t,0}(\cdot|s) \in \Delta\mathcal{A}_\varrho := \{a_1 e_1 + ... + a_{n_a} e_{n_a} | \sum_{i=0}^{n_a} a_i = 1, a_i \geq \varrho \quad \forall i = 1, ..., n_a\}$ for all $s \in \mathcal{S}$, where $e_1, ..., e_{n_a} \in \mathbb{R}^{n_a}$ are one-hot vectors for each action (e.g., $e_1$ is the vector of all $0$s except $1$ at the first location) and $\varrho > 0$. In particular, $\Delta\mathcal{A}_\varrho$ lies inside the regular simplex set $\Delta\mathcal{A}$ ($\varrho = 0$).*

Technically, Assumption 1 is a minimal requirement for the CRPO to provide any guarantees in a single task. This can be seen in (2): if $\pi_{t,0}$ does not have full support over the state/action space, then there may be a state $s$ and an action $a$ where $\pi_t^*(a|s) > 0$ but $\pi_{t,0}(a|s) = 0$, which would make the KL divergence term in (2) infinite. Furthermore, Assumption 1 ensures the following holds for the meta-initialization policy $\pi_{t,0}$ for any state $s \in \mathcal{S}$ with positive constants $C_\pi, L_g, L_\pi$ and $\mu_\pi$: (1) $|D_{KL}(\pi_t^*(\cdot|s)|\pi_{t,0}(\cdot|s))|, |D_{KL}(\hat{\pi}_t(\cdot|s)|\pi_{t,0}(\cdot|s))| \leq C_\pi$; (2) $D_{KL}(\pi_t^*(\cdot|s)|\pi_{t,0}(\cdot|s))$ is $L_g$-Lipschitz and $L_\pi$-smooth in $\pi_{t,0}(\cdot|s)$; (3) $D_{KL}(\pi_t^*(\cdot|s)|\pi_{t,0}(\cdot|s))$ is $\mu_\pi$-strongly convex in $\pi_{t,0}(\cdot|s)$. We use these conditions in the proof of Lemma 1, Lemma 3, and Theorem 3.2.

In this work, we develop algorithms whose TAOG and TACV scale with the task-similarity, which implies that the method will do well if tasks are similar. To understand the CMDP-within-online framework and the impact of task-similarity on the upper bounds of TAOG and TACV for Meta-SRL, we first present a simplified result under the ideal setting where $\{\nu_t^*\}_{t=1}^T$ and $\{\pi_t^*\}_{t=1}^T$ are available for each task $t$ and the task-similarity $D^{*2}$ is known.

**Lemma 1.** *Assume $\{\nu_t^*\}_{t=1}^T$ and $\{\pi_t^*\}_{t=1}^T$ are given after each task and the task-similarity $D^{*2}$ is known. For each task $t$, we run CRPO for $M$ iterations with $\alpha = \frac{(1-\gamma)^{\frac{3}{2}}}{\sqrt{2M|\mathcal{S}||\mathcal{A}|}} \sqrt{\frac{L_g^2 (\log T+1)}{\mu_\pi T} + D^{*2}}$. In addition, the initialization $\{\pi_{t,0}\}_{t=1}^T$ are determined by playing Follow-the-Regularized-Leader (FTRL) or online gradient descent (OGD) on the functions $\mathbb{E}_{s \sim \nu_t^*}[D_{KL}(\pi_t^*|\cdot)]$, for $t = 1, ..., T$.[2] Then, it holds that*

$$\bar{R}_0 \leq \mathcal{O}\left(\frac{1}{\sqrt{M}} \sqrt{\frac{\log T}{T} + D^{*2}}\right), \bar{R}_i \leq \mathcal{O}\left(\frac{1}{\sqrt{M}} \sqrt{\frac{\log T}{T} + D^{*2}}\right) \forall i = 1, ..., p.$$

The above result reveals an interesting benefit brought by including more tasks (the regret decays at a rate of $\log(T)/T$ with more similarity (i.e., lower $D^*$), which improves upon single-task guarantee and serves as the initial point of our study. However, there are several limitations. First, if the optimal policies $\pi_t^*$ and the induced state distributions $\nu_t^*$ are not revealed after each task, it is not likely that the plug-in estimator $\{\mathbb{E}_{s \sim \hat{\nu}_t}[D_{KL}(\hat{\pi}_t|\cdot)]\}_{t=1}^T$ with the learned policy $\hat{\pi}_t$ and estimated visitation distribution $\hat{\nu}_t$ is an unbiased estimator, ruling out existing analysis for FTRL or OGD in the bandit setting. Besides, while the knowledge of $D^*$ used to determine the learning rate can be relaxed (Khodak et al., 2019; Balcan et al., 2019), the resulting scheme is complex to implement—we expect that the learning rates can be chosen *adaptively* for different tasks based on losses observed in the past. We aim to address these challenges with a series of developments in the next section.

---

[2]When online learning is played on $\mathbb{E}_{s \sim \nu_t^*}[D_{KL}(\pi_t^*|\pi)]$ to determine $\pi_{t+1,0}$, we treat $\theta$ in $\pi_\theta$ for all $s \in \mathcal{S}$ and $a \in \mathcal{A}$ as the decision variable. For simplicity, we will refer to $\pi$ as the decision variable.

---

**Algorithm 1:** Inexact CMDP-within-online framework (exemplified with CRPO (Xu et al., 2021) as the within-task safe RL algorithm)

---

1: Initialize actor policy $\pi_{1,0}$ and learning rate $\alpha_1$
2: **for** task $t \in [T]$ **do**
3:     Run CRPO with initializations for actor policy $\pi_{t,0}$ and learning rates $\alpha_t$ to obtain a policy $\hat{\pi}_t$
4:     Estimate the discounted state visitation distribution $\hat{\nu}_t$ of $\hat{\pi}_t$ based on trajectory data collected within-task $t$ with DualDICE (Nachum et al., 2019)
5:     Run one or multiple steps of OGD on
       (a)   $INIT$: $\hat{f}_t^{init}(\phi) = \mathbb{E}_{\hat{\nu}_t}[D_{KL}(\hat{\pi}_t|\phi)]$.
       (b)   $SIM$: $\hat{f}_t^{sim}(\kappa) = \frac{c_1^t \mathbb{E}_{\hat{\nu}_t}[D_{KL}(\hat{\pi}_t|\pi_{t,0})]}{\kappa} + \kappa(c_2^t M + c_4^t \sqrt{M}) + c_3^t \sqrt{M}$
    to obtain $\pi_{t+1,0}$ and $\alpha_{t+1}$. Here $c_1^t = 2$, $c_2^t = \frac{4c_{max}^2 |\mathcal{S}||\mathcal{A}|}{(1-\gamma)^3}$, $c_3^t = \frac{3+(1-\gamma)^2}{(1-\gamma)^2}$, and $c_4^t = \frac{3c_{max}}{(1-\gamma)^2}$.
6: **end for**

---

# 3 PROVABLE GUARANTEES FOR PRACTICAL CMDP-WITHIN-ONLINE FRAMEWORK

## 3.1 INEXACT CMDP-WITHIN-ONLINE FRAMEWORK

One of the key steps to generalize the online-within-online methodology (Balcan et al., 2019) to Meta-SRL is to relax the assumption of accessing the exact upper bounds of within-task performance by designing algorithms to estimate and update their inexact versions.

**Estimation of upper bounds.** Once a CMDP task $t$ is complete, the meta-learner only has access to a suboptimal policy $\hat{\pi}_t$ and the trajectory dataset $\mathcal{D}_t$ produced by some safe RL algorithm. Let $\tilde{\nu}_t$ denote the discounted state visitation distribution induced by policy $\hat{\pi}_t$. To obtain an estimate $\hat{\nu}_t$ from $\mathcal{D}_t$, recent methods often rely on estimating discounted state visitation distribution corrections (Liu et al., 2018; Gelada & Bellemare, 2019). However, the main issues are that $\mathcal{D}_t$ is collected by multiple behavior policies during the learning period, and depending on how far these behavior policies are from the target policy, the per-step importance ratios involved in these methods may have large variance, which may result in a detrimental effect on stochastic algorithms. In this work, we use a methods from the distribution correction estimation (DICE) family, namely DualDICE (Nachum et al., 2019), which is agnostic to the number of behavior policies used and does not involve any per-step importance ratios, thus is less likely to be affected by their high variance. In particular, for each state-action pair $(s, a)$, the method aims to estimate the quantity $\omega_{\pi/\mathcal{D}_t}(s, a) = \frac{d^\pi(s,a)}{d^{\mathcal{D}_t}(s,a)}$, i.e., the likelihood that the target policy $\pi$ will experience the state-action pair normalized by the probability with which the state-action pair appears in the off-policy data $\mathcal{D}_t$. Thereby, we estimate $\mathbb{E}_{\nu_t^*}[D_{KL}(\pi_t^*|\pi)]$ with $\mathbb{E}_{\hat{\nu}_t}[D_{KL}(\hat{\pi}_t|\pi)]$ by plugging in $\hat{\pi}_t$ from the within-task CMDP and $\hat{\nu}_t$ from DualDICE in lieu of the optimal policy $\pi_t^*$ and the corresponding discounted state visitation distribution $\nu_t^*$.

**Bounding the estimation error.** We breakdown the error by sources of origin:

$$\mathbb{E}_{\nu_t^*}[D_{KL}(\pi_t^*|\pi)] - \mathbb{E}_{\hat{\nu}_t}[D_{KL}(\hat{\pi}_t|\pi)] = \underbrace{\mathbb{E}_{\nu_t^*}[D_{KL}(\pi_t^*|\pi)] - \mathbb{E}_{\tilde{\nu}_t}[D_{KL}(\pi_t^*|\pi)]}_{(A)} \quad (4)$$

$$+ \underbrace{\mathbb{E}_{\tilde{\nu}_t}[D_{KL}(\pi_t^*|\pi)] - \mathbb{E}_{\hat{\nu}_t}[D_{KL}(\pi_t^*|\pi)]}_{(B)} + \underbrace{\mathbb{E}_{\hat{\nu}_t}[D_{KL}(\pi_t^*|\pi)] - \mathbb{E}_{\hat{\nu}_t}[D_{KL}(\hat{\pi}_t|\pi)]}_{(C)},$$

where $(A)$ accounts for the mismatch between the discounted state visitation distributions of an optimal policy $\pi_t^*$ and a suboptimal one $\hat{\pi}_t$, $(B)$ originates from the estimation error of DualDICE, and $(C)$ is due to the difference between $\pi_t^*$ and $\hat{\pi}_t$ measured according to $\hat{\pi}_t$.

To bound $(A)$, we need to control the distance between $\nu_t^*$ and $\tilde{\nu}_t$, which can be bounded by the distance between the inducing policy parameters as long as they are Lipschitz continuous (Xu et al., 2020, Lemma 3). In addition, the bound on $(C)$ also depends on the distance between policies. Controlling the distance between a policy to an optimal policy based on the suboptimality gap requires the optimization to have some curvatures around the optima (e.g., Hölderian growth (Johnstone & Moulin, 2020)). However, to the best of our knowledge, available results are algorithm-dependent PL

inequalities for policy gradient (Mei et al., 2020) or quadratic growth with entropy regularization (Ding et al., 2021b). Given some mild assumptions on the objective/constraint functions and policy parametrization, we can show that a growth condition holds broadly for any CMDP problem.

**Assumption 2.** *The functions $J_{t,i}(\cdot)$ for $i = 0, 1, ..., p$ and $t \in [T]$ and parametric policy $\pi_\theta$ are definable in some o-minimal structure (Van den Dries & Miller, 1996).*

The definition of "o-minimal structure" is given in Appendix F.1. Assumption 2 is mild as practically all functions from real-world applications, including deep neural networks, are definable in some o-minimal structures. For Assumption 2 to hold, a sufficient condition requires that the reward and utility functions belong to the same o-minimal structure. We use Assumption 2 to bound the terms $(A)$ and $(C)$ as definable sets admit the property of Whitney stratification, and any stratifiable function enjoys a nonsmooth Kurdyka-Lojasiewicz inequality, which implies some curvature around the local/global minima. Details on tame geometry and proof of Theorem 3.1 is given in Appendix F.

**Theorem 3.1** (KL divergence estimation error bound). *The following bound holds:*

$$|\mathbb{E}_{\nu_t^*}[D_{KL}(\pi_t^*|\pi)] - \mathbb{E}_{\hat{\nu}_t}[D_{KL}(\hat{\pi}_t|\pi)]|$$
$$\leq \mathcal{O}\left(h\left(\frac{1}{\sqrt{M}}\right) + \frac{1}{\sqrt{M}} + \sqrt{\epsilon_{opt}} + \sqrt{\epsilon_{approx}(\mathcal{F}, \mathcal{H})}\right) = \epsilon_t,$$

*where $h$ is a strictly increasing continuous function with the property that $h(0) = 0$ as specified in Proposition 1, $\epsilon_{approx}(\mathcal{F}, \mathcal{H})$ and $\epsilon_{opt}$ are the approximation error and optimization error of DualDICE, defined in Equation 38 and Equation 39, respectively.*

**Remark 1.** *We define cumulative inexactness $\mathcal{E}_T := \sum_{t=1}^{T} \epsilon_t$. This quantity decays with $M$ at a rate of $\mathcal{O}\left(h\left(\frac{1}{\sqrt{M}}\right) + \frac{1}{\sqrt{M}}\right)$ up to some approximation and optimization errors $\epsilon_{approx}$ and $\epsilon_{opt}$. Moreover, there is a trade-off between $\epsilon_{approx}$ and $\epsilon_{opt}$: if the parametrization functions $\mathcal{F}$ and $\mathcal{H}$ used to solve DualDICE optimization are chosen as neural networks, then $\epsilon_{approx}$ can be reduced at the cost of increasing $\epsilon_{opt}$. If we use stochastic gradient descent as an optimization algorithm in DualDICE with $K$ steps, then $\epsilon_{opt}$ decays at a rate of $\mathcal{O}(1/K)$. Note that $h$ is a definable function used in the Kurdyka–Łojasiewicz (KL) inequality (see, e.g., (Bolte et al., 2007, Thm. 14)).*

With the above uniform bound on estimation error, our next step is to develop static regret bounds for the inexact online gradient descent, which are used to furnish the upper bounds on TAOG and TACV of the proposed inexact CMDP-within-online algorithm.

**Lemma 2** (Static regret bound for inexact OGD). *Denote $f_t(\pi_{t,0}) := \mathbb{E}_{\nu_t^*}[D_{KL}(\pi_t^*|\pi_{t,0})]$ for all $t \in [T]$. For any fixed comparator $\pi_0^* = \arg\min_{\pi_0 \in \Delta \mathcal{A}_\varrho^{|\mathcal{S}|}} \sum_{t=1}^{T} f_t(\pi_0)$, if OGD is run on a sequence of loss functions $\{\hat{f}_t\}_{t \in [T]}$, where $\hat{f}_t(\pi_{t,0}) := \mathbb{E}_{\hat{\nu}_t}[D_{KL}(\hat{\pi}_t|\pi_{t,0})]$ for all $t \in [T]$ with the step-size of $\mathcal{O}(1/\sqrt{T})$, then the following bound holds for static regret:*

$$\sum_{t=1}^{T} f_t(\pi_{t,0}) - \sum_{t=1}^{T} f_t(\pi_0^*) \leq \mathcal{O}\left(\sqrt{T} + \mathcal{E}_T\right),$$

*where $\mathcal{E}_T := \sum_{t=1}^{T} \epsilon_t$ is the cumulative inexactness, and $\epsilon_t$ is the upper bound from Theorem 3.1.*

The static regret analyzed above is defined with respect to the optimal *initial policy* $\pi_0^*$ in hindsight, not the *final learned policy*. A static regret with respect to an initial policy provides freedom for the safe RL algorithm to adapt the initial policy based on observations within the task. Once the static regret for the inexact OGD is established, we can obtain the upper bounds on TAOG and TACV for the proposed inexact CMDP-within-online algorithm in terms of the empirical task-similarity $\hat{D}^*$.

**Theorem 3.2.** *For each task $t$, we run CRPO for $M$ iterations with $\alpha = \mathcal{O}\left(\frac{1}{\sqrt{M}}\sqrt{\frac{1}{\sqrt{T}} + \frac{\mathcal{E}_T}{T} + \hat{D}^{*2}}\right)$ and we obtain $\{\hat{\nu}_t\}_{t=1}^{T}$ and $\{\hat{\pi}_t\}_{t=1}^{T}$. In addition, the initialization $\{\pi_{t,0}\}_{t=1}^{T}$ for each task $t$ are determined by playing inexact OGD (Algorithm 2) on $\mathbb{E}_{\hat{\nu}_t}[D_{KL}(\hat{\pi}_t|\cdot)]$, for $t = 1, \ldots, T$. Then, the following holds for TAOG ($i = 0$) and TACV ($i = 1, ..., p$):*

$$\bar{R}_i \leq \mathcal{O}\left(\frac{1}{\sqrt{M}}\left(\sqrt{\frac{1}{\sqrt{T}} + \frac{\mathcal{E}_T}{T} + \hat{D}^{*2}}\right)\right) \forall i = 0, 1, \ldots, p.$$

The benefit of task-similarity is preserved when we perform directly on the plug-in estimator $\mathbb{E}_{\hat{\nu}_t}[D_{KL}(\hat{\pi}_t|\cdot)]$, though we incur an additional cost on the inexactness $\mathcal{E}_T$ and the dependence on $T$ is worse compared to Lemma 1. As $\mathcal{E}_T/T$ diminishes when the learned policy becomes optimal across tasks, e.g., by increasing within-task steps $M$ or if meta-initialization is chosen such that a few steps suffice to reach optimal, we expect the inexactness to have a limited effect on the performance.

## 3.2 DYNAMIC REGRET AND TASK-RELATEDNESS

In many settings, we have a changing environment, so it is natural to study dynamic regret and compare with a sequence of potentially time-varying *initial policies* $\{\psi_t^*\}_{t=1}^T$. To measure task-similarity in this case, we define **task-relatedness** which can be measured by $V_\psi^2 = \frac{1}{T}\sum_{t=1}^T \mathbb{E}_{s\sim\nu_t^*}[D_{KL}(\pi_t^*|\psi_t^*)]$. This notion of task-relatedness gives the measure of how far optimal policies are in each task from some time-varying comparator. We denote **empirical task-relatedness** as $\hat{V}_\psi^2 := \frac{1}{T}\sum_{t=1}^T \mathbb{E}_{s\sim\hat{\nu}_t}[D_{KL}(\hat{\pi}_t|\psi_t^*)]$, which depends on the suboptimal policy returned by the within-task algorithm. To measure the performance of Meta-SRL in dynamic settings, we analyze the dynamic regret bound, i.e., $U_T := \sum_{t=1}^T f_t(\phi_t) - \sum_{t=1}^T f_t(\psi_t^*)$, where $\psi_t^* \in \arg\min_{x\in\mathcal{X}} f_t(x)$ is a sequence of minimizers for each loss, and $f_t(\cdot) = \mathbb{E}_{s\sim\nu_t^*}[D_{KL}(\pi_t^*|\cdot)]$. By exploiting the strong convexity of the loss function (KL divergence in our case), previous studies have shown that the dynamic regret can be upper bounded by the path-length of the comparator sequence, defined as $\mathcal{P}_T := \sum_{t=2}^T \|\psi_t^* - \psi_{t-1}^*\|$, which captures the cumulative difference between successive comparators (Zhao et al., 2020). The bound can be further improved for strongly convex functions as the minimum of the path-length and the squared path-length, $\mathcal{S}_T := \sum_{t=2}^T \|\psi_t^* - \psi_{t-1}^*\|^2$, which can be much smaller than the path-length (Zhang et al., 2017). We extend these results to the settings of inexact online gradient descent by allowing the learner to query the inexact gradient of the function.

**Lemma 3** (Dynamic regret bound for inexact OGD). *Denote $f_t(\phi_t) := \mathbb{E}_{\nu_t^*}[D_{KL}(\pi_t^*|\phi_t)]$ for all $t \in [T]$. For any dynamically varying comparator $\psi_t^*$, if single-step inexact OGD is run with the step-size $\beta \leq \frac{1}{2\mu_\pi}$ on a sequence of loss functions $\{\hat{f}_t\}_{t\in[T]}$, where $\hat{f}_t(\phi_t) = \mathbb{E}_{\hat{\nu}_t}[D_{KL}(\hat{\pi}_t|\phi_t)]$, then the following bound holds for dynamic regret:*

$$\sum_{t=1}^T f_t(\phi_t) - \sum_{t=1}^T f_t(\psi_t^*) \leq \mathcal{O}\left(\min(\mathcal{S}_T + \mathcal{E}_T, \mathcal{P}_T + \tilde{\mathcal{E}}_T)\right),$$

*where $\mathcal{P}_T := \sum_{t=2}^T \|\psi_t^* - \psi_{t-1}^*\|$ is the path-length of the comparator sequence, $\mathcal{S}_T := \sum_{t=2}^T \|\psi_t^* - \psi_{t-1}^*\|^2$ is the squared path-length, $\mathcal{E}_T := \sum_{t=1}^T \epsilon_t$ is the cumulative inexactness, $\tilde{\mathcal{E}}_T := \sum_{t=1}^T \sqrt{\epsilon_t}$ is the cumulative square root of inexactness, and $\epsilon_t$ is the upper bound from Theorem 3.1.*

## 3.3 DYNAMIC REGRET WITH ADAPTIVE LEARNING RATES

It can be observed from the last section that to set the learning rate $\alpha_t$ for the within-task algorithm CRPO, knowledge of optimal/suboptimal policies from all $T$ tasks is used. This makes the algorithm less applicable in online settings where tasks are encountered sequentially. Moreover, when the task-environment changes dynamically, a fixed policy initialization $\psi$ may not be the best candidate comparator, where it is natural to study dynamic regret by competing with a potentially time-varying sequence $\{\psi_t^*\}_{t=1}^T$. Also, the tasks may share some common aspects of the optimization landscape, so adapting learning rates based on prior experience may further improve performance. This is the direction we pursue. Recall the regret for suboptimality and constraint violation of the CRPO:

$$U_t(\pi_{t,0}, \alpha_t) := \frac{c_1^t}{\alpha_t}\mathbb{E}_{s\sim\nu_t^*}[D_{KL}(\pi_t^*|\pi_{t,0})] + \alpha_t(c_2^t M + c_4^t\sqrt{M}) + c_3^t\sqrt{M}, \tag{5}$$

where the constants $\{c_i^t\}_{i=1,...,4}$ are given in Algorithm 1. We assume that $\alpha_t \in \Lambda := \{\alpha_t \mid \alpha_t \geq \zeta\}$ for some $\zeta > 0$, where $\Lambda$ is a convex set. Overall, the goal of the meta-learner is to make a sequence of decisions, collected by $x_t = \{\pi_{t,0} \in \Delta\mathcal{A}_\varrho^{|\mathcal{S}|}, \alpha_t \in \Lambda\}$, such that TAOG and TACV are minimized.

To design the adaptive algorithm, we consider the following two parallel sequences of loss functions over initial policy $\phi$, $f_t^{init}(\phi) = \mathbb{E}_{\nu_t^*}[D_{KL}(\pi_t^*|\phi)]$, and learning rate $\kappa$,

$$f_t^{sim}(\kappa) = \frac{c_1^t \mathbb{E}_{\nu_t^*}[D_{KL}(\pi_t^*|\pi_{t,0})]}{\kappa} + \underbrace{\kappa(c_2^t M + c_4^t\sqrt{M}) + c_3^t\sqrt{M}}_{f_t^{rate}(\kappa)}.$$

Note that $f_t^{sim}(\alpha_t) = U_t(\pi_{t,0}, \alpha_t)$ matches the upper bound in 5. We also denote the inexact versions $\hat{f}_t^{init}(\phi)$ and $\hat{f}_t^{sim}(\kappa)$ by replacing $\mathbb{E}_{\nu_t^*}[D_{KL}(\pi_t^*|\phi)]$ with $\mathbb{E}_{\hat{\nu}_t}[D_{KL}(\hat{\pi}_t|\phi)]$ in the above. Inspired by (Khodak et al., 2019), instead of running one online algorithm on $U_t(\pi_t, \alpha_t)$, we will run two online algorithms separately for the function sequences $\hat{f}_t^{init}$ and $\hat{f}_t^{sim}$ by taking actions on the initial policy and learning rates, respectively, such that the overall regret can be bounded by an expression that depends on the regrets for each sequence. Let INIT and SIM be two algorithms, such that the actions $\pi_{t+1,0} := \text{INIT}(t)$ are taken over $\hat{f}_t^{init}$ and the actions $\alpha_{t+1} := \text{SIM}(t)$ are taken over $\hat{f}_t^{sim}$; these actions will then be used as policy initialization and learning rates for the next CMDP. We assume the following regret upper bounds for each algorithm:[3]

1. $U_T^{init}(\{\psi_t^*\}_{t=1}^T)$: upper bound on the dynamic regret for INIT over functions $\{\hat{f}_t^{init}\}_{t=1}^T$ with respect to a time-varying sequence $\{\psi_t^*\}_{t=1}^T$;

2. $U_T^{sim}(\kappa)$: upper bound on the static regret for SIM over functions $\{\hat{f}_t^{sim}\}_{t=1}^T$ with respect to a comparator $\kappa > 0$.

**Theorem 3.3.** *Let each within-task CMDP $t$ run $M$ steps of CRPO, initialized by policy $\pi_{t,0} :=$ INIT(t) and learning rates $\alpha_t := \text{SIM}(t)$. Let $\kappa^* := \arg\min L(\kappa)$, where*

$$L(\kappa) = U_T^{sim}(\kappa) + \frac{U_T^{init}(\{\psi_t^*\}_{t=1}^T)}{\kappa} + \frac{\mathcal{E}_T}{\kappa} + \sum_{t=1}^T \left[ \frac{\hat{f}_t^{init}(\psi_t^*)}{\kappa} + f_t^{rate}(\kappa) \right], \qquad (6)$$

*and $\{\psi_t^*\}_{t=1}^T$ is any comparator sequence. Then, the following bounds on TAOG and TACV hold:*

$$\bar{R}_i \leq \frac{L(\kappa^*)}{T}, \qquad \forall\, i = 0, ..., p. \qquad (7)$$

Note that the terms $U_T^{init}$ and $U_T^{sim}$ are simply placeholders for upper bounds on the respective regrets for some inexact online algorithms. In particular, INIT and SIM can be any inexact online algorithms in Algorithm 1, and the results of Theorem 3.3 can be instantiated by plugging in the respective $U_T^{init}$ and $U_T^{sim}$. The following corollary presents the TAOG, and TACV regret bounds when INIT and SIM are inexact OGD over the loss functions $\{\hat{f}_t^{init}\}_{t=1}^T$ and $\{\hat{f}_t^{sim}\}_{t=1}^T$ respectively.

**Corollary 1.** *If* INIT(t) *and* SIM(t) *are inexact OGD, and are run over the sequences $\{\hat{f}_t^{init}\}_{t=1}^T$ and $\{\hat{f}_t^{sim}\}_{t=1}^T$, then, the following bounds on TAOG and TACV hold for all $i = 0, \ldots, p$:*

$$\bar{R}_i \leq \mathcal{O}\left( \frac{1}{\sqrt{M}} \left( \frac{1}{\sqrt{MT}} + \frac{\mathcal{E}_T}{T\sqrt{M}} + \frac{1}{M^{1/4}\sqrt{T}} \sqrt{\frac{\min(\mathcal{S}_T + \mathcal{E}_T, \mathcal{P}_T + \tilde{\mathcal{E}}_T) + \mathcal{E}_T}{T} + \hat{V}_\psi^2} \right) \right). \qquad (8)$$

**Remark 2.** *The bounds are improved in terms of $M$ and $T$ due to the adaptive learning rate. Specifically, the bounds diminish at a rate $\mathcal{O}\left( \frac{1}{M^{3/4}\sqrt{T}} \left( \mathcal{E}_T + \sqrt{\frac{\mathcal{E}_T}{T} + \hat{V}_\psi^2} \right) \right)$ as compared to the previous rate $\mathcal{O}\left( \frac{1}{\sqrt{M}} \left( \sqrt{\frac{\mathcal{E}_T}{\sqrt{T}} + \hat{D}^{*2}} \right) \right)$. Note that $\hat{V}_\psi$ is same as $\hat{D}^*$ in the case of a fixed comparator $\psi^*$. Moreover, a practical aspect of our algorithm is that it does not require the knowledge of quantities like $\mathcal{S}_T, \mathcal{P}_T$ and $\mathcal{E}_T$ to decide the value of learning rate $\alpha_t$.*

## 4 EXPERIMENTS

In this section, we show the effectiveness of the proposed Meta-SRL framework and compare it with the following baselines: simple averaging (i.e., initialize with the average of learned policies from past CMDPs), pre-trained (i.e., initialize test task with the suboptimal policy from another CMDP), Follow the Average Leader (FAL), and random initialization strategies. Note that simple averaging takes the average of previous suboptimal policies obtained from random initializations on all CMDP tasks, while FAL does this in an online manner while tasks arrive sequentially. Different CMDPs are

---

[3]While we run the online learning algorithm on the inexact versions of the loss $\{\hat{f}_t\}_{t=1}^T$, the dynamic/static regret is the standard one measured using the exact losses: $U_T = \sum_{t=1}^T f_t(\phi_t) - \sum_{t=1}^T f_t(\psi_t^*)$.

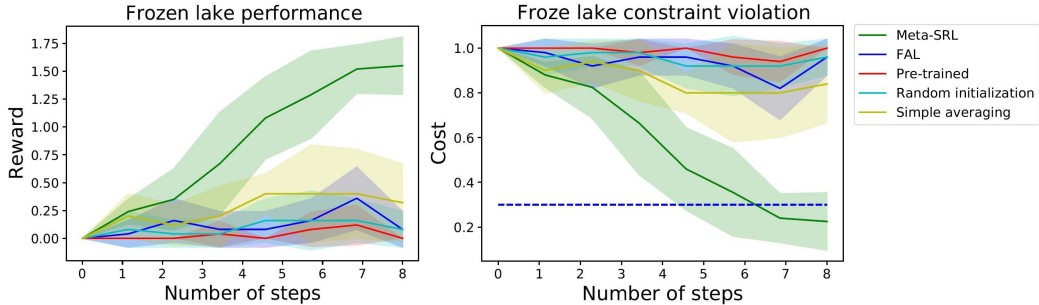

Figure 1: Frozen lake results for reward maximization and constraint violations when the task-relatedness is low. The Blue dashed line represents the averaged thresholds for the constraint violations. We do 10 runs on each baseline to get the performance plots with variance.

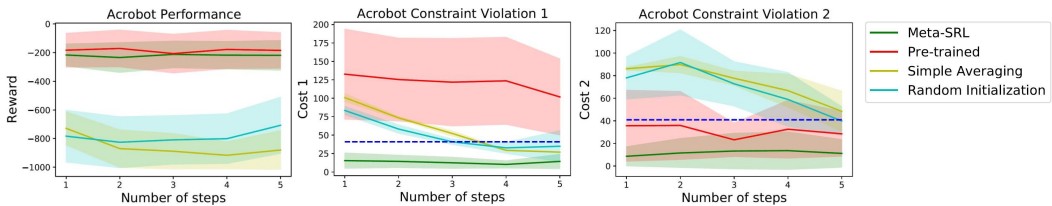

Figure 2: Acrobot results for reward maximization and constraint violations when the task-relatedness is low. Blue dashed line represents the averaged thresholds for the constraint violations.

generated using a probability distribution over the parameters of CMDPs (e.g., rewards, transition dynamics), similar to the latent CMDP model (Chen et al., 2021a). We consider the Frozen lake, acrobot, half-Cheetah, and humanoid environments from the OpenAI gym (Brockman et al., 2016) and MuJoco Todorov et al. (2012) under constrained settings. For more details on experimental setups, distribution shift, and extra experiments on Mujoco, please refer to Appendix H.

We can observe from Figure 1 that Meta-SRL achieves higher rewards and lower constraint violations more quickly than baseline initializations. The baseline FAL which simply takes the average of previous suboptimal policies, performs poorly. This illustrates the benefit of incorporating stationary distribution correction estimation and adaptive learning rates. Indeed, for Frozen lake, different locations of the hole can result in different stationary distributions—it is more sensible to put higher weights on policies that frequently visit a particular state since it implies that the corresponding strategies can have a substantial impact on the case of low task-similarity conditions. We also observe similar trends for the Acrobot in Figure 2, where Meta-SRL achieves higher rewards quickly and zero constraint violations as compared to other baseline initializations under low task-relatedness settings. The pre-trained baseline was able to achieve higher rewards but did not achieve constraint satisfaction for both constraints. Under high task-similarity settings, we expected all the methods (except vanilla CRPO) to perform well; however, we noticed that simple averaging does poorly even in this setting, possibly due to adverse interference among different tasks.

## 5 CONCLUSION AND FUTURE DIRECTIONS

We introduced a novel framework, Meta-SRL, for meta-learning over CMDPs. The proposed framework does not assume access to globally optimal policies from the training tasks, and instead performs online learning over inexact within-task bounds estimated by stationary distribution correction. Moreover, strategies for learning rate adaptation are designed to further exploit task-relatedness. One limitation of the proposed method is that it only considers CRPO as the within-task algorithm; nevertheless, our framework can be potentially adapted to more single-task algorithms by making the dependence of guarantees on initial policy/step sizes explicit. Some potential future directions could be to design Meta-SRL with zero constraint violation (Liu et al., 2021b), non-stationary environments (Ding & Lavaei, 2022), and multi-agent settings (De Nijs et al., 2021).

## 6 Broader Impact Statements

There is an increasing need to address fairness as a constraint in learning settings. Existing works that aim to achieve zero-shot generalization without any task-specific adaptation have limited capability to adapt to shifting environments. While online meta-learning is a principled technique to learn good priors over model parameters for fast adaptation in a sequential setting, existing methods often do not address constraints and thus have limited applications in fairness-aware learning.

The proposed CMDP-within-online framework can potentially be adapted to reinforcement learning tasks with fairness constraints in a **non-stationary environment**. In practice, this can provide a strategy that learns priors over policy parameters not only to master the current fairness-aware task but also to become proficient with quick adaptation at learning newly arrived tasks. Our theoretical analysis can be leveraged to provide a sublinear bound on the "task-averaged fairness violation" regret. Similar ideas have been explored by (Zhao et al., 2021) in the supervised learning setting, while we are not aware of any work on the reinforcement learning counterpart. Thus, it can be an extension for future work to explore the extent to which our method can address this important problem.Nevertheless, fairness constraints present a unique challenge for meta-safe RL settings, as fairness constraints should rarely be violated in a real-world setting due to the implicated discrimination or bias. Additional efforts, such as incorporating pessimism Bai et al. (2021) or developing offline methods, may be entailed to reduce fairness violations during initial deployment.

## 7 Acknowledgments

The authors acknowledge the generous support by NSF, the Commonwealth Cyber Initiative (CCI), 4-VA collaborative research grant, C3.ai Digital Transformation Institute, and the U.S. Department of Energy. We would also like to thank the anonymous reviewers, which helped us to improve the quality of the manuscript and Tengyu Xu for providing the code for CRPO.

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

APPENDIX

In this section, we start with a summary of related work in Sec. A followed by a brief recapitulation of the CRPO algorithm in Sec. B. Note that CRPO will be our focus as the exemplary within-task safe RL algorithm. We also introduce notations therein that will be used in later analysis. In Sec. C, we give the pseudo-code of our inexact CMDP-within-online algorithm, with further discussions on key aspects. Sec. D provides the proof for Sec. 2 of the main paper, which focuses on an elementary yet illustrative example of the CMDP-within-online approach. We start by providing a simplified proof to help the reader understand the main approach of CRPO, and then demonstrate the potential improvement by exploiting inter-task-relatedness (Lemma 5). Sec. E contains the key developments in extending online learning approaches, specifically online gradient descent, to the case of inexact loss functions. We start with some preliminaries on $\epsilon$-subgradient (Sec. E.1). Then, we conduct the analysis for static regret (Thm. E.1) and dynamic regret (Thm. E.3). In Sec. F, we provide a detailed analysis of the KL divergence estimation error bound, which contributes to one of our main contributions in understanding the key aspects of the proposed inexact CMDP-within-online framework. Our development leverages the seminal results developed for tame geometry, which we briefly review in Sec. F.1. We also briefly set up the notations and recall basic properties of subgradient flow systems F.2. Through a series of bounds, the final result is obtained in Thm. F.1. We then provide proofs for Sec. 3.3. In Sec. G.1, we first extend the analysis of CRPO to the case of adaptive learning rates. Then, we provide the proof for Thm. 3.3 and Corollary 1 in Sec. G.2. Experimental details are provided in Sec. H. Frequently used notations and constants are listed in Sec. J.

## A    RELATED WORK

**Meta-reinforcement learning:** Current state-of-the-art meta-RL includes learning the initial conditions (Finn et al., 2017), hyperparameters (Jaderberg et al., 2019), step directions (Li et al., 2017) and stepsizes (Young et al., 2018), and training recurrent neural networks to embed previous task experience (Duan et al., 2016) (see also (Chen et al., 2021a) for sim-to-real transfer, and Suilen et al. (2022) for the extension to robust MDPs), with recent developments on improving meta-optimization (Rothfuss et al., 2018; Liu et al., 2019; Song et al., 2019) (see (Hospedales et al., 2020) for a review). Recently, (Fallah et al., 2021; Ji et al., 2022) provided theoretical studies on the convergence of model-agnostic meta-RL. However, these works all focus on the unconstrained meta-RL and their local optimality convergence, while our work is the first to obtain provable guarantees for optimality and constraint satisfaction for CMDPs.

**Online meta-learning/learning-to-learn (LTL).** Most initialization-based meta-learning studies focus on the setting with decomposable within-task loss functions that are often convex (Finn et al., 2019; Denevi et al., 2019; Balcan et al., 2019); nonconvex within-task settings are studied usually for multi-task representation learning (Balcan et al., 2015; Maurer et al., 2016; Du et al., 2020; Tripuraneni et al., 2020). Theoretically, our work is inspired by the Average Regret-Upper-Bound Analysis (ARUBA) strategy (Khodak et al., 2019) for obtaining a meta-procedure, which has been recently extended to learning nonconvex piecewise-Lipschitz functions (Balcan et al., 2021); the main technical advance in our work is in providing the guarantees for CMDPs, which is challenging due to the interplay between the nonconvexity and stochasticity of the optimization and the complexity of the within-task safe RL algorithms that involve policy update, critic learning, and the proper choice of stepsizes for reward/constraints.

**Inexact online learning.** Online learning with access to inexact loss/gradient information has been studied for stochastic zero-biased noise (Cesa-Bianchi et al., 2011; Yang et al., 2016; Bedi et al., 2018; Dixit et al., 2019), deterministic error/nonzero-biased stochastic noise (Bedi et al., 2018; Dixit et al., 2019), and adversarial perturbation (Resler & Mansour, 2019). Our analysis for static regret uses the formalism of $\epsilon-$subgradient (Jean-Baptiste, 2010, Chap. XI); for dynamic regret, we extend the work (Zhang et al., 2017) to the inexact setting allowing multiple updates per round and provide improved rates than prior results (Bedi et al., 2018; Dixit et al., 2019).

**safe RL and CMDP.** Direct policy search methods have had substantial empirical successes in solving CMDPs (Borkar, 2005; Uchibe & Doya, 2007; Bhatnagar & Lakshmanan, 2012; Achiam et al., 2017; Chow et al., 2017) (see, e.g., (Garcıa & Fernández, 2015) for a survey of safe RL).

Recently, major progress in understanding the theoretical nonasymptotic global convergence behavior of policy-based methods for CMDPs has also been achieved (Chow et al., 2018; Paternain et al., 2022; Efroni et al., 2020; Ding et al., 2021a; Ding & Lavaei, 2022; Ying et al., 2022; Yu et al., 2019; Xu et al., 2021; Chen et al., 2021b; Liu et al., 2021b;a). However, most of these works only study a single CMDP task and don't seek to make the algorithm perform well on new, potentially related CMDP tasks. In addition, while our work uses the constraint-rectified policy optimization (CRPO) algorithm proposed in (Xu et al., 2021) as a building block, our framework can be potentially adapted to most of the existing RL literature by making the dependence of guarantees on initial policy/step sizes explicit, e.g., safe exploration (Efroni et al., 2020), regularization (Geist et al., 2019), off-policy evaluation (Duan et al., 2020; Tennenholtz et al., 2020), and offline RL under constraints (Le et al., 2019; Wu et al., 2021; Lee et al., 2021; Thomas et al., 2021).

## B  CRPO Algorithm and notations

We provide some preliminaries and notations for the CRPO algorithm for the sake of completeness. CRPO (Xu et al., 2021) is a primal-based CMDP algorithm, which performs policy optimization (natural gradient ascent on the reward) when constraints are not violated, or constraint minimization (natural gradient descent on the constraint function) for the corresponding violated constraint. There are three crucial components in the overall strategy to solve the CMDP problem 1:

1. **Policy evaluation:** In each step $m$ of task $t$, for a certain policy $\pi_{t,m}$, the action-value functions $Q^i_{t,\pi_{t,m}}$ are estimated for the reward ($i = 0$) and constraints ($i = 1, ..., p$). TD-learning is employed for critic evaluation (Bhandari et al., 2018).

2. **Estimation of constraint violation:** Once the Q-estimates $\bar{Q}^i_{t,\pi_{t,m}}(s,a)$ are obtained, then a weighted average is taken to estimate expected constraint violation $\bar{J}_{t,i}(\pi_{t,m})$ under a given policy $\pi_{t,m}$.

3. **Policy optimization:** After the constraint estimation, it is checked if the expected constraint violation $\bar{J}^i_{t,\pi_{t,m}}$ exceeds the given safety threshold, i.e., if $\bar{J}_{t,i}(\pi_{t,m}) \leq d_{t,i} + \eta_t$ for all $i = 1, \ldots, p$. If none of the constraints are violated, then one step of natural policy gradient ascent is performed to maximize the objective. If one or more constraints are violated, then one step of natural policy gradient descent is conducted to minimize one of the unsatisfied constraints.

The set of time steps the policy optimization for reward maximization takes place is denoted by $\mathcal{N}_{t,0}$, and the set of time steps constraint minimization takes place is denoted by $\mathcal{N}_{t,i}$. Thus $|\mathcal{N}_{t,0}| + \sum_{i=1}^{p} |\mathcal{N}_{t,i}| = M$ for any task $t$.

## C  Inexact CMDP-within-online Algorithm

Algorithm 1 presents the inexact-CMDP-within-online algorithm for Meta-SRL. The first step in the algorithm is to initialize with some random actor policy $\phi_1$, and the learning rate $\alpha_1$ for the first task. Then, for each task $t$, a within-task algorithm (i.e., CRPO) is run for $M$ steps to obtain a policy $\hat{\pi}_t$. The discounted state visitation distribution $\hat{\nu}_t$ induced by $\hat{\pi}_t$ is then estimated using the trajectory data collected within task $t$. Afterward, an inexact OGD method is run on the new loss functions to update the meta-initialization policy $\phi_{t+1}$, and the learning rate $\alpha_{t+1}$. The online learning loop is iterated for all tasks $t \in [T]$.

## D  Proof in Section 2

We first present a simplified proof for the results in Equation 2. This result also shows in (9) how the safety threshold $\eta_t$ can be chosen to achieve sublinear convergence rate in $M$.

**Lemma 4.** *For CRPO (Xu et al., 2021) with the softmax parametrization and the exact critic estimation (i.e., no critic evaluation error), if we have*

$$\eta_t \geq \frac{2}{\alpha M} \left( \mathbb{E}_{s \sim \nu_t^*} [D(\pi_t^* | \pi_{t,0})] + \frac{2M\alpha^2 c_{max}^2 |\mathcal{S}||\mathcal{A}|}{(1-\gamma)^3} \right), \tag{9}$$

*then the following holds*

1. $\mathcal{N}_{t,0} \neq \emptyset$, *i.e.,* $\hat{\pi}_t$ *is well-defined,*

2. $J_{t,0}(\pi_t^*) - J_{t,0}(\hat{\pi}_t) \leq \eta_t.$

3. $J_{t,i}(\pi_t^*) - J_{t,i}(\hat{\pi}_t) \leq \eta_t,$ *for* $i = 1, \ldots, p.$

*Proof.* The following inequality holds due to Lemma 7 in (Xu et al., 2021):

$$\alpha \sum_{m \in \mathcal{N}_{t,0}} (J_{t,0}(\pi_t^*) - J_{t,0}(\pi_{t,m})) + \alpha \eta_t \sum_{i=1}^p |\mathcal{N}_{t,i}| \leq \mathbb{E}_{s \sim \nu_t^*}[D(\pi_t^*|\pi_{t,0})] + \frac{2M\alpha^2 |\mathcal{S}||\mathcal{A}|}{(1-\gamma)^3}. \quad (10)$$

We first verify item 1. If $\mathcal{N}_{t,0} = \emptyset$, then $\sum_{i=1}^p |\mathcal{N}_{t,i}| = M$, and 10 implies that

$$\alpha \eta_t M \leq \mathbb{E}_{s \sim \nu_t^*}[D(\pi_t^*|\pi_{t,0})] + \frac{2M\alpha^2 |\mathcal{S}||\mathcal{A}|}{(1-\gamma)^3}$$

which contradicts 9. Thus, we must have $\mathcal{N}_{t,0} \neq \emptyset$.

We then proceed to verify item 2. If $\sum_{m \in \mathcal{N}_{t,0}} (J_{t,0}(\pi_t^*) - J_{t,0}(\pi_{t,m})) > \eta_t |\mathcal{N}_{t,0}|$, then 10 implies that

$$\alpha \eta_t M \leq \mathbb{E}_{s \sim \nu_t^*}[D(\pi_t^*|\pi_{t,0})] + \frac{2M\alpha^2 |\mathcal{S}||\mathcal{A}|}{(1-\gamma)^3},$$

which contradicts 9. Hence, item 2 holds.

Finally, the item 3 holds obviously since $\hat{\pi}_t$ is sampled from $\mathcal{N}_{t,0}$. This completes the proof. $\qquad\square$

We now prove Lemma 1 in Section 2.

**Lemma 5.** *Assume* $\{\nu_t^*\}_{t=1}^T$ *and* $\{\pi_t^*\}_{t=1}^T$ *are given. For each task t, we run CRPO for M iterations with* $\alpha = \frac{(1-\gamma)^{\frac{3}{2}}}{\sqrt{2M|\mathcal{S}||\mathcal{A}|}} \sqrt{\left(\frac{L_g^2(\log T + 1)}{\mu_\pi T} + D^{*2}\right)}$. *In addition, the initialization* $\{\pi_{t,0}\}_{t=1}^T$ *are determined by playing FTRL or OGD on the functions* $\mathbb{E}_{s \sim \nu_t^*}[D_{KL}(\pi_t^*|\cdot)]$, *for* $t = 1, \ldots, T$. *Then, it holds that*

$$\bar{R}_i \leq \frac{\sqrt{8|\mathcal{S}||\mathcal{A}|}}{\sqrt{M(1-\gamma)^3}} \sqrt{\left(\frac{L_g^2(\log T + 1)}{\mu_\pi T} + D^{*2}\right)}, \ \forall i = 1, \ldots, p.$$

*Proof.* By the within-task guarantee for CMDP, we know that $\bar{R}_0$ and $\{\bar{R}_i\}_{i=1}^p$ are well-defined. In addition, it holds that

$$\bar{R}_0 \leq \frac{1}{T} \sum_{t=1}^T \left( \frac{2\mathbb{E}_{s \sim \nu_t^*}[D_{KL}(\pi_t^*|\pi_{t,0})]}{\alpha M} + \frac{4\alpha |\mathcal{S}||\mathcal{A}|}{(1-\gamma)^3} \right)$$

$$= \frac{2}{T} \sum_{t=1}^T \left( \frac{\mathbb{E}_{s \sim \nu_t^*}[D_{\mathrm{KL}}(\pi_t^*|\phi_t)] - \mathbb{E}_{s \sim \nu_t^*}[D_{KL}(\pi_t^*|\phi^*)]}{\alpha M} \right)$$

$$+ \frac{2}{T} \sum_{t=1}^T \left( \frac{\mathbb{E}_{s \sim \nu_t^*}[D_{KL}(\pi_t^*|\phi^*)]}{\alpha M} + \frac{2\alpha |\mathcal{S}||\mathcal{A}|}{(1-\gamma)^3} \right).$$

where $\phi_t = \pi_{t,0}$. The first inequality follows from the choice of $\eta_t$ from Lemma 4. The key step is the last step, which splits the total loss into the loss of the meta-update algorithm and the the loss if we had always initialized at $\phi^*$.

Since each $\mathbb{E}_{s \sim \nu_t^*}[D_{KL}(\pi_t^*|\cdot)]$ is $\mu_\pi$-strongly convex due to Assumption 1, and each $\phi_t$ is determined by playing OGD, we have that:

$$\frac{2}{T} \sum_{t=1}^T \left( \frac{\mathbb{E}_{s \sim \nu_t^*}[D_{KL}(\pi_t^*|\phi_t)] - \mathbb{E}_{s \sim \nu_t^*}[D_{KL}(\pi_t^*|\phi^*)]}{\alpha M} \right) \leq \frac{2L_g^2(\log T + 1)}{\mu_\pi \alpha M T},$$

where $L_g$ is the upper bound on $\nabla_\phi \mathbb{E}_{s \sim \nu_t^*} [D_{KL} (\pi_t^* | \phi)]$, $\mu_\pi$ is the strong convexity parameter for the KL divergence of the softmax policy.

Since $\phi^* = \arg\min_\phi \sum_{t=1}^T \mathbb{E}_{s \sim \nu_t^*} [D_{KL} (\pi_t^* | \phi)]$, by the definition of $D^*$, we have $\mathbb{E}_{s \sim \nu_t^*} [D_{KL} (\pi_t^* | \phi^*)] \leq D^{*2}$. Thus, by substituting the definition of $\phi^*$, it holds that

$$\frac{2}{T} \sum_{t=1}^T \left( \frac{\mathbb{E}_{s \sim \nu_t^*} [D_{KL} (\pi_t^* | \phi^*)]}{\alpha M} + \frac{2\alpha |\mathcal{S}||\mathcal{A}|}{(1-\gamma)^3} \right) = \frac{2D^{*2}}{\alpha M} + \frac{4\alpha |\mathcal{S}||\mathcal{A}|}{(1-\gamma)^3}.$$

Setting the value of $\alpha = \frac{\sqrt{\left( \frac{L_g^2 (\log T + 1)}{\mu_\pi T} + D^{*2} \right) (1-\gamma)^3}}{\sqrt{2M |\mathcal{S}||\mathcal{A}|}}$, we can obtain the TAOG $\bar{R}_0$ as:

$$\bar{R}_0 \leq \frac{\sqrt{8 \left( \frac{L_g^2 (\log T + 1)}{\mu_\pi T} + D^{*2} \right) |\mathcal{S}||\mathcal{A}|}}{\sqrt{M(1-\gamma)^3}}$$

The bound for $\bar{R}_i$ can be derived similarly. □

The lemma above shows how the parameters like learning rate $\alpha$ and safety threshold $\eta_t$ can be chosen to achieve decreasing TAOG and TACV in the number of updates per task $M$ and the number of tasks $T$.

# E INEXACT ONLINE GRADIENT DESCENT

## E.1 BASICS FOR $\epsilon$-SUBGRADIENT

We start with some basics for $\epsilon$-subdifferential used in the subsequent analysis. This material is based on (Jean-Baptiste, 2010, Chap. XI). Throughout this section, we consider a convex, closed, and proper function $f : \mathbb{R}^d \to \mathbb{R} \cup \{+\infty\}$ with domain $\text{Dom}(f)$. We always consider a positive $\epsilon > 0$.

**Definition 2** ($\epsilon$-subgradient (Jean-Baptiste, 2010)). *Given $\hat{x} \in \text{Dom}(f)$, the vector $u \in \mathbb{R}^d$ is called $\epsilon$-subgradient of $f$ at $\hat{x}$ when the following property holds for any $x \in \mathbb{R}^d$:*

$$f(x) \geq f(\hat{x}) + \langle u, x - \hat{x} \rangle - \epsilon.$$

*The set of all $\epsilon$-subgradients of $f$ at $\hat{x}$ is the $\epsilon$-subdifferential of $f$ at $\hat{x}$, denoted by $\partial_\epsilon f(\hat{x})$.*

In view of the exact subdifferential $\partial f(x)$, $\partial_\epsilon f(\hat{x})$ can be called an approximate subdifferential, which is a set-valued function with a convex graph. For practical use, $\partial_\epsilon f(\hat{x})$ can be used to characterize the $\epsilon$-solution to a convex minimization problem.

**Lemma 6.** *((Jean-Baptiste, 2010, Thm. 1.1.5)) The following two properties are equivalent.*

$$0 \in \partial_\epsilon f(\hat{x}) \iff f(\hat{x}) \leq f(x) + \epsilon, \qquad \text{for all } x \in \mathbb{R}^d.$$

One useful result that stems directly from the definition is to link the $\epsilon$-subdifferential of two uniformly close functions (e.g., an expectation of a function and its empirical version).

**Lemma 7.** *Consider two convex functions $f$ and $g$, with the property that $\|f - g\|_\infty \leq \epsilon$, where $\|f - g\|_\infty = \max_x |f(x) - g(x)|$. Then, for any $x \in \mathbb{R}^d$ and $u \in \partial f(x)$ in the subdifferential of $f$ at $x$, it is also in the $2\epsilon$-subdifferential of $g$ at $x$, i.e., $u \in \partial_{2\epsilon} g(x)$.*

*Proof.* The proof follows directly by convexity and the uniform condition:

$$\begin{aligned} g(y) &\geq f(y) - \epsilon \\ &\geq f(x) + \langle s, y - x \rangle - \epsilon \\ &\geq g(x) + \langle s, y - x \rangle - 2\epsilon, \end{aligned}$$

where the second inequality is by the convexity of $f$, and the first and last inequalities are due to the supremum norm condition. □

---

**Algorithm 2:** Inexact OGD Algorithm

---

**Input:** Learning rate $\alpha$, $x_1 = 0$

1: **for** $t = 1, .., T$ **do**
2:     Incur loss $\ell_t(x_t)$ and compute $\epsilon$-gradient $\hat{\nabla}_t \ell_t(x_t)$
3:     $x_{t+1} = P_X(x_t - \alpha \hat{\nabla}_t \ell_t(x_t))$
4: **end for**

---

Our next result is concerned with bounding the distance (measured in $\ell_2$ norm) between the true gradient and the $\epsilon$-subgradient of the function, assuming the function is differentiable and smooth.

**Lemma 8.** *Suppose a function $f$ is convex, differentiable, and $L$-smooth over $\mathrm{Dom}(f)$, and $u \in \partial_\epsilon f(x)$ is an $\epsilon$-subgradient of $f$ at $x \in \mathrm{Dom}(f)$. Then,*

$$\|u - \nabla f(x)\|_2^2 \leq \frac{2\epsilon}{2C_1 - C_1^2 L},$$

*for any $C_1 \in \{c \in (0, \frac{2}{L}) : x + c(u - \nabla f(x)) \in \mathrm{Dom}(f)\}$. In particular, if $x + \frac{1}{L}(u - \nabla f(x)) \in \mathrm{Dom}(f)$, then $\|u - \nabla f(x)\|_2^2 \leq 2\epsilon L$.*

*Proof.* Since $u$ is an $\epsilon$-gradient, $f(y) \geq f(x) + \langle u, y - x \rangle - \epsilon$ for all $y \in \mathrm{Dom}(f)$. Thus,

$$0 \leq f(x) - f(y) + \langle \nabla f(x), y - x \rangle + \frac{L}{2}\|y - x\|^2$$

$$\leq \langle \nabla f(x) - u, y - x \rangle + \frac{1}{2}\|y - x\|^2 + \epsilon$$

Choose $y = x + c(u - \nabla f(x))$ for $c \in (0, \frac{2}{L})$ such that $x + c(u - \nabla f(x)) \in \mathrm{Dom}(f)$, we have that

$$\|u - \nabla f(x)\|_2^2 \leq \frac{2\epsilon}{2c - c^2 L}.$$

$\square$

The smoothness condition in the above seems necessary, as we can construct counterexamples that drive the distance of an $\epsilon$-subgradient and its exact counterpart arbitrarily large without the smoothness condition. In fact, it is known that the set-valued mapping $(x, \epsilon) \to \partial_\epsilon f(x)$ is inner semi-continuous for a Lipschitz-continuous $f$, which is implied by the fact that the distance (using the Hausdorff distance for sets) between any two subdifferential $\partial_\epsilon f(x)$ and $\partial_{\epsilon'} f(x')$ for all $x, x' \in \mathbb{R}^d$ and $\epsilon, \epsilon'$ is positive, and shown to be bounded by $\mathcal{O}\left(\frac{1}{\min\{\epsilon, \epsilon'\}}(\|x - x'\| + |\epsilon - \epsilon'|)\right)$ (Jean-Baptiste, 2010, Thm. 4.1.3). While the exact gradient can be interpreted as $\epsilon$-subgradient driving $\epsilon \to 0^+$, the existing bound provided by (Jean-Baptiste, 2010, Thm. 4.1.3) is vacuous in this case; on the other hand, the bound provided in Lemma 8 remains meaningful.

### E.2 STATIC REGRET FOR THE INEXACT OGD ALGORITHM

In the following, we consider the online learning setup, where a sequence of loss functions $\{\ell_t\}_{t \in [T]}$ are revealed sequentially, and the performance of the OGD algorithm (see Algorithm 2) is measured against a static decision in hindsight:

$$\text{(static regret)} \quad \sum_{t=1}^{T} \ell_t(x_t) - \min_{x \in X} \sum_{t=1}^{T} \ell_t(x) \tag{11}$$

where $\{x_t \in X\}_{t \in [T]}$ is a sequence of actions played by the online algorithm. For simplicity, we assume that $X$ belongs to the domains of $\ell_t$ for all $t \in [T]$. Furthermore, we define the following cumulative inexact error bounds:

$$\mathcal{E}_T := \sum_{t=1}^{T} \epsilon_t, \tag{12}$$

where $\epsilon_t$ corresponds to the inexactness of the $\epsilon_t$-subgradient in each round of OGD.

**Theorem E.1** (Static regret bound for the inexact OGD). *Assume that $\{\ell_t\}_{t\in[T]}$ are convex and $L_2$-smooth, with bounded gradient, i.e., $\|\nabla\ell_t(x)\|_2 \le L_1$ for all $t \in [T]$ and all $x \in X$. Then, for any comparator $x \in X$, with the stepsize $\alpha := \frac{\|x\|}{L_1\sqrt{2T}}$, we have that*

$$\sum_{t=1}^{T} \ell_t(x_t) - \sum_{t=1}^{T} \ell_t(x) \le L_1\|x\|\sqrt{2T} + \left(1 + \frac{\sqrt{2}cL_1L_2\|x\|}{\sqrt{T}}\right)\sum_{t=1}^{T}\epsilon_t,$$

*where $\epsilon_t$ is the amount of inexactness at each step $t$.*

*Proof.* By convexity and the fact that $\hat{\nabla}_t$ is an $\epsilon_t$-subgradient of $\ell_t$ at $x_t$, we have that

$$\ell_t(x_t) - \ell_t(x) \le \langle\hat{\nabla}_t, x_t - x\rangle + \epsilon_t, \quad \forall x \in X$$

Hence, summing over $t = 1, ..., T$, we get

$$\frac{1}{T}\sum_{t=1}^{T}\ell_t(x_t) - \ell_t(x) \le \frac{1}{T}\sum_{t=1}^{T}\langle\hat{\nabla}_t, x_t - x\rangle + \epsilon_t.$$

To bound the RHS, observe that

$$\|x_{t+1} - x\|^2 \le \|x_t - \alpha\hat{\nabla}_t - x\|^2$$
$$= \|x_t - x\|^2 - 2\alpha\langle x_t - x, \hat{\nabla}_t\rangle + \alpha^2\|\hat{\nabla}_t\|^2,$$

where the first inequality is due to the OGD update rule and the nonexpansiveness of the projection operator. Thus, rearranging the terms and exploiting the telescopic sum over $t \in [T]$, we have that

$$\sum_{t=1}^{T}\langle x_t - x, \hat{\nabla}_t\rangle \le \frac{1}{2\alpha}(\|x_1 - x\|^2 - \|x_{T+1} - x\|^2) + \frac{\alpha}{2}\sum_{t=1}^{T}\|\hat{\nabla}_t\|^2$$

$$\le \frac{1}{2\alpha}\|x_1 - x\|^2 + \frac{\alpha}{2}\sum_{t=1}^{T}\|\hat{\nabla}_t\|^2.$$

Furthermore, since $\ell_t$ is $L_2$-smooth with bounded gradient, and $\hat{\nabla}_t$ is an $\epsilon_t$-gradient for any $t \in [T]$, by Lemma 8, the following holds:

$$\|\hat{\nabla}_t\|^2 \le 2\|\nabla_t\|^2 + 2\|\nabla_t - \hat{\nabla}_t\|^2$$
$$\le 2L_1^2 + 2cL_2\epsilon_t,$$

where the constant $c$ is specified by Lemma 8. Hence, by combining the above relations, we get

$$\frac{1}{T}\sum_{t=1}^{T}\ell_t(x_t) - \ell_t(x) \le \frac{1}{2\alpha T}\|x_1 - x\|^2 + \frac{1}{T}\sum_{t=1}^{T}\left(\frac{\alpha}{2}\|\hat{\nabla}_t\|^2 + \epsilon_t\right)$$

$$\le \frac{1}{2\alpha T}\|x_1 - x\|^2 + \alpha L_1^2 + \left(\frac{\alpha cL_2 + 1}{T}\right)\sum_{t=1}^{T}\epsilon_t.$$

Let $\alpha = \frac{\|x\|}{L_1\sqrt{2T}}$, then we get the RHS as

$$L_1\|x\|\sqrt{\frac{2}{T}} + \frac{1 + \frac{\sqrt{2}cL_1L_2\|x\|}{\sqrt{T}}}{T}\sum_{t=1}^{T}\epsilon_t.$$

$\square$

**Remark 3.** *We can relax the dependence of setting the stepsize on $T$ by using a standard doubling trick (first proposed in (Auer et al., 2002), see also, e.g., (Balcan et al., 2019; Khodak et al., 2019)).*

After establishing the static regret for the inexact OGD, we can use this result to obtain the proof of Lemma 2, which gives the static regret if we run inexact OGD over the loss sequences $\mathbb{E}_{\nu_t^*}[D_{KL}(\pi_t^*|\pi_{t,0})]$ for all $t \in [T]$. Overall, the following regret bound will eventually help us establish the proof of Theorem 3.2, where we will utilize the static regret upper bound for inexact OGD over the loss sequences $\mathbb{E}_{\nu_t^*}[D_{KL}(\pi_t^*|\pi_{t,0})]$ for all $t \in [T]$. Also, we denote the norm of the policy with respect to the state distribution $\nu$ as $\|\pi\|_\nu = \sum_{s \in \mathcal{S}} \nu(s)\pi(s)$. Now we proceed to present the proof for Lemma 2.

**Lemma 9.** *Denote $\ell_t(\pi_{t,0}) := \mathbb{E}_{\nu_t^*}[D_{KL}(\pi_t^*|\pi_{t,0})]$ for all $t \in [T]$. For any fixed comparator $\pi_0^* = \underset{\pi_0 \in \Delta \mathcal{A}_\varrho^{|\mathcal{S}|}}{\arg\min} \sum_{t=1}^T \ell_t(\pi_0)$, if OGD is run on a sequence of loss functions $\{\hat{\ell}_t\}_{t \in [T]}$, where $\hat{\ell}_t := \mathbb{E}_{\hat{\nu}_t}[D_{KL}(\hat{\pi}_t|\pi_{t,0})]$ with the step-size $\alpha := \frac{\|\pi_0^*\|}{L_g\sqrt{2T}}$, then the following bound holds for static regret:*

$$\sum_{t=1}^T \ell_t(\pi_{t,0}) - \sum_{t=1}^T \ell_t(\pi_0^*) \le \sqrt{2T}L_g\|\pi_0^*\| + \left(1 + \frac{4\sqrt{2}L_g L_\pi \|\pi_0^*\|}{(2C_1 - C_1^2 L_\pi)\sqrt{T}}\right)\mathcal{E}_T,$$

*for any $C_1 \in \{c \in \left(0, \frac{2}{L_\pi}\right) : \pi_0^* + c(\hat{\nabla}_t - \nabla_t) \in \Delta \mathcal{A}_\varrho^{|\mathcal{S}|}\}$ where $\hat{\nabla}_t$ and $\nabla_t$ are an $\epsilon_t$-subgradient and exact subgradient of $\mathbb{E}_{\nu_t^*}[D_{KL}(\pi_t^*|\pi_{t,0})]$ at $\pi_{t,0}$, respectively, $\mathcal{E}_T := \sum_{t=1}^T \epsilon_t$ is the cumulative inexactness.*

*Proof.* The proof follows directly after substituting $c = \frac{4}{2C_1 - C_1^2 L_\pi}$ and other appropriate constants in Theorem E.1. $\qquad\square$

Note that the inexactness bound $\epsilon_t$ can be obtained from Theorem 3.1. Establishing the above Corollary, we can finally provide the proof for Theorem 3.2.

**Proof of Theorem 3.2**

**Theorem E.2.** *Let $\hat{D}^{*2} = \underset{\phi \in \Delta \mathcal{A}_\varrho^{|\mathcal{S}|}}{\min} \frac{1}{T} \sum_{t=1}^T \mathbb{E}_{s \sim \hat{\nu}_t}[D_{KL}(\hat{\pi}_t|\phi)]$ be the empirical task-similarity, and let $c_1 = \sqrt{2}L_g\|\phi^*\|$, and $c_2 = \left(2 + \frac{4\sqrt{2}L_g L_\pi \|\phi^*\|}{(2C_1 - C_1^2 L_\pi)\sqrt{T}}\right)$, where $\phi^*$ is the fixed optimal meta-initialization for all the tasks given by $\phi^* = \underset{\phi \in \Delta \mathcal{A}_\varrho^{|\mathcal{S}|}}{\arg\min} \frac{1}{T} \sum_{t=1}^T \mathbb{E}_{s \sim \hat{\nu}_t}[D_{KL}(\hat{\pi}_t|\phi)]$. For each task $t$, we run CRPO for $M$ iterations with $\alpha = \sqrt{\frac{|\mathcal{S}||\mathcal{A}|}{2M(1-\gamma)^3}}\sqrt{\left(\frac{c_1}{\sqrt{T}} + \frac{c_2\mathcal{E}_T}{T} + \hat{D}^{*2}\right)}$, and we obtain $\{\hat{\nu}_t\}_{t=1}^T$ and $\{\hat{\pi}_t\}_{t=1}^T$. In addition, the initialization $\{\pi_{t,0}\}_{t=1}^T$ are determined by playing OGD on the functions $\mathbb{E}_{s \sim \hat{\nu}_t}[D_{KL}(\hat{\pi}_t|\cdot)]$, for $t = 1, \dots, T$. Then, it holds that*

$$\bar{R}_i \le \frac{\sqrt{8|\mathcal{S}||\mathcal{A}|}}{\sqrt{M}(1-\gamma)^{3/2}}\left(\sqrt{\frac{\sqrt{2}L_g\|\phi^*\|}{\sqrt{T}} + \left(2 + \frac{4\sqrt{2}L_g L_\pi \|\phi^*\|}{(2C_1 - C_1^2 L_\pi)\sqrt{T}}\right)\frac{\mathcal{E}_T}{T} + \hat{D}^{*2}}\right) \quad \forall i = 0, \dots, p.$$

*Proof.* We know that $\bar{R}_0$ and $\{\bar{R}_i\}_{i=1}^p$ are well-defined. In addition, it holds that

$$
\begin{aligned}
\bar{R}_0 \leq & \frac{2}{T} \sum_{t=1}^T \left( \frac{\mathbb{E}_{s\sim\nu_t^*}\left[D_{KL}\left(\pi_t^*|\phi_t\right)\right]}{\alpha M} + \frac{2\alpha|\mathcal{S}||\mathcal{A}|}{(1-\gamma)^3} \right) \\
= & \frac{2}{T} \sum_{t=1}^T \left( \frac{\mathbb{E}_{s\sim\nu_t^*}\left[D_{\mathrm{KL}}\left(\pi_t^*|\phi_t\right)\right] - \mathbb{E}_{s\sim\nu_t^*}\left[D_{KL}\left(\pi_t^*|\phi^*\right)\right]}{\alpha M} \right) \\
& + \frac{2}{T} \sum_{t=1}^T \left( \frac{\mathbb{E}_{s\sim\nu_t^*}\left[D_{KL}\left(\pi_t^*|\phi^*\right)\right]}{\alpha M} + \frac{2\alpha|\mathcal{S}||\mathcal{A}|}{(1-\gamma)^3} \right) \\
\leq & \frac{2}{T} \sum_{t=1}^T \left( \frac{\mathbb{E}_{s\sim\nu_t^*}\left[D_{\mathrm{KL}}\left(\pi_t^*|\phi_t\right)\right] - \mathbb{E}_{s\sim\nu_t^*}\left[D_{KL}\left(\pi_t^*|\phi^*\right)\right]}{\alpha M} \right) \\
& + \frac{2}{T} \sum_{t=1}^T \left( \frac{\mathbb{E}_{s\sim\hat{\nu}_t}\left[D_{KL}\left(\hat{\pi}_t|\phi^*\right)\right] \pm \epsilon_t}{\alpha M} + \frac{2\alpha|\mathcal{S}||\mathcal{A}|}{(1-\gamma)^3} \right).
\end{aligned}
$$

where $\phi_t = \pi_{t,0}$. Second equality follows from the fact that the total loss can be split into the loss of the meta-update algorithm and the the loss if we had always initialized at $\phi^*$. Last inequality follows from the KL-divergence estimation error bound in Theorem 3.1.

Since each $\mathbb{E}_{s\sim\hat{\nu}_t}\left[D_{KL}\left(\hat{\pi}_t|\cdot\right)\right]$ is $\mu_\pi$-strongly convex due to Assumption 1, and since each $\phi_t$ is determined by playing FTL or inexact OGD, the following term can be upper bounded using Lemma 9 as follows:

$$
\begin{aligned}
\frac{2}{T} \sum_{t=1}^T & \left( \frac{\mathbb{E}_{s\sim\nu_t^*}\left[D_{KL}\left(\pi_t^*|\phi_t\right)\right] - \mathbb{E}_{s\sim\nu_t^*}\left[D_{KL}\left(\pi_t^*|\phi^*\right)\right]}{\alpha M} \right) \leq \\
& \frac{2}{\alpha M} \left( \frac{\sqrt{2}L_g\|\phi^*\|}{\sqrt{T}} + \left(1 + \frac{4\sqrt{2}L_g L_\pi\|\phi^*\|}{(2C_1 - C_1^2 L_\pi)\sqrt{T}}\right) \frac{\mathcal{E}_T}{T} \right),
\end{aligned}
$$

where the constants are from the Corollary 9. Now, we will upper bound the second term. Since $\phi^* = \arg\min_\phi \frac{1}{T} \sum_{t=1}^T \mathbb{E}_{s\sim\hat{\nu}_t}\left[D_{KL}\left(\hat{\pi}_t|\phi\right)\right]$, by the definition of $\hat{D}^*$, we have $\mathbb{E}_{s\sim\hat{\nu}_t}\left[D_{KL}\left(\hat{\pi}_t|\phi\right)\right] \leq \hat{D}^{*2}$. Thus, by substituting the definition of $\phi^*$, it holds that

$$
\frac{2}{T} \sum_{t=1}^T \left( \frac{\mathbb{E}_{s\sim\hat{\nu}_t}\left[D_{KL}\left(\hat{\pi}_t|\phi^*\right)\right] \pm \epsilon_t}{\alpha M} + \frac{2\alpha|\mathcal{S}||\mathcal{A}|}{(1-\gamma)^3} \right) \leq \frac{2\hat{D}^{*2}}{\alpha M} + \frac{2\mathcal{E}_T}{T\alpha M} + \frac{4\alpha|\mathcal{S}||\mathcal{A}|}{(1-\gamma)^3}.
$$

Setting the value of $\alpha = \frac{(1-\gamma)^{3/2}\sqrt{\frac{c_1}{\sqrt{T}} + \frac{c_2 \mathcal{E}_T}{T} + \hat{D}^{*2}}}{\sqrt{2M|\mathcal{S}||\mathcal{A}|}}$, where $c_1 = \sqrt{2}L_g\|\phi^*\|$, $c_2 = \left(2 + \frac{4\sqrt{2}L_g L_\pi\|\phi^*\|}{(2C_1 - C_1^2 L_\pi)\sqrt{T}}\right)$, we can obtain the TAOG $\bar{R}_0$ as follows:

$$
\bar{R}_0 \leq \frac{\sqrt{8\left(\frac{c_1}{\sqrt{T}} + \frac{c_2 \mathcal{E}_T}{T} + \hat{D}^{*2}\right)|\mathcal{S}||\mathcal{A}|}}{\sqrt{M(1-\gamma)^3}}
$$

The bound for $\bar{R}_i$ can be derived similarly.

$\square$

### E.3 DYNAMIC REGRET FOR THE INEXACT OGD ALGORITHM

In the following, we consider a stronger notion of regret that measures the performance of the OGD algorithm (see Algorithm 2) against a dynamically changing sequence in hindsight (see, e.g., (Zinkevich, 2003; Jadbabaie et al., 2015; Zhang et al., 2017)):

$$
\text{(dynamic regret)} \quad \sum_{t=1}^T \ell_t(x_t) - \sum_{t=1}^T \ell_t(x_t^*) \tag{13}
$$

where $x_t^* \in \arg\min_{x \in X} \ell_t(x)$ is the optimal decision for the loss $\ell_t$. It is well known that in the worst case, it is impossible to achieve a sub-linear dynamic regret bound, due to the arbitrary fluctuation in the functions (Zinkevich, 2003; Besbes et al., 2015; Yang et al., 2016). Thus, it is common to upper bound the dynamic regret in terms of a certain regularity of the comparator sequence. One possible regularity condition is the path length of the comparator sequence (Zinkevich, 2003; Jadbabaie et al., 2015):

$$\mathcal{P}_T := \sum_{t=2}^{T} \|x_t^* - x_{t-1}^*\|, \tag{14}$$

which captures the cumulative Euclidean norm of the difference between successive comparators (note that we will use $\|\cdot\|$ for the Euclidean norm, unless otherwise specified). The path-length measure is also the regularity condition used in existing inexact OGD literature (Bedi et al., 2018; Dixit et al., 2019). However, as remarked in (Zhang et al., 2017), a potentially tighter bound can be achieved by examining the squared path-length measure:

$$\mathcal{S}_T := \sum_{t=2}^{T} \|x_t^* - x_{t-1}^*\|^2, \tag{15}$$

which can be much smaller than $\mathcal{P}_T$ when the local variations are small. For example, when $\|x_t^* - x_{t-1}^*\| = \Theta(1/\sqrt{T})$ for all $t \in [T]$, we have $\mathcal{P}_T = \Theta(\sqrt{T})$ but $\mathcal{S}_T = \Theta(1)$. In this section, we provide analysis with respect to both measures for strongly convex and smooth functions. Furthermore, we propose to apply inexact OGD multiple times in each round, and demonstrate that the dynamic regret is reduced from $\mathcal{O}(\mathcal{P}_T + \mathcal{E}_T)$ to $\mathcal{O}(\min\{\mathcal{P}_T + \mathcal{E}_T, \mathcal{S}_T + \tilde{\mathcal{E}}_T\})$, where

$$\tilde{\mathcal{E}}_T := \sum_{t=1}^{T} \sqrt{\epsilon_t}$$

is the cumulative square root inexactness bounds. Note that our results improve over existing bounds for inexact online learning (Bedi et al., 2018; Dixit et al., 2019) and can be regarded as a generalization of (Zhang et al., 2017) to the inexact settings. We start with a result that will be used in later analysis.

**Lemma 10.** *Assume that $f : X \to \mathbb{R}$ is $\lambda$-strongly convex and $L$-smooth, and let $x^* = \arg\min_{x \in X} f(x)$ be the unique optimal solution. Let $v = P_X(x - \alpha\hat{\nabla}f(x))$, where $\hat{\nabla}f(u) \in \partial_\epsilon f(u)$ and $\alpha \leq \frac{1}{2L}$, we have that*

$$\|v - x^*\|^2 \leq \frac{1}{\lambda\alpha + 1}\|x^* - x\|^2 + \frac{c\alpha + 2L\alpha}{\lambda L\alpha + L}\epsilon,$$

*where the constant $c$ is specified in Lemma 8.*

*Proof.* By the update rule, we have that

$$v = \arg\min_{x' \in X} f(x) + \langle\hat{\nabla}f(x), x' - x\rangle + \frac{1}{2\alpha}\|x' - x\|^2. \tag{16}$$

By strong convexity of the objective above,

$$\langle\hat{\nabla}f(x), v - x\rangle + \frac{1}{2\alpha}\|v - x\|^2 \leq \langle\hat{\nabla}f(x), x^* - x\rangle + \frac{1}{2\alpha}\|x^* - x\|^2 - \frac{1}{2\alpha}\|v - x^*\|^2. \tag{17}$$

Since, $f(x)$ is $\lambda$-strongly convex and $L$-smooth, we have that

$$f(x^*) - \frac{\lambda}{2}\|x^* - x\|^2 \geq f(x) + \langle\nabla f(x), x^* - x\rangle, \tag{18}$$

and

$$f(x^*) \leq f(x) + \langle\nabla f(x), x^* - x\rangle + \frac{L}{2}\|x^* - x\|^2. \tag{19}$$

Also, since $\hat{\nabla}f(x)$ is an $\epsilon$-subgradient, we can write

$$f(x^*) \geq f(x) + \langle\hat{\nabla}f(x), x^* - x\rangle - \epsilon. \tag{20}$$

---

**Algorithm 3:** Inexact Online Multiple Gradient Descent Algorithm

---

**Input:** Learning rate $\alpha$, $x_1 = 0$

1: **for** $t = 1, .., T$ **do**
2:     Incur loss $\ell_t(x_t)$
3:     $z_t^1 = x_t$
4:     **for** $k = 1, ..., K$ **do**
5:         $z_t^{k+1} = P_X(x_t - \alpha \hat{\nabla}\ell_t(z_t^k))$
6:     **end for**
7:     $x_{t+1} = z_t^{K+1}$
8: **end for**

---

Combining 18, 19 and 20, we have that

$$f(x^*) + \frac{L - \lambda}{2}\|x^* - x\|^2 \geq f(x) + \langle \hat{\nabla}f(x), x^* - x \rangle - \epsilon.$$

Combining the above relations, we have that

$$
\begin{aligned}
f(v) &\leq f(x) + \langle \nabla f(x), v - x \rangle + \frac{L}{2}\|v - x\|^2 \\
&= f(x) + \langle \hat{\nabla}f(x), v - x \rangle + \frac{L}{2}\|v - x\|^2 + \langle \nabla f(x) - \hat{\nabla}f(x), v - x \rangle \\
&\overset{(i)}{\leq} f(x) + \langle \hat{\nabla}f(x), x^* - x \rangle + \left(\frac{L}{2} - \frac{1}{2\alpha}\right)\|v - x\|^2 \\
&\qquad\qquad + \frac{1}{2\alpha}\|x^* - x\|^2 - \frac{1}{2\alpha}\|v - x^*\|^2 + \langle \nabla f(x) - \hat{\nabla}f(x), v - x \rangle \\
&\overset{(ii)}{\leq} f(x^*) + \left(\frac{L}{2} - \frac{1}{2\alpha}\right)\|v - x\|^2 + \frac{1}{2\alpha}\|x^* - x\|^2 \\
&\qquad\qquad - \frac{1}{2\alpha}\|v - x^*\|^2 + \langle \nabla f(x) - \hat{\nabla}f(x), v - x \rangle + \epsilon \\
&\overset{(iii)}{\leq} f(v) - \left(\frac{\lambda}{2} + \frac{1}{2\alpha}\right)\|v - x^*\|^2 + \left(\frac{L}{2} - \frac{1}{2\alpha}\right)\|v - x\|^2 \\
&\qquad\qquad + \frac{1}{2\alpha}\|x^* - x\|^2 + \langle \nabla f(x) - \hat{\nabla}f(x), v - x \rangle + \epsilon \\
&\overset{(iv)}{\leq} f(v) - \left(\frac{\lambda}{2} + \frac{1}{2\alpha}\right)\|v - x^*\|^2 + \left(\frac{L}{2} - \frac{1}{2\alpha}\right)\|v - x\|^2 \\
&\qquad\qquad + \frac{1}{2\alpha}\|x^* - x\|^2 + \|\nabla f(x) - \hat{\nabla}f(x)\|\|v - x\| + \epsilon \\
&\overset{(v)}{\leq} f(v) - \left(\frac{\lambda}{2} + \frac{1}{2\alpha}\right)\|v - x^*\|^2 \\
&\qquad\qquad + \left(\frac{L}{2} - \frac{1}{2\alpha} + \frac{\kappa}{2}\right)\|v - x\|^2 + \frac{1}{2\alpha}\|x^* - x\|^2 + \left(\frac{c}{2\kappa} + 1\right)\epsilon,
\end{aligned}
$$

where the first inequality is due to $L$-smoothness, $(i)$ follows from 17, $(ii)$ is due to convexity, $(iii)$ is due to strong convexity, $(iv)$ follows from Cauchy-Schwarz inequality, and $(v)$ is due to the inequality $ab \leq \frac{1}{2\kappa}a^2 + \frac{\kappa}{2}b^2$ for $a, b \geq 0$ and $\kappa > 0$ and the constant $c$ comes from Lemma 8. Choosing $\kappa = L$, $\alpha \leq \frac{1}{2L}$, and rearranging the above, we have then proved the claim. $\qquad\square$

**Theorem E.3** (Dynamic regret for inexact OGD with multiple updates). *Assume that $\ell_t : X \to \mathbb{R}$ is $\lambda$-strongly convex, $L_1$-Lipschitz, and $L_2$-smooth for all $t \in [T]$. By setting $\alpha \leq \frac{1}{2L_2}$, $K := \lceil \frac{\ln 2}{\ln(1+\lambda\alpha)} \rceil$, then, for any $\beta > 0$, we have that*

$$\sum_{t=1}^{T} \ell_t(x_t) - \ell_t(x_t^*) \leq \min \left( C_1 \|x_1 - x_1^*\|^2 + C_2 \mathcal{E}_T + C_3 S_T + \frac{1}{2\beta} \sum_{t=1}^{T} \|\nabla \ell_t(x_t^*)\|^2, \right.$$

$$\left. C_4 \|x_1 - x_1^*\| + C_5 \sum_{t=1}^{T} \sqrt{\epsilon_t} + C_4 P_T \right),$$

where $C_1 = 2(L_2 + \beta)$, $C_2 = (L_2 + \beta)\frac{3c\alpha + 6\alpha L_2}{2\lambda\alpha L_2}$, $C_3 = 3(L_2 + \beta)$, $C_4 = \frac{2L_1}{2 - \sqrt{2}}$ and $C_5 = \frac{2L_1}{2 - \sqrt{2}} \sqrt{\frac{c\alpha + 2L_2\alpha}{2\alpha\lambda L_2}}$.

*Proof.* The proof has two parts, where we use different techniques to bound the dynamic regret by $S_T$ and $\mathcal{E}_T$, as well as $\mathcal{P}_T$ and $\tilde{\mathcal{E}}_T$. Then the final bound is obtained by taking the minimum between the two bounds.

**Bounding the dynamic regret by $S_T$ and $\mathcal{E}_T$.** Since $\ell_t$ is $L_2$-smooth, we have that

$$\ell_t(x_t) - \ell_t(x_t^*) \leq \langle \nabla \ell_t(x_t^*), x_t - x_t^* \rangle + \frac{L_2}{2} \|x_t - x_t^*\|^2 \tag{21}$$

$$\leq \|\nabla \ell_t(x_t^*)\| \|x_t - x_t^*\| + \frac{L_2}{2} \|x_t - x_t^*\|^2 \tag{22}$$

$$\leq \frac{1}{2\beta} \|\nabla \ell_t(x_t^*)\|^2 + \frac{L_2 + \beta}{2} \|x_t - x_t^*\|^2, \tag{23}$$

where the second inequality is due to Cauchy–Schwartz and the third inequality is due to $ab \leq \frac{1}{2\beta} a^2 + \frac{\beta}{2} b^2$ for $a, b \geq 0$ and $\beta > 0$.

Now, using $\|x - y\|^2 \leq (1 + \iota)\|x - z\|^2 + \left(1 + \frac{1}{\iota}\right) \|z - y\|^2$, we can bound

$$\sum_{t=1}^{T} \|x_t - x_t^*\|^2 \leq \|x_1 - x_1^*\|^2 + \sum_{t=2}^{T} (1 + \iota)\|x_t - x_{t-1}^*\|^2 + \left(1 + \frac{1}{\iota}\right) \|x_t^* - x_{t-1}^*\|^2. \tag{24}$$

Recall the updating rule $z_{t-1}^{j+1} = P_X(z_{t-1}^j - \alpha \hat{\nabla} f_{t-1}(z_{t-1}^j))$, $j = 1, ..., K$; then, we can write that

$$\|x_t - x_{t-1}^*\|^2 = \|z_{t-1}^{K+1} - x_{t-1}^*\|^2 \tag{25}$$

$$\leq \left(\frac{1}{\lambda\alpha + 1}\right)^K \|x_{t-1} - x_{t-1}^*\|^2 + \frac{1 - \left(\frac{1}{\lambda\alpha + 1}\right)^K}{1 - \frac{1}{\lambda\alpha + 1}} \frac{c\alpha + 2L_2\alpha}{\lambda L_2\alpha + L_2} \epsilon_{t-1},$$

where we recursively apply the result from Lemma 10. Thus, by plugging in 25 into 24, and using the definitions of $S_T$ and $\mathcal{S}_T$, we have that

$$\sum_{t=1}^{T} \|x_t - x_t^*\|^2 \leq \|x_1 - x_1^*\|^2 + (1 + \iota)\left(\frac{1}{\lambda\alpha + 1}\right)^K \sum_{t=1}^{T} \|x_t - x_t^*\|^2 \tag{26}$$

$$+ (1 + \iota)\frac{1 - \left(\frac{1}{\lambda\alpha + 1}\right)^K}{1 - \frac{1}{\lambda\alpha + 1}} \frac{c\alpha + 2L_2\alpha}{\lambda L_2\alpha + L_2} \mathcal{E}_T + \left(1 + \frac{1}{\iota}\right) S_T.$$

Rearranging the terms, the above relation implies that

$$\sum_{t=1}^{T} \|x_t - x_t^*\|^2 \leq \frac{(1 + \lambda\alpha)^K}{(1 + \lambda\alpha)^K - (1 + \iota)} \|x_1 - x_1^*\|^2 + \left(1 + \frac{1}{\iota}\right) \frac{(1 + \lambda\alpha)^K}{(1 + \lambda\alpha)^K - (1 + \iota)} S_T$$

$$+ (1 + \iota)\frac{(1 + \lambda\alpha)^K - 1}{(1 + \lambda\alpha)^K - (1 + \iota)} \frac{c\alpha + 2L_2\alpha}{\lambda L_2\alpha} \mathcal{E}_T$$

Let $\iota = \frac{1}{2}$ and choose $K = \lceil \frac{\log 2}{\log(1 + \lambda\alpha)} \rceil$, we have

$$\sum_{t=1}^{T} \|x_t - x_t^*\|^2 \leq 4\|x_1 - x_1^*\|^2 + \frac{3c\alpha + 6L_2\alpha}{\lambda\alpha L_2} \mathcal{E}_T + 6S_T.$$

Combine the above with 23, and summing over $t \in [T]$, we have that

$$\sum_{t=1}^{T} \ell_t(x_t) - \ell_t(x_t^*)$$

$$\leq \frac{1}{2\beta} \sum_{t=1}^{T} \|\nabla \ell_t(x_t^*)\|^2 + 3(L_2 + \beta)\mathcal{S}_T + (L_2 + \beta)\frac{3c\alpha + 6L\alpha}{2\lambda\alpha L}\mathcal{E}_T + 2(L_2 + \beta)\|x_1 - x_1^*\|^2,$$

which holds true for any positive $\beta > 0$.

**Bounding the dynamic regret by $\mathcal{P}_T$ and $\tilde{\mathcal{E}}_T$.** By 26 and the choice of $K = \lceil \frac{\log 2}{\log(1+\lambda\alpha)} \rceil$, we have that:

$$\|x_t - x_{t-1}^*\|^2 \leq \frac{1}{2}\|x_{t-1} - x_{t-1}^*\|^2 + \frac{c\alpha + 2L_2\alpha}{2\alpha\lambda L_2}\epsilon_{t-1}.$$

Thus,

$$\|x_t - x_{t-1}^*\| \leq \sqrt{\frac{1}{2}\|x_{t-1} - x_{t-1}^*\|^2 + \frac{c\alpha + 2L_2\alpha}{2\alpha\lambda L_2}\epsilon_{t-1}}$$

$$\leq \frac{1}{\sqrt{2}}\|x_{t-1} - x_{t-1}^*\| + \sqrt{\frac{c\alpha + 2L_2\alpha}{2\alpha\lambda L_2}}\sqrt{\epsilon_{t-1}}, \tag{27}$$

where the last inequality follows from $\sqrt{a+b} \leq \sqrt{a} + \sqrt{b}$. Due to the bounded gradient assumption, we have that

$$\sum_{t=1}^{T} \ell_t(x_t) - \ell_t(x_t^*) \leq L_1 \sum_{t=1}^{T} \|x_t - x_t^*\| \tag{28}$$

To bound $\sum_{t=1}^{T} \|x_t - x_t^*\|$, notice that

$$\sum_{t=1}^{T} \|x_t - x_t^*\| \leq \|x_1 - x_1^*\| + \sum_{t=2}^{T} \|x_t - x_{t-1}^*\| + \|x_{t-1}^* - x_t^*\|$$

$$\leq \|x_1 - x_1^*\| + \frac{1}{\sqrt{2}} \sum_{t=1}^{T} \|x_t - x_t^*\| + \sqrt{\frac{c\alpha + 2L_2\alpha}{2\alpha\lambda L_2}}\tilde{\mathcal{E}}_T + \mathcal{P}_T,$$

which implies that

$$\sum_{t=1}^{T} \|x_t - x_t^*\| \leq \frac{2}{2 - \sqrt{2}}\|x_1 - x_1^*\| + \frac{2}{2 - \sqrt{2}}\sqrt{\frac{c\alpha + 2L_2\alpha}{2\alpha\lambda L_2}}\tilde{\mathcal{E}}_T + \frac{2}{2 - \sqrt{2}}\mathcal{P}_T.$$

Plugging the above in 28 proves the claim. $\qquad\square$

In the above result, the number of OGD updates per round is on the order of $\mathcal{O}(L_2/\alpha)$, where $L_2/\alpha$ is the condition number of each loss function. Below, we also provide a dynamic regret bound for standard OGD (single update per round); as a result, we only provide the bound in terms of $\mathcal{P}_T$ (similar to (Jadbabaie et al., 2015; Mokhtari et al., 2016).

After establishing the dynamic regret for the inexact OGD, we can use this result to obtain the proof of Lemma 3 in the main paper, which provides the dynamic regret of inexact OGD over the loss sequences $\mathbb{E}_{\nu_t^*}[D_{KL}(\pi_t^*|\phi_t)]$ for all $t \in [T]$. Here, we present the full statement with constants for the Lemma 3.

**Lemma 11** (Dynamic regret bound for inexact OGD). *Denote $\ell_t(\phi_t) := \mathbb{E}_{\nu_t^*}[D_{KL}(\pi_t^*|\phi_t)]$ for all $t \in [T]$. For any dynamically varying comparator $\psi_t^* = \underset{\psi_t \in \Delta \mathcal{A}_\varrho^{|\mathcal{S}|}}{\arg\min} \sum_{t=1}^{T} \mathbb{E}_{\nu_t^*}[D_{KL}(\pi_t^*|\phi_t)]$ if OGD is run on a sequence of loss functions $\hat{\ell}_t(\phi_t)$, where $\hat{\ell}_t(\phi_t) = \mathbb{E}_{\nu_t^*}[D_{KL}(\hat{\pi}_t|\phi_t)]$ for all $t \in [T]$ with*

*the step-size $\alpha \leq \frac{1}{2\mu_\pi}$, number of iterations $K := \lceil \frac{\ln 2}{\ln(1+\mu_\pi\alpha)} \rceil$ then the following bound holds for dynamic regret for any $\beta > 0$:*

$$\sum_{t=1}^{T} \ell_t(\phi_t) - \sum_{t=1}^{T} \ell_t(\psi_t^*) \leq \min \left( C_1\|\phi_1 - \psi_1^*\|^2 + C_2\mathcal{E}_T + C_3\mathcal{S}_T + \frac{1}{2\beta}\sum_{t=1}^{T}\|\nabla\ell_t(\psi_t^*)\|^2, \right.$$

$$\left. C_4\|\phi_1 - \psi_1^*\| + C_5\tilde{\mathcal{E}}_T + C_4\mathcal{P}_T \right),$$

*where $C_1 = 2(L_\pi + \beta)$, $C_2 = (L_\pi + \beta)\frac{3C_6\alpha + 6\alpha L_\pi}{2\mu_\pi\alpha L_\pi}$, $C_3 = 3(L_\pi + \beta)$, $C_4 = \frac{2L_g}{2-\sqrt{2}}$, and $C_5 = \frac{2L_g}{2-\sqrt{2}}\sqrt{\frac{C_6\alpha + 2L_\pi\alpha}{2\alpha\mu_\pi L_\pi}}$, for any $C_6 \in \{c \in \left(0, \frac{2}{L_\pi}\right) : \psi_t^* + c(\hat{\nabla}_t - \nabla_t) \in \Delta\mathcal{A}_\varrho^{|S|}\}$ where $\hat{\nabla}_t$ and $\nabla_t$ are an $\epsilon_t$-subgradient and exact subgradient of $\mathbb{E}_{\nu_t^*}[D_{KL}(\pi_t^*|\psi_t)]$ at $\psi_t$, respectively, $\mathcal{E}_T := \sum_{t=1}^{T}\epsilon_t$ is the cumulative inexactness.*

*Proof.* The proof directly follows after plugging in the constants from Theorem E.3. □

## F  KL DIVERGENCE ESTIMATION ERROR BOUND

We recall the following notations. For each task $t$, the initial state distribution is denoted by $\rho_t$, the state distribution for the optimal policy $\pi_t^*$ is given by $\nu_t^*$, the state distribution for the policy $\hat{\pi}_t$ is denoted by $\tilde{\nu}_t$, and the state distribution estimated using the trajectory sample dataset $\mathcal{D}_t$ is denoted as $\hat{\nu}_t$.

In the main paper, we breakdown the KL divergence estimation error by the sources of origin:

$$\mathbb{E}_{\nu_t^*}[D_{KL}(\pi_t^*|\pi)] - \mathbb{E}_{\hat{\nu}_t}[D_{KL}(\hat{\pi}_t|\pi)] = \underbrace{\mathbb{E}_{\nu_t^*}[D_{KL}(\pi_t^*|\pi)] - \mathbb{E}_{\tilde{\nu}_t}[D_{KL}(\pi_t^*|\pi)]}_{(A)} \tag{29}$$

$$+ \underbrace{\mathbb{E}_{\tilde{\nu}_t}[D_{KL}(\pi_t^*|\pi)] - \mathbb{E}_{\hat{\nu}_t}[D_{KL}(\pi_t^*|\pi)]}_{(B)} + \underbrace{\mathbb{E}_{\hat{\nu}_t}[D_{KL}(\pi_t^*|\pi)] - \mathbb{E}_{\hat{\nu}_t}[D_{KL}(\hat{\pi}_t|\pi)]}_{(C)},$$

where $(A)$ accounts for the mismatch between the discounted state visitation distributions of an optimal policy $\pi_t^*$ and a suboptimal one $\hat{\pi}_t$, $(B)$ originates from the estimation error of DualDICE, and $(C)$ is due to the difference between $\pi_t^*$ and $\hat{\pi}_t$ measured according to $\hat{\pi}_t$. By the triangle inequality, we can bound the total error by controlling each term separately. This decomposition is general in the sense that it provides a guideline to bound each term with potentially different strategies. In particular, the term $(B)$ can be bounded differently if we replace DualDICE with another stationary distribution estimation algorithm. To bound the terms $(A)$ and $(C)$, we have developed new techniques based on tame geometry and subgradient flow systems. To streamline the presentation, we consider the tabular setting with softmax parametrization.

To bound $(A)$, we need to control the distance between $\nu_t^*$ and $\tilde{\nu}_t$, which can be bounded by the distance between the inducing policy parameters as long as they are Lipschitz continuous (Xu et al., 2020, Lemma 3). In addition, the bound on $(C)$ also depends on the distance between policies. In general, controlling the distance between a policy to an optimal policy based on the suboptimality gap requires the optimization to have some curvatures around the optima (e.g., quadratic growth (Drusvyatskiy & Lewis, 2018) or Hölderian growth (Johnstone & Moulin, 2020)). However, to the best of the knowledge of the authors, the only available results are algorithm-dependent PL inequalities for policy gradient (Mei et al., 2020) or quadratic growth with entropy regularization (Ding et al., 2021b).

**Discussion on Assumption 1:** As discussed in the main text, Assumption 1 implies boundedness and Lipschitzness of the KL divergence. We make use of this in bounding the terms $(A)$ and $(C)$ in (29) and eventually obtain Theorems 3.1 and 3.3. We expect that Assumption 1 is also needed in unconstrained meta-learning by adapting our method, i.e., the MDP-within-online framework. Technically, Assumption 1 is a minimum requirement even for single-task CRPO to provide provable guarantees. This can be seen in the convergence guarantee of the original CRPO method (Lemma 4 and Lemma 21 in our paper, or Theorem 3 in (Xu et al., 2021) last line of their proof before the term $D_{KL}(\pi_t^*|\pi_{t,0})$ is submerged in the big-O notation). For example, as shown in our (2),

$$R_0 = J_{t,0}(\pi_t^*) - \mathbb{E}[J_{t,0}(\hat{\pi}_t)] \le \frac{2}{\alpha_t M}\mathbb{E}_{s \sim \nu_t^*}[D_{KL}(\pi_t^*|\pi_{t,0})] + \frac{4\alpha_t c_{max}^2 |\mathcal{S}||\mathcal{A}|}{(1-\gamma)^3}.$$

To ensure that the bound is nontrivial, we need to bound the term $D_{KL}(\pi_t^*|\pi_{t,0})$. However, if $\pi_{t,0}$ does not have full support over the state/action space, then there may be a state $s$ where $\pi_t^*(s) > 0$ but $\pi_{t,0}(s) = 0$, which would make the KL divergence infinite.

### F.1 Preliminaries on tame geometry

For the sake of completeness, let us recall some fundamental concepts/results in tame geometry, which allows us to study the global geometry of the solution maps of a wide range of optimization problems, which will be used in bounding the estimation error for the KL divergence. More information can be found in (Davis et al., 2020; Van den Dries & Miller, 1996). Recall that a class of functions on a bounded set is called $C^p$ smooth when it possesses the uniformly bounded partial derivatives up to order $p$.

**Definition 3** (Whitney Stratification). *A Whitney $C^k$ stratification of a set $I$ is a partition of $I$ into finitely many nonempty $C^k$ manifolds, called strata, satisfying the following compatibility conditions:*

1. *For any two strata $I_a$ and $I_b$, the implication $I_a \cap I_b \ne \emptyset$ implies that $I_a \subset \mathrm{cl}I_b$ holds, where $\mathrm{cl}I_b$ denotes the closure of the set $I_b$.*

2. *For any sequence of points $x_k$ in a stratum $I_a$, converging to a point $x^\star$ in a stratum $I_b$, if the corresponding normal vectors $v_k \in N_{I_a}(x_k)$ converge to a vector $v$, then the inclusion $v \in N_{I_b}(x^\star)$ holds. Here $N_{I_a}(x_k)$ denotes the normal cone to $I_a$ at $x_k$.*

Roughly speaking, stratification is a locally finite partition of a given set into differentiable manifolds, which fit together in a regular manner (property 1 in Def. 3). Whitney stratification as defined above is a special type of stratification for which the strata are such that their tangent spaces (as viewed from normal cones) also fit regularly (property 2).

There are several paths to verifying Whitney stratifiability. For instance, one can show that the function under study belongs to one of the well-known function classes, such as semialgebraic functions (Davis et al., 2020), whose members are known to be Whitney stratifiable. However, to study the solution function of a general convex optimization problem, we need a far-reaching axiomatic extension of semialgebraic sets to classes of functions definable on "o-minimal structures," which are very general classes and share several attractive analytic features as semialgebraic sets, including Whitney stratifiability (Davis et al., 2020; Van den Dries & Miller, 1996).

**Definition 4** (o-minimal structure). *(Van den Dries & Miller, 1996) An o-minimal structure is defined as a sequence of Boolean algebras $O_v$ of subsets of $\mathbb{R}^v$, such that for each $n_v \in \mathbb{N}$, the following properties hold:*

1. *If some set $X$ belongs to $O_v$, then $X \times \mathbb{R}$ belong to $O_{v+1}$.*

2. *Let $P_{proj} : \mathbb{R}^v \times \mathbb{R} \to \mathbb{R}^v$ denote the coordinate projection operator onto $\mathbb{R}^v$, then for any $X$ in $O_{v+1}$, the set $P_{proj}(X)$ belongs to $O_v$.*

3. *$O_v$ contains all sets of the form $\{x \in \mathbb{R}^v : y(x) = 0\}$, where $y(x)$ is a polynomial in $\mathbb{R}^v$.*

4. *The elements of $O_1$ are exactly the finite unions of intervals (possibly infinite) and points.*

*Then all the sets that belong to $O_v$ are called definable in the o-minimal structure.*

Definable sets have broader applicability than semialgebraic sets (in the sense that the latter is a special kind of definable sets) but enjoys the same, remarkable stability property: the composition of definable mappings (including sum, inf-convolution, and several other classical operations of analysis involving a finite number of definable objects) in some o-minimal structure remains in the same structure. We will crucially exploit these properties in the following sections.

## F.2 Basic properties of subgradient flow systems

We also recall some basic definitions and properties of the subgradient flow system (see, e.g., (Bolte et al., 2010, Thm. 13)). Let $f : \mathbb{R}^d \to \mathbb{R} \cup \{+\infty\}$ be a proper lower semicontinuous function.

**Definition 5** (Subgradient flow system). *For every $x \in \mathrm{dom}(f)$, there exists a unique absolutely continuous curve (called trajectory or subgradient curve) $\theta(\tau) : [0, +\infty) \to \mathbb{R}^d$ that satisfies*

$$\begin{cases} \dot{\theta}(\tau) \in -\partial f(\theta(\tau)) & \text{a.e. on} \quad (0, +\infty) \\ \theta(0) = \theta_0 \in \mathrm{dom}(f). \end{cases} \tag{30}$$

Moreover, the trajectory also satisfies the following properties (Bolte et al., 2010, Thm. 13):

1. $\theta(\tau) \in \mathrm{dom}(\partial f)$ for all $\tau \in (0, +\infty)$.

2. For all $\tau > 0$, the right derivative $\dot{\theta}(\tau^+)$ is well defined and equal to
$$\dot{\theta}(\tau^+) = -\partial^0 f(\theta(\tau)),$$
   where $\partial^0 f(\theta)$ is the minimum norm subgradient in $\partial f(\theta)$. In particular, we have that $\dot{\theta}(\tau) = -\partial^0 f(\theta(\tau))$, for almost all $\tau$.

## F.3 Bounding the distance $\|\hat{\theta}_t - \theta_t^*\|$

Recall Assumption 2, which requires that the objective/constraint functions and policy parametrization are definable in some o-minimal structure (Van den Dries & Miller, 1996). This is a mild assumption as practically all functions from real-world applications, including deep neural networks, are definable in some o-minimal structure (Davis et al., 2020); also, the composition of mappings, along with the sum, inf-convolution, and several other classical operations of analysis involving a finite number of definable objects in some o-minimal structure remains in the same structure (Van den Dries & Miller, 1996). The far-reaching consequence of definability, exploited in this study, is that definable sets and functions admit, for each $k \geq 1$, a $C^k$–Whitney stratification with finitely many strata (see, for instance, (Van den Dries & Miller, 1996, Result 4.8)). This remarkable property, combined with the result that any stratifiable functions enjoys a nonsmooth Kurdyka–Łojasiewicz inequality (Bolte et al., 2007), provides the foundation to bound the distance $\|\pi_t^* - \hat{\pi}_t\|$ by the suboptimality gap. Note that without further specifications, $\pi_t^*$ is understood as one of the optimal policies that are closest to the policy $\hat{\pi}_t$ (i.e., the projection of $\hat{\pi}_t$ onto the optimal policy set).

We start with the following elementary result. Here and throughout the section, we use $\mathcal{F}_{t,\tilde{d}} = \{\pi_{t,\theta} : J_{t,i}(\pi_{t,\theta}) \leq \tilde{d}_{t,i}\}$ to denote the feasible set with upper bounds $\tilde{d}$. Note that $\mathcal{F}_{t,d}$ is the original feasible set. We also let $\mathbb{I}_{\mathcal{F}_{t,\tilde{d}}}(\cdot)$ be the indicator function for the set $\mathcal{F}_{t,\tilde{d}}$.

**Lemma 12.** *The function (with variable $\theta$) $J_{t,0}(\pi_{t,\theta}) + \mathbb{I}_{\mathcal{F}_{t,\tilde{d}}}(\pi_{t,\theta})$, where $\tilde{d}_t$ is any vector such that $\mathcal{F}_{t,\tilde{d}}$ is non-empty, is definable.*

*Proof.* Since $J_{t,i}(\cdot)$ is definable for $i = 1, ..., p$, by the rule of composition, which is due to the definable counterpart of the Tarski-Seidenberg theorem, $J_{t,i}(\pi_{t,\theta}) - d_{t,i}$ is definable for $i = 1, ..., p$. Thus, $\mathcal{F}_{t,\tilde{d}}$ is definable on the same o-minimal structure by definition. Furthermore, $\mathbb{I}_{\mathcal{F}_{t,\tilde{d}}}(\cdot)$ is definable as the indicator of $\mathcal{F}_{t,\tilde{d}}$. The definability of $J_{t,0}(\pi_{t,\theta})$ follows similarly. Since definability is preserved under addition, the function $J_{t,0}(\pi_{t,\theta}) + \mathbb{I}_{\mathcal{F}_{t,\tilde{d}}}(\pi_{t,\theta})$ is definable. $\qquad \square$

For the convenience of the reader, we restate the result for non-smooth Kurdyka–Łojasiewicz (KL) inequality from (Bolte et al., 2007, Thm. 14).

**Proposition 1** (Non-smooth Kurdyka–Łojasiewicz inequality). *Let $f$ be a lower semicontinuous definable function. There exists $\rho > 0$, a strictly increasing continuous definable function $h : [0, \rho] \to (0, \infty)$ which is $C^1$ smooth on $(0, \rho)$, with $h(0) = 0$, and a continuous definable function $\mathcal{X} : \mathbb{R}_+ \to (0, \rho)$ such that*

$$\|\partial^0 f(x)\| \geq \frac{1}{h'(|f(x)|)},$$

*whenever $0 < |f(x)| \leq \mathcal{X}(\|x\|)$.*

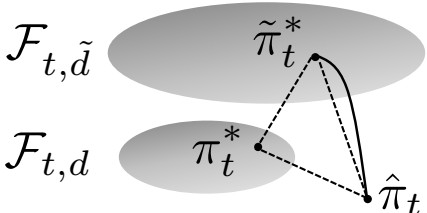

Figure 3: To bound the distance between $\pi_t^*$ and $\hat{\pi}_t$, we first bound the distance between $\tilde{\pi}_t^*$ and the optimal policy with respect to a larger feasible set $\mathcal{F}_{t,\tilde{d}}$ by an argument based on subgradient flow curve. Note that $\hat{\pi}_t \in \mathcal{F}_{t,\tilde{d}}$ may be infeasible with respect to the original set of constraints but feasible with respect to the relaxed constraints. We then bound the distance between the optimal policies $\pi_t^*$ and $\tilde{\pi}_t^*$, which correspond to the original feasible set $\mathcal{F}_{t,d}$ and the enlarged set $\mathcal{F}_{t,\tilde{d}}$. By the triangle inequality, we can then derive the desired bound on the distance between $\pi_t^*$ and $\hat{\pi}_t$. Note that for better visualization, we vertically separate the sets $\mathcal{F}_{t,d}$ and $\mathcal{F}_{t,\tilde{d}}$, which also aims to indicate that in general the optimal solution $\tilde{\pi}_t^*$ has a higher objective than $\pi_t^*$ due to the relaxed constraints.

Let $\theta$ and $\theta_t^*$ denote the parameters of a policy $\pi_\theta$ and $\pi_t^*$, respectively. Directly bounding the distance between $\theta$ and $\theta_t^*$ is difficult because $\pi$ may be infeasible (this is even true for $\hat{\pi}_t$, since it is only guaranteed to approximately satisfy the constraints), i.e., $\theta \notin \mathcal{F}_{t,d}$. Thus, the typical approach of following the subgradient flow of $J_{t,0}(\pi_\theta) + \mathbb{I}_{\mathcal{F}_{t,d}}(\pi_\theta)$ to reach $\theta_t^*$ is not applicable. The idea is to enlarge the feasible set $\mathcal{F}_{t,d}$ by increasing the violation threshold $\tilde{d}_{t,i} \geq d_{t,i} + \delta$, for any $\delta > 0$, such that with high probability, $\theta \in \mathcal{F}_{t,\tilde{d}}$. Then by following the subgradient flow for $J_{t,0}(\pi_\theta) + \mathbb{I}_{\tilde{\mathcal{F}}_t}(\pi_\theta)$, we can arrive at a critical point $\tilde{\theta}_t^*$ (corresponding to the policy $\tilde{\pi}_t^*$), which is most likely different from $\theta_t^*$. It then remains to bound the distance between $\theta_t^*$ and $\tilde{\theta}_t^*$, which is possible due to the preservation of definability through $\inf$ projection. This is the roadmap we will follow. A graphical illustration of the approach is shown in Fig. 3.

**Bounding the term** $\|\theta_t^* - \tilde{\theta}_t^*\|$**.** In this part, we will bound the term $\|\theta_t^* - \tilde{\theta}_t^*\|$, which will be used to bound $\|\pi_t^* - \tilde{\pi}_t^*\|$. Firstly, we will prove that the parameter $\theta_t$, which represents the solution map of an optimization with definable objective and constraints, is definable.

**Proposition 2.** *Let* $\theta_t(d) \in \arg\min\{J_{t,0}(\pi_{t,\theta}), \text{ s.t. } J_{t,i}(\pi_{t,\theta}) \leq d_{t,i}, \forall i = 1, ..., p\}$ *be the solution map of the constraint parameters* $d$*. Then, the function* $\theta_t(d)$ *is continuous and definable. Furthermore, there exists a finite partition of the space such that the restriction of* $\theta_t(d)$ *to each partition is* $C^p$ *smooth.*

*Proof.* First, it can be seen that the solution map $\theta_t(d) \in \arg\min J_{t,0}(\pi_{t,\theta}) + \mathbb{I}_{\mathcal{F}_{t,d}}(\pi_{t,\theta})$. Let $\phi_t(d) = \min J_{t,0}(\pi_{t,\theta}) + \mathbb{I}_{\mathcal{F}_{t,d}}(\pi_{t,\theta})$ be the optimal value function. Since $J_{t,0}(\pi_{t,\theta}) + \mathbb{I}_{\mathcal{F}_{t,d}}(\pi_{t,\theta})$ is definable by Proposition 12, and definability is preserved under $\inf$ projection, $\phi_t(d)$ is definable. Since $\theta_t(d) = \{\theta : J_{t,0}(\pi_{t,\theta}) + \mathbb{I}_{\mathcal{F}_{t,d}}(\pi_{t,\theta}) = \phi_t(d)\}$, by the Tarski-Seidenberg Theorem, $\theta_t(d)$ is definable. The continuity property follows directly from Berge's Maximum theorem.

Following the discussion of Whitney stratifications in (Bolte et al., 2007), since the graph of a definable function is Whitney stratifiable, we can construct a partition by projecting the stratification into the function domain, which will be a Whitney $C^p$-stratification by the constant rank theorem. Furthermore, the restriction of the definable function to each stratum is $C^p$-smooth. Alternatively, we can directly use the fact that for any definable function, there exists a $C^p$-decomposition which has a finite number of cells, and the restriction to each cell is $C^p$-smooth (Van den Dries & Miller, 1996). This completes the proof. $\square$

Now that we have proved that the function $\theta_t(d)$ is definable, we can obtain the bound $\|\theta_t^* - \tilde{\theta}_t^*\|$. Intuitively, our proof exploits the fact that continuous and definable functions exhibit controlled behaviors along any path, even if it crosses over a finite number of Whitney strata.

**Lemma 13.** *For any* $\tilde{d}$ *such that* $\mathcal{F}_{t,\tilde{d}}$ *is non-empty, the following holds:*

$$\|\theta_t^* - \tilde{\theta}_t^*\| = \|\theta(d) - \theta(\tilde{d})\| = \mathcal{O}(\|d - \tilde{d}\|).$$

*Proof.* Since every smooth function over a bounded set is Lipschitz, let us denote $L_d$ as the maximum of the Lipschitz constants for all the cells of the Whitney stratification of $\theta_t(d)$. Let $d(\lambda) = \lambda d + (1 - \lambda)\tilde{d}$, where $0 \leq \lambda \leq 1$, be the curve that connects between $d$ and $\tilde{d}$. Also, let $0 = \lambda_1 \leq ... \leq \lambda_n = 1$ be the partition such that $\theta_t(d(\lambda))$ belongs to one cell for all $\lambda_i < \lambda < \lambda_{i+1}$ for $i = 1, ..., n - 1$. We know that $n < \infty$ since $d(\theta_t)$ is Whitney stratifiable. Thus,

$$\|\theta_t(d) - \theta_t(\tilde{d})\| \leq \sum_{i=1}^{n-1} \|\theta_t(d(\lambda_i)) - \theta_t(d(\lambda_{i+1}))\|$$

$$\leq L_d \sum_{i=1}^{n-1} \|d(\lambda_i) - d(\lambda_{i+1})\|$$

$$\leq L_d \|d - \tilde{d}\| \sum_{i=1}^{n-1} |\lambda_{i+1} - \lambda_i|$$

$$= L_d \|d - \tilde{d}\|$$

where the first inequality is due to triangle inequality, the second inequality is due to Lipschitz continuity, the third inequality is due to the definition of $d(\lambda)$, the first equality is due to the non-decreasing sequence of $\lambda_i$. $\qquad\square$

**Bounding the term $\|\tilde{\theta}_t^* - \hat{\theta}_t\|$.** Recall that $\hat{\theta}_t$ is the parameter for $\hat{\pi}_t$ (output of within-task CRPO), and $\tilde{\theta}_t^*$ is the parameter for an optimal solution with an enlarged feasible set $\mathcal{F}_{t,\tilde{d}}$. In this subsection, we will obtain the upper bound for the term $\|\tilde{\theta}_t^* - \hat{\theta}_t\|$. Let $f(\theta, \tilde{d}) = J_{t,0}(\pi_\theta) + \mathbb{I}_{\mathcal{F}_{t,\tilde{d}}}(\pi_\theta)$, and choose $\tilde{d}_i = \mathcal{O}(1/\sqrt{M})$ for $i = 1, ..., p$, which coincides with the upper bound on constraint violation for within-task CRPO such that $\hat{\theta}_t \in \mathcal{F}_{t,\tilde{d}}$ with high probability. In the next result, we will condition on this high-probability event.

**Lemma 14.** *With the choice of $\tilde{d} = d + \delta$, where $\delta = \mathcal{O}(1/\sqrt{M})$ coincides with the upper bound on constraint violation for within-task CRPO such that $\hat{\theta}_t \in \mathcal{F}_{t,\tilde{d}}$, the following holds:*

$$\|\tilde{\theta}_t^* - \hat{\theta}_t\| \leq \mathcal{O}\left(h\left(1/\sqrt{M}\right)\right),$$

*where $h$ is a strictly increasing continuous function with the property that $h(0) = 0$ as specified in Lemma 1.*

*Proof.* Without loss of generality, consider $f(\theta) = J_{t,0}(\pi_\theta) + \mathbb{I}_{\mathcal{F}_{t,\tilde{d}}}(\pi_\theta) + c$, where $c := -\inf J_{t,0}(\pi_\theta) + \mathbb{I}_{\tilde{\mathcal{F}}_t}(\pi_\theta)$, so that the minimal value of $f$ is translated to 0. For simplicity, also assume that $f(\hat{\theta}_t) \leq \mathcal{X}(\|\hat{\theta}_t\|)$. Note that this assumption can be relaxed by using the concept of "curves of maximal slope" at the cost of slightly more complicated analysis and bounds (Ioffe, 2009).

Now, consider a subgradient flow $\dot{\theta}(\tau) \in -\partial f(\theta(\tau))$ (see Definition 5), initialized at $\theta(0) = \hat{\theta}_t$ then, for any $0 \leq s' < s$, we have that

$$h\big(f(\theta(s'))\big) - h\big(f(\theta(s))\big) = \int_s^{s'} \frac{d}{d\tau} h\big(f(\theta(\tau))\big) d\tau$$

$$= \int_{s'}^s h'\big(f(\theta(\tau))\big) \|\dot{\theta}(\tau)\|^2 d\tau$$

$$\geq \int_{s'}^s \|\dot{\theta}(\tau)\| d\tau$$

$$\geq \left\| \int_{s'}^s \dot{\theta}(\tau) d\tau \right\|$$

$$= \|\theta(s) - \theta(s')\|$$

where the second equality is due to the property of the subgradient flow (see Sec. F.2), the first inequality is due to $\|\partial^0(hf)\big(\theta(\tau)\big)\| \geq 1$ from Proposition 1, and the second inequality is due to the

triangle inequality. Thus, by taking $s' = 0$ and $s \to \infty$, we have shown that

$$h(f(\hat{\theta}_t)) \geq \|\hat{\theta}_t - \tilde{\theta}_t^*\|.$$

Therefore,

$$\|\hat{\theta}_t - \tilde{\theta}_t^*\| \leq h(f(\hat{\theta}_t) - f(\tilde{\theta}_t^*))$$
$$\overset{(i)}{\leq} h(J_{t,0}(\hat{\pi}_t) - J_{t,0}(\tilde{\pi}_t^*)),$$

where the first inequality is due to the optimality of $\tilde{\pi}_t^*$, and $(i)$ follows since both $\hat{\pi}_t$ and $\tilde{\pi}_t^*$ are feasible for $\mathcal{F}_{t,\tilde{d}}$. Note that the suboptimality bound can be split as

$$h(J_{t,0}(\hat{\pi}_t) - J_{t,0}(\tilde{\pi}_t^*)) = h\left(J_{t,0}(\hat{\pi}_t) - J_{t,0}(\pi_t^*) + J_{t,0}(\pi_t^*) - J_{t,0}(\tilde{\pi}_t^*)\right).$$

By CRPO, we can bound the value difference $J_{t,0}(\hat{\pi}_t) - J_{t,0}(\pi_t^*) \leq \mathcal{O}(1/\sqrt{M})$. Moreover, Since the value function is Lipschitz (Xu et al., 2020, Lemma 4), the value difference $J_{t,0}(\pi_t^*) - J_{t,0}(\tilde{\pi}_t^*)$ can be bounded by the distance $\|\theta_t^* - \tilde{\theta}_t^*\|$, which is bounded again by $\mathcal{O}(1/\sqrt{M})$ according to Lemma 13. Hence, recognizing that $h$ is strictly increasing, we have proved the claim. $\square$

**Bounding the term** $\|\theta_t^* - \hat{\theta}_t\|$**.** Finally, we are able to bound the term of our original interests.

**Lemma 15.** *Under assumption 2, the following holds:*

$$\|\theta_t^* - \hat{\theta}_t\| \leq \mathcal{O}\left(h\left(\frac{1}{\sqrt{M}}\right) + \frac{1}{\sqrt{M}}\right),$$

*where $h$ is a strictly increasing continuous function with the property that $h(0) = 0$ as specified in Lemma 1.*

*Proof.* The claim follows directly from Lemmas 13 and 14 and the triangle inequality. $\square$

**Remark 4.** *Note that our strategy to bound $\|\theta_t^* - \hat{\theta}_t\|$ is algorithmic-agnostic as it only relies on the optimization landscape. The only place we rely on the algorithm is to bound the suboptimality gap, which is then converted to a bound on $\|\tilde{\theta}_t^* - \hat{\theta}_t\|$ in Lemma 14. Also, the enlargement of the feasible set should be viewed as a proof technique and has no implications for the algorithm design. Indeed, the motivation for the enlargement is to properly design a subgradient flow system. Thus, the result of Lemma 15 is not conditioned on how the enlargement is performed. Also, note that definability is used differently in Lemmas 13 and 14. In the former case, we exploit the Whitney stratification property to provide an upper bound, while in the latter case, we exploit the KL property to obtain a lower bound, hence they serve different purposes.*

## F.4 BOUNDING TERM (A): $|\mathbb{E}_{\nu_t^*}[D_{KL}(\pi_t^*|\pi)] - \mathbb{E}_{\tilde{\nu}_t}[D_{KL}(\pi_t^*|\pi)]|$

The result from Lemma 15 can be used directly to provide bounds for *(A)* and *(C)*. We start with the term *(A)*.

**Lemma 16.** *The following bound holds:*

$$|\mathbb{E}_{s \sim \nu_t^*}[D_{KL}(\pi_t^*|\pi)] - \mathbb{E}_{s \sim \tilde{\nu}_t}[D_{KL}(\pi_t^*|\pi)]| = \mathcal{O}\left(h\left(\frac{1}{\sqrt{M}}\right) + \frac{1}{\sqrt{M}}\right) \tag{31}$$

*where $h$ is a strictly increasing continuous function with the property that $h(0) = 0$ as specified in Lemma 1.*

*Proof.*

$$|\mathbb{E}_{s \sim \nu_t^*}[D_{KL}(\pi_t^*|\pi)] - \mathbb{E}_{s \sim \tilde{\nu}_t}[D_{KL}(\pi_t^*|\pi)]|$$
$$= \left|\sum_{s \in \mathcal{S}_t} \left(\nu_t^*(s) - \tilde{\nu}_t(s)\right) D_{KL}(\pi_t^*(s)|\pi(s))\right|$$
$$\leq C_\pi \|\nu_t^* - \tilde{\nu}_t\|_1$$
$$\leq 2C_\pi C_\nu \|\theta_t^* - \hat{\theta}_t\|_2$$

where the first equality is by definition, the first inequality is due to Assumption 1, and the second inequality is due to (Xu et al., 2020, Lem. 3), which also specifies the constant $C_\nu$, and (Levin & Peres, 2017, Prop. 4.2). The result then follows by recalling the result from Lemma 15. $\qquad\square$

## F.5 BOUNDING TERM (C): $|\mathbb{E}_{\hat{\nu}_t}[D_{KL}(\pi_t^*|\pi)] - \mathbb{E}_{\hat{\nu}_t}[D_{KL}(\hat{\pi}_t|\pi)]|$

Similarly, we can prove the upper bound for the error term $(C)$.

**Lemma 17.** *The following bound holds:*

$$|\mathbb{E}_{s\sim\hat{\nu}_t}[D_{KL}(\pi_t^*|\pi)] - \mathbb{E}_{s\sim\hat{\nu}_t}[D_{KL}(\hat{\pi}_t|\pi)]| = \mathcal{O}\left(h\left(\frac{1}{\sqrt{M}}\right) + \frac{1}{\sqrt{M}}\right) \qquad (32)$$

*Proof.*

$$|\mathbb{E}_{s\sim\hat{\nu}_t}[D_{KL}(\pi_t^*|\pi)] - \mathbb{E}_{s\sim\hat{\nu}_t}[D_{KL}(\hat{\pi}_t|\pi)]|$$
$$= \left|\sum_{s\in\mathcal{S}} \hat{\nu}_t(s)\left(D_{KL}(\pi_t^*|\pi) - D_{KL}(\hat{\pi}_t|\pi)\right)\right|$$
$$\leq \sum_{s\in\mathcal{S}} \hat{\nu}_t(s)\left|D_{KL}(\pi_t^*|\pi) - D_{KL}(\hat{\pi}_t|\pi)\right|$$
$$\leq L_g \sum_{s\in\mathcal{S}} \hat{\nu}_t(s)\|\pi_t^*(s) - \hat{\pi}_t(s)\|_2$$
$$\leq L_g \sum_{s\in\mathcal{S}} \hat{\nu}_t(s)\|\theta_t^* - \hat{\theta}_t - c'1\|_\infty$$
$$\leq L_g\|\theta_t^* - \hat{\theta}_t\|_2$$

where the first inequality is due to the non-negativity of $\hat{\nu}_t(s)$ and triangle inequality, the second inequality is due to Assumption 1, the third inequality holds for any constant $c'$ and is due to (Mei et al., 2020, Lem. 24), and the last inequality is due to $\sum_{s\in\mathcal{S}} \hat{\nu}_t(s) = 1$ and by choosing $c' = 0$. The result then follows by recalling the result from Lemma 15. $\qquad\square$

## F.6 BOUNDING TERM (B): $|\mathbb{E}_{\tilde{\nu}_t}[D_{KL}(\pi_t^*|\pi)] - \mathbb{E}_{\hat{\nu}_t}[D_{KL}(\pi_t^*|\pi)]|$

Now, we will upper bound the error term $(B)$. The proof follows DualDICE (Nachum et al., 2019). We introduce the following notations. Let $\hat{\mathbb{E}}_{d^{\mathcal{D}_t}}$ denote an average of empirical samples where $\{s_i, a_i, r_i, s_i'\}_{i=1}^N \sim d^{\mathcal{D}_t}$, and $\rho_t$ be the initial state distribution for the CMDP task $t$. The number of data points $N = \mathcal{O}(M^{1+1/\sigma})$, where $\sigma$ is any positive number $\sigma \in (0,1)$. Note that the additional factor of $\mathcal{O}(M^{1/\sigma})$ results from the critic evaluation per policy update (see (Xu et al., 2021, Thm. 1)). We will roughly bound $N = \mathcal{O}(M^2)$ in the following to simplify the presentation. The stationary distribution correction factor is denoted as $w_{\hat{\pi}_t/\mathcal{D}_t}(s,a) = \frac{\tilde{\nu}_t(s,a)}{d^{\mathcal{D}_t}(s,a)}$.

We make the following regularity assumption on the distribution $d^{\mathcal{D}_t}$ with respect to the target policy $\hat{\pi}_t$ (Nachum et al., 2019, Asm. 1).

**Assumption 3** (Reference distribution property). *For any $(s,a)$, $\tilde{\nu}_t(s,a) > 0$ implies that $d^{\mathcal{D}_t}(s,a) > 0$. Furthermore, the correction terms are bounded by some finite constant $C_\omega$: $\|\omega_{\tilde{\nu}_t/\mathcal{D}_t}\|_\infty \leq C_\omega$.*

For convenience, we recapitulate the key points from DualDICE, where we also omit the task dependence $t$ (i.e., we use $d^{\mathcal{D}}$, $\pi$, and $\rho$ in lieu of $d^{\mathcal{D}_t}$, $\hat{\pi}_t$, and $\rho_t$, respectively). The objective function is given by

$$J(z,\zeta) = \mathbb{E}_{(s,a,s')\sim,a'\sim(s')}\left[(z(s,a) - \gamma z(s',a'))\zeta(s,a) - \zeta(s,a)^2/2\right] \qquad (33)$$
$$- (1-\gamma)\,\mathbb{E}_{s_0\sim\beta,a_0\sim(s_0)}\left[z(s_0,a_0)\right]. \qquad (34)$$

The objective in the form prior to introduction of $\zeta$ is denoted as $J(z)$:

$$J(z) = \frac{1}{2}\mathbb{E}_{(s,a)\sim}\left[(z-z)(s,a)^2\right] - (1-\gamma)\,\mathbb{E}_{s_0\sim\beta,a_0\sim(s_0)}\left[z(s_0,a_0)\right]. \qquad (35)$$

Let $\hat{J}(z,\zeta)$ denotes the empirical surrogate of $J(z,\zeta)$ with optimal solution as $(\hat{z}^*, \hat{\zeta}^*)$. We denote $z_-^* \arg\min_{z\in} J(z)$ and $z^* = \arg\min_{z:S\times A\to\mathbb{R}} J(z)$. We denote $L(z) = \max_{\zeta\in\mathcal{H}} J(z,\zeta)$ and $\hat{L}(z) = \max_{\zeta\in\mathcal{H}} \hat{J}(z,\zeta)$ as the primal objectives, and $\ell(\zeta) = \min_{z\in\mathcal{F}} J(z,\zeta)$, $\hat{\ell}(\zeta) = \min_{z\in\mathcal{F}} \hat{J}(z,\zeta)$ as the dual objectives. We apply some optimization algorithm $OPT$ for optimizing $\hat{J}(z,\zeta)$ with samples $\{s_i, a_i, r_i, s_i'\}_{i=1}^N \sim$, $\{s_0^i\}_{i=1}^N \sim \beta$, and target actions $a_i' \sim (s_i'), a_0^i \sim (s_0^i)$ for $i = 1,\ldots,N$. The output of $OPT$ is denoted by $(\hat{z},\hat{\zeta})$. We also make the following definitions to capture the error of approximation with $\mathcal{F}$ for $z$ and $\mathcal{H}$ for $\zeta$ in optimizing $\hat{J}(z,\zeta)$:

$$\epsilon_{approx}(\mathcal{F}) := \sup_{z\in S\times A\to\mathbb{R}} \inf_{z_{\mathcal{F}}\in\mathcal{F}} (\|z_{\mathcal{F}} - z\|_{\mathcal{D},1} + \|z_{\mathcal{F}} - z\|_{\rho\pi,1}) \tag{36}$$

$$\epsilon_{approx}(\mathcal{H}) := \sup_{\zeta\in S\times A\to\mathbb{R}} \inf_{\zeta_{\mathcal{H}}\in\mathcal{H}} (\|\zeta_{\mathcal{H}} - \zeta\|_{\mathcal{D},1} + \|\zeta_{\mathcal{H}} - \zeta\|_{\rho\pi,1}) \tag{37}$$

$$\epsilon_{approx}(\mathcal{F},\mathcal{H}) := \epsilon_{approx}(\mathcal{F}) + \epsilon_{approx}(\mathcal{H}) \tag{38}$$

We also define

$$\epsilon_{opt} := \|\hat{\zeta} - \hat{\zeta}^*\|_{\mathcal{D}_t}^2 + \left\|(\hat{z}^* - \hat{\mathcal{B}}^\pi \hat{z}^*) - (\hat{z} - \hat{\mathcal{B}}^\pi \hat{z})\right\|_{\mathcal{D}_t}^2 \tag{39}$$

as the optimization error of OPT from DualDICE.

**Lemma 18.** *By estimating $\hat{\nu}_t$ with DualDICE, the following bound holds:*

$$|\mathbb{E}_{\tilde{\nu}_t}[D_{KL}(\pi_t^*|\pi)] - \mathbb{E}_{\hat{\nu}_t}[D_{KL}(\pi_t^*|\pi)]| = \mathcal{O}\left(\sqrt{\frac{1}{M} + \epsilon_{opt} + \epsilon_{approx}(\mathcal{F},\mathcal{H})}\right),$$

*Proof.* We begin with the following decomposition:

$$(\mathbb{E}_{\tilde{\nu}_t}[D_{KL}(\pi_t^*|\pi)] - \mathbb{E}_{\hat{\nu}_t}[D_{KL}(\pi_t^*|\pi)])^2 \leq$$

$$\underbrace{2\left(\hat{\mathbb{E}}_{d^{\mathcal{D}_t}} \sum_a ((\hat{z} - \hat{\mathcal{B}}^{\hat{\pi}_t}\hat{z})(s,a) - (\hat{z}^* - \hat{\mathcal{B}}^{\hat{\pi}_t}\hat{z}^*)(s,a)) D_{KL}(\pi_t^*|\pi)\right)^2}_{\epsilon_1}$$

$$+ \underbrace{2\left(\hat{\mathbb{E}}_{d^{\mathcal{D}_t}} \sum_a (\hat{z}^* - \hat{\mathcal{B}}^{\hat{\pi}_t}\hat{z}^*)(s,a) D_{KL}(\pi_t^*|\pi) - \mathbb{E}_{d^{\mathcal{D}_t}} \sum_a \omega_{\hat{\pi}_t/\mathcal{D}_t}(s,a) D_{KL}(\pi_t^*|\pi)\right)^2}_{\epsilon_2}.$$

We will bound each term above separately.

$$\epsilon_1 \leq 2C_\pi^2 \left(\hat{\mathbb{E}}_{d^{\mathcal{D}_t}} \sum_a (\hat{z} - \hat{\mathcal{B}}^{\hat{\pi}_t}\hat{z})(s,a) - (\hat{z}^* - \hat{\mathcal{B}}^{\hat{\pi}_t}\hat{z}^*)(s,a)\right)^2$$

$$\leq 2C_\pi^2 \left(\underbrace{\|\hat{\zeta} - \hat{\zeta}^*\|_{\mathcal{D}_t}^2 + \left\|(\hat{z}^* - \hat{\mathcal{B}}^\pi \hat{z}^*) - (\hat{z} - \hat{\mathcal{B}}^\pi \hat{z})\right\|_{\mathcal{D}_t}^2}_{\epsilon_{opt}}\right),$$

where the error $\epsilon_{opt}$ is induced by the optimization OPT. The error term $\epsilon_2$ can be decomposed as

$$\epsilon_2 \leq 2\,C_\pi^2 \underbrace{\left(\hat{\mathbb{E}}_{d^{\mathcal{D}_t}} \sum_a (\hat{z}^* - \hat{\mathcal{B}}^{\hat{\pi}_t}\hat{z}^*)(s,a) - \mathbb{E}_{d^{\mathcal{D}_t}} \sum_a (\hat{z}^* - \mathcal{B}^{\hat{\pi}_t}\hat{z}^*)(s,a)\right)^2}_{\epsilon_{stat}}$$

$$+ 2C_\pi^2 \left(\mathbb{E}_{d^{\mathcal{D}_t}} \sum_a ((\hat{z}^* - \mathcal{B}^{\hat{\pi}_t}\hat{z}^*)(s,a) - \omega_{\hat{\pi}_t/\mathcal{D}_t}(s,a))\right)^2 \tag{40}$$

$$= 2\epsilon_{stat} + 2C_\pi^2 \left(\mathbb{E}_{d^{\mathcal{D}_t}} \sum_a ((\hat{z}^* - \mathcal{B}^{\hat{\pi}_t}\hat{z}^*)(s,a) - (z^* - \mathcal{B}^{\hat{\pi}_t}z^*)(s,a))\right)^2,$$

where the equality is due to the result that $z^* - \mathcal{B}^{\hat{\pi}_t}z^*(s,a) = \omega_{\hat{\pi}_t/\mathcal{D}_t}(s,a)$ (see (Nachum et al., 2019, Eq. 17)) and $\epsilon_{stat}$ is the error due to the finite number error. By (Nachum et al., 2019, Lem.

7), $\epsilon_{stat} = \mathcal{O}\left(\frac{\log M + \log \frac{1}{\delta}}{M^2}\right)$ with probability at least $1 - \delta$, where we use the bound on the number of data as $\mathcal{O}(M^2)$. To bound the second term, use the fact that $J(z)$ as defined in 35 is 1-strongly convex. Hence,

$$
\begin{aligned}
&\left(\mathbb{E}_{d^{\mathcal{D}_t}} \sum_a \left(\left(\hat{z}^* - \mathcal{B}^{\hat{\pi}_t}\hat{z}^*\right)(s, a) - \left(z^* - \mathcal{B}^{\hat{\pi}_t}z^*\right)(s, a)\right)\right)^2 \\
&\leq \|\left(\hat{z}^* - \mathcal{B}^{\hat{\pi}_t}\hat{z}^*\right) - \left(z^* - \mathcal{B}^{\hat{\pi}_t}z^*\right)\|_{\mathcal{D}_t}^2 \\
&\leq 2\left(J(\hat{z}^*) - J(z^*)\right)
\end{aligned}
$$

where $(i)$ follows from (Nachum et al., 2019, Section D.1) $\epsilon_{approx}(\mathcal{F})$ is the error due to the approximation with $\mathcal{F}$ for $z$, $\epsilon_{approx}(\mathcal{H})$ is the error due to the approximation with $\mathcal{H}$ for $\zeta$, and $\epsilon_{est}$ is the estimation error, and $L$ is the Lipschitz constant for $f$.

To bound the error between $J(\hat{z}^*)$ and $J(z^*)$, we use the decomposition suggested in (Nachum et al., 2019):

$$J(\hat{z}^*) - J(z^*) = \underbrace{J(\hat{z}^*) - L(\hat{z}^*)}_{(i)} + \underbrace{L(\hat{z}^*) - L(z_{\mathcal{F}}^*)}_{(ii)} + \underbrace{L(z_{\mathcal{F}}^*) - J(z_{\mathcal{F}}^*)}_{(iii)} + \underbrace{J(z_{\mathcal{F}}^*) - J(z^*)}_{(iv)}, \quad (41)$$

where $(i) \leq \frac{2C_\omega}{1-\gamma}\|\zeta_{\mathcal{H}}^* - \zeta^*\|_{\mathcal{D}_t, 1} \leq \frac{2C_\omega}{1-\gamma}\epsilon_{approx}(\mathcal{H})$, $(ii) = \mathcal{O}\left(\frac{\sqrt{\log M + \log \frac{1}{\delta}}}{M}\right)$ by (Nachum et al., 2019, Lem. 6) (by also plugging in $N = \mathcal{O}(M^2)$), $(iii) \leq 0$ by definition, and $(iv) = \mathcal{O}(\epsilon_{approx}(\mathcal{F}))$. Note that we refer the reader to (Nachum et al., 2019, Sec. D.1) for the above bounds. Therefore, we can bound $J(\hat{z}^*) - J(z^*)$ on the order of $\mathcal{O}\left(\epsilon_{approx}(\mathcal{H}) + \epsilon_{approx}(\mathcal{F}) + \frac{\sqrt{\log M + \log \frac{1}{\delta}}}{M}\right)$.

Combining the above relations, while noting that $\frac{1}{\sqrt{M}}$ decreases slower than $\frac{1}{M}$ in terms of $M$ and is thus kept as the upper bound, we have shown the result. $\qquad \square$

### F.7 PUTTING IT TOGETHER: BOUNDING THE KL DIVERGENCE ESTIMATION ERROR

**Theorem F.1** (KL divergence estimation error bound). *The following bound holds:*

$$
\begin{aligned}
&\left|\mathbb{E}_{\nu_t^*}[D_{KL}(\pi_t^*|\pi)] - \mathbb{E}_{\hat{\nu}_t}[D_{KL}(\hat{\pi}_t|\pi)]\right| \\
&= \mathcal{O}\left(h\left(\frac{1}{\sqrt{M}}\right) + \frac{1}{\sqrt{M}} + \sqrt{\epsilon_{opt}} + \sqrt{\epsilon_{approx}(\mathcal{F}, \mathcal{H})}\right),
\end{aligned}
$$

*where $h$ is a strictly increasing continuous function with the property that $h(0) = 0$ as specified in Lemma 1, $\epsilon_{approx}(\mathcal{F}, \mathcal{H})$ is defined in 38, and $\epsilon_{opt}$ is defined in 39.*

*Proof.* The result follows by combining the upper bounds for the error terms $(A)$, $(B)$ and $(C)$, as specified by Lemmas 16, 18, and 17. We also apply the elementary inequality $\sqrt{a + b + c} \leq \sqrt{a} + \sqrt{b} + \sqrt{c}$ to further simplify the bound. $\qquad \square$

**Remark 5.** *The bound above depends on the number of iterations $M$ per task in different ways. By increasing $M$, we can expect to reduce the suboptimality gap, which can help reduce the distance between $\hat{\pi}_t$ to the optimal set of policies. Also, increasing $M$ results in a larger dataset used to estimate its stationary distribution offline by DualDICE, which reduces the estimation error. The bound indicates that the only terms that do not vanish as we increase the number of iterations per task are those due to the inherent optimization error $\epsilon_{opt}$ and function approximation error $\epsilon_{approx}(\mathcal{F}, \mathcal{H})$. In the case those terms are negligible (which are possible in view of the recent breakthrough in overparametrized deep learning (Zhang et al., 2021; Neyshabur et al., 2019; Li & Liang, 2018; Zou et al., 2018; Arora et al., 2019), see also (Fan et al., 2021) for a survey), then the KL divergence estimation can be driven to arbitrary accuracy.*

# G    PROOFS FOR THE SECTION 3.3

## G.1    TAOG AND TACV BOUNDS FOR CRPO WITH ADAPTIVE LEARNING RATES

This section presents the task-averaged regret upper bounds for the CRPO when the adaptive learning rates $\alpha_t$ are used for each task, and the Q-estimation error is accounted for. We also recall that $d_{t,i}$ is the constraint upper bound for $i = 1, ..., p$ and $\eta_t$ is the tolerance for constraint violation (i.e., increasing the upper bound to $d_{t,i} + \eta_t$). For a single run of CRPO in task $t$, we denote $\mathcal{N}_{t,0}$ as the set of time steps the reward is maximized and $\mathcal{N}_{t,i}$ as the set of time step constraint $i$ is minimized. The Q-function in the CRPO algorithm is learned through TD learning with the total number of iterations denoted by $K_{in}$. The Q-function of objective $i$ for policy $\pi_{t,m}$ at time step $m$ is denoted by $Q_{t,m}^i$, and the estimated Q-function is denoted by $\bar{Q}_{t,m}^i$. The maximum value for both rewards and constraints is assumed to be $c_{max}$.

With all notations for CRPO in place, we present the following result (Xu et al., 2021)

**Lemma 19.** *For the CRPO algorithm in the tabular settings with learning rates $\alpha_t$, the following bound holds:*

$$\alpha_t \sum_{m \in \mathcal{N}_{t,0}} \left( J_{t,0}(\pi_t^*) - J_{t,0}(\pi_{t,m}) \right) + \eta_t \alpha_t \sum_{i=1}^{p} |\mathcal{N}_{t,i}|$$

$$\leq \mathbb{E}_{s \sim \nu_t^*}[D_{KL}(\pi_t^* | \pi_{t,0})] + \frac{2c_{max}^2 |\mathcal{S}||\mathcal{A}|}{(1-\gamma)^3} \alpha_t^2 \sum_{i=0}^{p} |\mathcal{N}_{t,i}|$$

$$+ \sum_{i=0}^{p} \sum_{m \in \mathcal{N}_{t,i}} \frac{\alpha_t(3 + (1-\gamma)^2 + 3\alpha_t c_{max})}{(1-\gamma)^2} \|Q_{t,m}^i - \bar{Q}_{t,m}^i\|_2$$

*Proof.* If $m \in \mathcal{N}_{t,0}$, by (Xu et al., 2021, Lemma 7), we have that:

$$\alpha_t \left( J_{t,0}(\pi_t^*) - J_{t,0}(\pi_{t,m}) \right) \leq \mathbb{E}_{\nu_t^*}[D_{KL}(\pi_t^* | \pi_{t,m}) - D_{KL}(\pi_t^* | \pi_{t,m+1})] + \frac{2c_{max}^2 |\mathcal{S}||\mathcal{A}|}{(1-\gamma)^3} \alpha_t^2$$

$$+ \frac{3\alpha_t(1 + \alpha_t c_{max})}{(1-\gamma)^2} \|Q_{t,m}^0 - \bar{Q}_{t,m}^0\|_2. \tag{42}$$

Similarly, if $m \in \mathcal{N}_{t,i}$, we can write

$$\alpha_t \left( J_{t,i}(\pi_{t,m}) - J_{t,i}(\pi_t^*) \right) \leq \mathbb{E}_{\nu_t^*}[D_{KL}(\pi_t^* | \pi_{t,m}) - D_{KL}(\pi_t^* | \pi_{t,m+1})] + \frac{2c_{max}^2 |\mathcal{S}||\mathcal{A}|}{(1-\gamma)^3} \alpha_t^2$$

$$+ \frac{3\alpha_t(1 + \alpha_t c_{max})}{(1-\gamma)^2} \|Q_{t,m}^i - \bar{Q}_{t,m}^i\|_2. \tag{43}$$

Taking the summation of 42 and 43 from $m = 0$ to $M - 1$, we get

$$\alpha_t \sum_{m \in \mathcal{N}_{t,0}} \left( J_{t,0}(\pi_t^*) - J_{t,0}(\pi_{t,m}) \right) + \sum_{i=1}^{p} \sum_{m \in \mathcal{N}_{t,i}} \alpha_t \left( J_{t,i}(\pi_{t,m}) - J_{t,i}(\pi_t^*) \right)$$

$$\leq \mathbb{E}_{\nu_t^*}[D_{KL}(\pi_t^* | \pi_{t,0})] + \frac{2c_{max}^2 |\mathcal{S}||\mathcal{A}|}{(1-\gamma)^3} \sum_{i=0}^{p} \alpha_t^2 |\mathcal{N}_{t,i}|$$

$$+ \sum_{i=0}^{p} \sum_{m \in \mathcal{N}_{t,i}} \frac{3\alpha_t(1 + \alpha_t c_{max})}{(1-\gamma)^2} \|Q_{t,m}^i - \bar{Q}_{t,m}^i\|_2$$

Since $J_{t,i}(\pi_{t,m}) - J_{t,i}(\pi_t^*) \geq \eta_t - \|Q_{t,m}^i - \bar{Q}_{t,m}^i\|$ (Xu et al., 2021, Eq. 15), by rearranging the terms above, we obtain the result.                          □

Next, we study the condition on the maximum constraint violation threshold $\eta_t$ and how it affects $\mathcal{N}_{t,0}$ and the upper bounds for TAOG and TACV. We make the following assumption to proceed.

**Assumption 4.** *Assume that $\sum_{m \in \mathcal{N}_{t,0}} J_{t,0}(\pi_t^*) - J_{t,0}(\pi_{t,m}) \geq c_J$ for some $c_J \in (-\frac{1}{2}\alpha_t \eta_t M, 0]$.*

The assumption above indicates that the policies in $\mathcal{N}_{t,0}$ do not have rewards higher than the optimal policy by more than $\frac{1}{2}\alpha_t\eta_t$ on average. Note that it is indeed possible to have rewards higher than the optimal policy if the corresponding policy does not satisfy some safety constraints (i.e., infeasible policy). However, it is not a strong assumption since we are comparing it with the optimal policy. The above assumption is not present in (Xu et al., 2021), which invalidates one of its derivation steps (in particular, (Xu et al., 2021, Thm. 3)), and is thus introduced to rectify the proof.

**Lemma 20.** *Suppose that the following condition holds:*

$$\frac{1}{2}\eta_t M \alpha_t \geq \mathbb{E}_{\nu_t^*}[D_{KL}(\pi_t^* | \pi_{t,0})] + \frac{2c_{max}^2 |\mathcal{S}||\mathcal{A}| M}{(1-\gamma)^3} \sum_{i=0}^{p} \alpha_t^2$$

$$+ \sum_{i=0}^{p} \sum_{m \in \mathcal{N}_{t,i}} \frac{\alpha_t(3 + (1-\gamma)^2 + 3\alpha_t c_{max})}{(1-\gamma)^2} \|Q_{t,m}^i - \bar{Q}_{t,m}^i\|_2. \tag{44}$$

*Then, we have that $\mathcal{N}_{t,0} \neq \emptyset$, i.e., $\hat{\pi}_t$ is well-defined; also, one the following two statements must hold,*

1. *$|\mathcal{N}_{t,0}| \geq M/2$,*

2. *$\sum_{m \in \mathcal{N}_{t,0}} \left( J_{t,0}(\pi_t^*) - J_{t,0}(\pi_{t,m}) \right) \leq 0$.*

*Under assumption 4, we also have the following holds:*

$$|\mathcal{N}_{t,0}| \geq \left( \frac{1}{2} - \kappa \right) M$$

*for some $\kappa \in (0, \frac{1}{2})$.*

*Proof.* The proof for $\mathcal{N}_{t,0} \neq \emptyset$ follows directly from (Xu et al., 2021, Lem. 9). For the second statement, we consider the case that $\sum_{m \in \mathcal{N}_{t,0}} \left( J_{t,0}(\pi_t^*) - J_{t,0}(\pi_{t,m}) \right) > 0$. From Lemma 19, it implies that

$$\eta_t \sum_{i=1}^{p} \alpha_t |\mathcal{N}_{t,i}| \leq \mathbb{E}_{s \sim \nu^*}[D_{KL}(\pi_t^* | \pi_{t,0})] + \frac{2c_{max}^2 |\mathcal{S}||\mathcal{A}|}{(1-\gamma)^3} \sum_{i=0}^{p} \alpha_t^2 |\mathcal{N}_{t,i}|$$

$$+ \sum_{i=0}^{p} \sum_{m \in \mathcal{N}_{t,i}} \frac{\alpha_t(3 + (1-\gamma)^2 + 3\alpha_t c_{max})}{(1-\gamma)^2} \|Q_{t,m}^i - \bar{Q}_{t,m}^i\|_2.$$

Suppose that $|\mathcal{N}_{t,0}| < M/2$, then $\sum_{i=1}^{p} |\mathcal{N}_{t,i}| > M/2$, we have that

$$\frac{1}{2}\alpha_t \eta_t M < \mathbb{E}_{s \sim \nu^*}[D_{KL}(\pi_t^* | \pi_{t,0})] + \frac{2c_{max}^2 |\mathcal{S}||\mathcal{A}|}{(1-\gamma)^3} \sum_{i=0}^{p} \alpha_t^2 |\mathcal{N}_{t,i}|$$

$$+ \sum_{i=0}^{p} \sum_{m \in \mathcal{N}_{t,i}} \frac{\alpha_t(3 + (1-\gamma)^2 + 3\alpha_t c_{max})}{(1-\gamma)^2} \|Q_{t,m}^i - \bar{Q}_{t,m}^i\|_2,$$

which contradicts 44. Hence, we must have $|\mathcal{N}_{t,0}| \geq M/2$.

Next, we show that $|\mathcal{N}_{t,0}| \geq \left( \frac{1}{2} - \kappa \right) M$ for some $\kappa \in (0, \frac{1}{2})$. Under assumption 4 and by Lemma 19, we have that

$$\eta_t \sum_{i=1}^{p} \alpha_t |\mathcal{N}_{t,i}| \leq \mathbb{E}_{s \sim \nu^*}[D_{KL}(\pi_t^* | \pi_{t,0})] + \frac{2c_{max}^2 |\mathcal{S}||\mathcal{A}|}{(1-\gamma)^3} \sum_{i=0}^{p} \alpha_t^2 |\mathcal{N}_{t,i}|$$

$$+ \sum_{i=0}^{p} \sum_{m \in \mathcal{N}_{t,i}} \frac{\alpha_t(3 + (1-\gamma)^2 + 3\alpha_t c_{max})}{(1-\gamma)^2} \|Q_{t,m}^i - \bar{Q}_{t,m}^i\|_2 - \alpha_t c_J.$$

Choose $\kappa := \frac{-c_J}{\alpha_t \eta_t M}$. Since $-c_J < \frac{1}{2}\alpha_t \eta_t M$ by assumption, we have that $\kappa \in (0, \frac{1}{2})$. Consider the case that $|\mathcal{N}_{t,0}| < (\frac{1}{2} - \kappa)M$, which implies that $\sum_{i=1}^p |\mathcal{N}_{t,i}| > (\frac{1}{2} + \kappa)M$. This implies that

$$
\left(\frac{1}{2} + \kappa\right)\alpha_t \eta_t M \leq \mathbb{E}_{s \sim \nu^*}[D_{KL}(\pi_t^*|\pi_{t,0})] + \frac{2c_{max}^2|\mathcal{S}||\mathcal{A}|}{(1-\gamma)^3} \sum_{i=0}^p \alpha_t^2 |\mathcal{N}_{t,i}|
$$

$$
+ \sum_{i=0}^p \sum_{m \in \mathcal{N}_{t,i}} \frac{\alpha_t(3 + (1-\gamma)^2 + 3\alpha_t c_{max})}{(1-\gamma)^2} \|Q_{t,m}^i - \bar{Q}_{t,m}^i\|_2,
$$

which again contradicts 44. Hence, we must have $|\mathcal{N}_{t,0}| \geq (\frac{1}{2} - \kappa)M$. □

Now, we prove the upper bound of suboptimality and constraint violation per task.

**Lemma 21.** *Let the violation tolerance be chosen as:*

$$
\eta_t = \frac{2\mathbb{E}_{s \sim \nu_t^*}[D_{KL}(\pi_t^*|\pi_{t,0})]}{M\alpha_t} + \frac{\alpha_t 4c_{max}^2|\mathcal{S}||\mathcal{A}|}{(1-\gamma)^3} + \sum_{i=0}^p \frac{2\alpha_t(3 + (1-\gamma)^2 + 3\alpha_t c_{max})}{\sqrt{M}\alpha_t(1-\gamma)^2},
$$

*Then, the following holds*

$$
U_{t,0}(\pi_{t,0}, \alpha_t) = \frac{c_1^t}{\alpha_t M}\mathbb{E}_{s \sim \nu_t^*}[D_{KL}(\pi_t^*|\pi_{t,0})] + c_2^t\alpha_t + \sum_{i=0}^p \frac{c_3^t\alpha_t + c_4^t\alpha_t^2}{\alpha_t\sqrt{M}} \tag{45}
$$

$$
U_{t,i}(\pi_{t,0}, \alpha_t) = U_{t,0}(\pi_{t,0}, \alpha_t) + \frac{c_5^t}{\sqrt{M}}, \tag{46}
$$

*where* $c_1^t = 2$, $c_2^t = \frac{4c_{max}^2|\mathcal{S}||\mathcal{A}|}{(1-\gamma)^3}$, $c_3^t = \frac{3 + (1-\gamma)^2}{(1-\gamma)^2}$, $c_4^t = \frac{3c_{max}}{(1-\gamma)^2}$, *and* $c_5^t = \frac{2\sqrt{(1-\gamma)|\mathcal{S}||\mathcal{A}|}}{1-2\kappa}$.

*Proof.* From Lemma 19, we have that

$$
\alpha_t \sum_{m \in \mathcal{N}_{t,0}} \left(J_{t,0}(\pi_t^*) - J_{t,0}(\pi_{t,m})\right) + \eta_t \sum_{i=1}^p \alpha_t |\mathcal{N}_{t,i}|
$$

$$
\leq \mathbb{E}_{s \sim \nu_t^*}[D_{KL}(\pi_t^*|\pi_{t,0})] + \frac{2c_{max}^2|\mathcal{S}||\mathcal{A}|}{(1-\gamma)^3} \sum_{i=0}^p \alpha_t^2 |\mathcal{N}_{t,i}|
$$

$$
+ \sum_{i=0}^p \sum_{m \in \mathcal{N}_{t,i}} \frac{\alpha_t(3 + (1-\gamma)^2 + 3\alpha_t c_{max})}{(1-\gamma)^2} \|Q_{t,\pi_m}^i - \bar{Q}_{t,\omega_m}^i\|_2
$$

If $\sum_{m \in \mathcal{N}_{t,0}} \left(J_{t,0}(\pi_t^*) - J_{t,0}(\pi_{t,m})\right) \leq 0$, then $J_{t,0}(\pi_t^*) - \mathbb{E}[J_{t,0}(\hat{\pi}_t)] \leq 0$. If $\sum_{m \in \mathcal{N}_{t,0}} \left(J_{t,0}(\pi_t^*) - J_{t,0}(\pi_{t,m})\right) > 0$, then, by Lemma 20, we have $|\mathcal{N}_{t,0}| \geq M/2$. Hence,

$$
J_{t,0}(\pi_t^*) - \mathbb{E}[J_{t,0}(\hat{\pi}_t)] = \frac{1}{|\mathcal{N}_{t,0}|} \sum_{m \in \mathcal{N}_{t,0}} [J_{t,0}(\pi_t^*) - J_{t,0}(\pi_{t,m})]
$$

$$
\leq \frac{2}{\alpha_t M}\mathbb{E}_{\nu_t^*}[D_{KL}(\pi_t^*|\pi_{t,0})] + c_2^t\alpha_t + \sum_{i=0}^p \sum_{m \in \mathcal{N}_{t,i}} \frac{(c_3^t\alpha_t + c_4^t\alpha_t^2)}{M\alpha_t} \|Q_{t,\pi_m}^i - \bar{Q}_{t,\omega_m}^i\|_2,
$$

$$
\leq \frac{2}{\alpha_t M}\mathbb{E}_{\nu_t^*}[D_{KL}(\pi_t^*|\pi_{t,0})] + c_2^t\alpha_t + \sum_{i=0}^p \sum_{m \in \mathcal{N}_{t,i}} \frac{(c_3^t\alpha_t + c_4^t\alpha_t^2)}{\sqrt{M}\alpha_t}
$$

where the last inequality is due to the choice of $K_{in} = \Theta(M^{1/\sigma}\log^{2/\sigma}(|\mathcal{S}|^2|\mathcal{A}|^2 M^{1+2/\sigma}/\delta))$ as specified by (Xu et al., 2021, Lem. 8) for critic evaluations. Here, the constants are chosen as $c_1^t = 2$,

$c_2^t = \frac{4c_{max}^2|\mathcal{S}||\mathcal{A}|}{(1-\gamma)^3}$, $c_3^t = \frac{3+(1-\gamma)^2}{(1-\gamma)^2}$, $c_4^t = \frac{3c_{max}}{(1-\gamma)^2}$. For constraint violation, consider any $i = 1, ..., p$, we have

$$
\begin{aligned}
\mathbb{E}[J_{t,i}(\hat{\pi}_t)] - d_{t,i} &= \frac{1}{|\mathcal{N}_{t,0}|} \sum_{m \in \mathcal{N}_{t,0}} J_{t,i}(\pi_{t,m}) - d_{t,i} \\
&\le \frac{1}{|\mathcal{N}_{t,0}|} \sum_{m \in \mathcal{N}_{t,0}} \left( \bar{J}_{t,i}(\pi_{t,m}) - d_{t,i} \right) + \frac{1}{|\mathcal{N}_{t,0}|} \sum_{m \in \mathcal{N}_{t,0}} |\bar{J}_{t,i}(\pi_{t,m}) - J_{t,i}(\pi_{t,m})| \\
&\le \eta_t + \frac{1}{|\mathcal{N}_{t,0}|} \sum_{i=0}^{p} \sum_{m \in \mathcal{N}_{t,i}} \|Q_{t,\pi_m}^i - \bar{Q}_{t,\pi_m}^i\|_2 \\
&\le \eta_t + \frac{2}{(1-2\kappa)M} \sum_{i=0}^{p} \sum_{m \in \mathcal{N}_{t,i}} \|Q_{t,\pi_m}^i - \bar{Q}_{t,\pi_m}^i\|_2
\end{aligned}
$$

where the first inequality is due to triangle inequality, the second inequality is by the design of the CRPO algorithm, and the third inequality is due to Lemma 20, where $\kappa \in (0, \frac{1}{2})$. By the choice of $K_{in}$, we have that $\sum_{i=0}^{p} \sum_{m \in \mathcal{N}_{t,i}} \|Q_{t,\pi_m}^i - \bar{Q}_{t,\pi_m}^i\|_2 \le \sqrt{(1-\gamma)|\mathcal{S}||\mathcal{A}|M}$. Plugging the value of $\eta_t$ yields the desired result. $\qquad \square$

Finally, we are able to provide the following bounds on TAOG and TACV in the case of adaptive learning rates.

**Theorem G.1** (Bounds on TAOG and TACV). *Suppose we run the CRPO algorithm for $M$ steps per task $t$ with learning rates $\alpha_t$. Then, after $T$ tasks, the TAOG $\bar{R}_0$ is given by*

$$
\bar{R}_0 = \frac{1}{T} \sum_{t=1}^{T} \left[ \frac{c_1^t}{\alpha_t M} \mathbb{E}_{s \sim \nu_t^*}[D_{KL}(\pi_t^*|\pi_{t,0})] + c_2^t \alpha_t + \sum_{i=0}^{p} \frac{c_3^t \alpha_t + c_4^t \alpha_t^2}{\alpha_t \sqrt{M}} \right], \tag{47}
$$

*and the TACV $\bar{R}_i$ is given by*

$$
\bar{R}_i = \frac{1}{T} \sum_{t=1}^{T} \left[ \frac{c_1^t}{\alpha_t M} \mathbb{E}_{s \sim \nu_t^*}[D_{KL}(\pi_t^*|\pi_{t,0})] + c_2^t \alpha_t + \sum_{i=0}^{p} \frac{c_3^t \alpha_t + c_4^t \alpha_t^2}{\alpha_t \sqrt{M}} + \frac{c_5^t}{\sqrt{M}} \right], \tag{48}
$$

*for $i = 1, ..., p$, where $c_1^t = 2$, $c_2^t = \frac{4c_{max}^2|\mathcal{S}||\mathcal{A}|}{(1-\gamma)^3}$, $c_3^t = \frac{3+(1-\gamma)^2}{(1-\gamma)^2}$, $c_4^t = \frac{3c_{max}}{(1-\gamma)^2}$, and $c_5^t = \frac{2\sqrt{(1-\gamma)|\mathcal{S}||\mathcal{A}|}}{1-2\kappa}$.*

*Proof.* The proof follows directly by summing the results 45 and 46 over $t = 1, ..., T$. $\qquad \square$

## G.2 ADAPTING TO THE DYNAMIC REGRET USING ADAPTIVE LEARNING RATES

We restate the theorem below for convenience.

**Theorem G.2.** *Let each within-task CMDP $t$ run $M$ steps of CRPO, initialized by policy $\pi_{t,0} := $ INIT$(t)$ and learning rates $\{\alpha_t\}_{i=0}^{p} := $ SIM$(t)$. Let $\kappa^* := \arg \min L(\kappa)$, where*

$$
L(\kappa) = U_T^{sim}(\kappa) + \frac{U_T^{init}(\{\psi_t\}_{t=1}^{T})}{\kappa} + \frac{\mathcal{E}_T}{\kappa} + \sum_{t=1}^{T} \left[ \frac{\hat{f}_t^{init}(\phi_t)}{\kappa} + f_t^{rate}(\kappa) \right], \tag{49}
$$

*and $\{\psi_t^*\}_{t=1}^{T}$ is any comparator sequence. Then, the following bounds on TAOG and TACV hold:*

$$
\bar{R}_i \le \frac{L(\kappa^*)}{T}, \qquad \forall i = 0, ..., p. \tag{50}
$$

*Proof.* The idea of the proof is to freeze the learning rates first to obtain a dynamic regret bound based on policy initialization and then optimize over the learning rates to obtain a tighter characterization.

Also, since TAOG and TACV only differ by a bias term that does not depend on either the learning rates or the initial policy, we can treat them indistinguishably. In particular,

$$\sum_{t=1}^{T} U_t(\pi_{t,0}, \alpha_t) = \sum_{t=1}^{T} f_t^{sim}(\alpha_t) \tag{51}$$

$$\leq \min_{\kappa} \; U_T^{sim}(\kappa) + \sum_{t=1}^{T} f_t^{sim}(\kappa) \tag{52}$$

$$\leq \min_{\kappa} \; U_T^{sim}(\kappa) + \frac{U_T^{init}(\Psi)}{\kappa} + \sum_{t=1}^{T} \left[ \frac{f_t^{init}(\psi_t)}{\kappa} + f_t^{rate}(\kappa) \right] \tag{53}$$

$$\leq \min_{\kappa} \; U_T^{sim}(\kappa) + \frac{U_T^{init}(\Psi)}{\kappa} + \frac{\mathcal{E}_T}{\kappa} + \sum_{t=1}^{T} \left[ \frac{\hat{f}_t^{init}(\phi_t)}{\kappa} + f_t^{rate}(\kappa) \right]. \tag{54}$$

where $\Psi := \{\phi_t\}_{t=1}^{T}$. Let

$$L(\kappa) = U_T^{sim}(\kappa) + \frac{U_T^{init}(\Psi)}{\kappa} + \frac{\mathcal{E}_T}{\kappa} + \sum_{t=1}^{T} \left[ \frac{\hat{f}_t^{init}(\phi_t)}{\kappa} + f_t^{rate}(\kappa) \right] \tag{55}$$

and define

$$\kappa^* = \arg\min L(\kappa). \tag{56}$$

Thus, plugging $\kappa = \kappa^*$ in 54 results in 50. $\qquad\square$

Now, we provide the regret upper bound for $U_T^{sim}(\kappa)$, when $\mathrm{SIM}(t)$ is OGD over the sequence $\hat{f}_t^{sim}(\kappa)$.

**Corollary 2.** *For any fixed comparator $\alpha^* = \arg\min_{\kappa} \sum_{t=1}^{T} \hat{f}_t^{sim}$, if $\mathrm{SIM}(t)$ is OGD which is run on a sequence of loss functions $\{\hat{f}_t^{sim}\}_{t\in[T]}$ with the step-size $\frac{\alpha^*}{K_\alpha \sqrt{2T}}$, then the following bound holds for static regret:*

$$\sum_{t=1}^{T} \hat{f}_t^{sim}(\kappa) - \sum_{t=1}^{T} \hat{f}_t^{sim}(\alpha^*) \leq \sqrt{2T} K_\alpha |\alpha^*| + \left( 1 + \frac{4\sqrt{2} K_\alpha L_\alpha |\alpha^*|}{(2C_1 - C_1^2 L_\alpha)\sqrt{T}} \right) \mathcal{E}_T,$$

*for any $C_1 \in \{ c \in \left( 0, \frac{2}{L_\alpha} \right) : \alpha^* + c(u - \hat{\nabla}_t) \in \Lambda \}$ where $u$ is an $\epsilon$-subgradient of $f_t^{sim}(\kappa)$ with respect to $\kappa$, $\mathcal{E}_T := \sum_{t=1}^{T} \epsilon_t$ is the cumulative inexactness, $\epsilon_t$ is the upper bound from Theorem 3.1.*

*Proof.* The proof follows directly after substituting $c = \frac{4}{2C_1 - C_1^2 L_\alpha}$ and other appropriate constants in Theorem E.1. $\qquad\square$

Next, we provide the proof of Corollary 1 TAOG and TACV regret bounds when INIT and SIM are either FTL or inexact OGD.

**Corollary 3.** *For any fixed comparator $\alpha^* = \arg\min_{\kappa} \sum_{t=1}^{T} \hat{f}_t^{sim}$ and $c_3^t = \frac{3+(1-\gamma)^2}{(1-\gamma)^2}$. If $\mathrm{INIT}(t)$ and $\mathrm{SIM}(t)$ are FTL or inexact OGD over the sequences $\hat{f}_t^{init}$ and $\hat{f}_t^{sim}$, then, the following bounds on TAOG and TACV hold:*

$$\bar{R}_i \leq \frac{1}{\sqrt{M}} \left( \frac{\sqrt{2} K_\alpha |\alpha^*|}{\sqrt{MT}} + \left( c_3^t + \frac{4\sqrt{2} K_\alpha L_\alpha |\alpha^*|}{(2C_1 - C_1^2 L_\alpha)\sqrt{T}} \right) \frac{\mathcal{E}_T}{\sqrt{MT}} + \frac{1}{\sqrt{c_2^t}} \frac{\sqrt{U_T^{init}(\{\psi_t^*\}_{t\in[T]}) + \mathcal{E}_T + T\hat{V}_\psi^2}}{M^{1/4}T} \right), \tag{57}$$

*for all $i = 0, \ldots, p$, where $\{\psi_t^*\}_{t\in[T]}$ is any comparator sequence, $c_3^t = \frac{3+(1-\gamma)^2}{(1-\gamma)^2}$, and $c_2^t = \frac{4c_{max}^2 |\mathcal{S}||\mathcal{A}|}{(1-\gamma)^3}$.*

*Proof.* Firstly, the following inexact upper bounds hold for $U_T^{init}$ and $U_T^{sim}$:

$$U_T^{init}(\{\psi_t^*\}_{t \in T}) = \min \left( C_1 \|\phi_1 - \psi_1^*\|^2 + C_2 \mathcal{E}_T + C_3 \mathcal{S}_T + \frac{1}{2\beta} \sum_{t=1}^{T} \|\nabla \ell_t(\psi_t^*)\|^2, \right.$$
$$\left. C_4 \|\phi_1 - \psi_1^*\| + C_5 \tilde{\mathcal{E}}_T + C_4 \mathcal{P}_T \right), \tag{58}$$

$$U_T^{sim}(\kappa) = \sqrt{2T} K_\alpha |\alpha^*| + \left( 1 + \frac{4\sqrt{2} K_\alpha L_\alpha |\alpha^*|}{(2C_1 - C_1^2 L_\alpha)\sqrt{T}} \right) \mathcal{E}_T, \tag{59}$$

where the constants in the 58 and 59 are from the Lemma 11 and Corollary 2 respectively.

Plugging these in the equation 49, we can obtain the following:

$$L(\kappa) \leq \sqrt{2T} K_\alpha |\alpha^*| + \left( 1 + \frac{4\sqrt{2} K_\alpha L_\alpha |\alpha^*|}{(2C_1 - C_1^2 L_\alpha)\sqrt{T}} \right) \mathcal{E}_T + \frac{1}{\kappa} \min \left( C_1 \|\phi_1 - \psi_1^*\|^2 \right.$$
$$+ C_2 \mathcal{E}_T + C_3 \mathcal{S}_T + \frac{1}{2\beta} \sum_{t=1}^{T} \|\nabla \ell_t(\psi_t^*)\|^2, C_4 \|\phi_1 - \psi_1^*\| + C_5 \tilde{\mathcal{E}}_T + C_4 \mathcal{P}_T \right)$$
$$+ \frac{\mathcal{E}_T}{\kappa} + \frac{T \hat{V}_\psi^2}{\kappa} + f_t^{rate}(\kappa),$$

where the relation $\hat{V}_\psi^2 = \frac{1}{T} \sum_{t=1}^{T} \mathbb{E}_{\hat{\nu}_t}[D_{KL}(\hat{\pi}_t | \psi_t^*)]$ is used to get the second last term $\frac{T \hat{V}_\psi^2}{\kappa}$. To get $\kappa^*$ such that $L(\kappa)$ is minimized, we proceed using the KKT optimality conditions to get $\frac{dL}{d\kappa}$ as

$$\frac{dL}{d\kappa} = \frac{-U_T^{init}(\{\psi\}_{t \in [T]})}{\kappa^2} - \frac{\mathcal{E}_T}{\kappa^2} - \frac{T \hat{V}_\psi}{\kappa^2} + c_2^t M + c_4^t \sqrt{M}.$$

Therefore, we obtain $\kappa^* = \sqrt{\frac{U_T^{init}(\psi) + \mathcal{E}_T + T \hat{V}_\psi^2}{c_2^t M + c_4^t \sqrt{M}}}$.

Substituting this value of $\kappa^*$ in G.2, and using the approximation $\sqrt{\frac{1}{M + \sqrt{M}}} \approxeq \frac{1}{M^{1/4}}$, we can obtain the final result. $\qquad \square$

## H ADDITIONAL EXPERIMENTS AND DETAILS

In this section, we describe our experimental setup and other additional details for the OpenAI gym experiments under constrained settings. We present the performance of Meta-SRL on the Acrobot and the Frozen lake under high task-similarity conditions. We first present some details on the baselines used to avoid any confusion.

**Details of baselines used:** Denote the policy initialization for task $t$ as $\phi_t$. First baseline is the FAL, which initializes the policy $\phi_{t+1}$ for the test task $t+1$ as $\phi_{t+1} = \frac{1}{t} \sum_{i=1}^{t} \phi_i$. This is done online as the tasks are encountered sequentially.

For the simple averaging baseline, we run CRPO on 10 tasks with the random initialization and evaluate the performance on a test task by initializing with the average of all the suboptimal policies from the batch of suboptimal policies, i.e., $\phi_T = \frac{1}{T-1} \sum_{i=1}^{T-1} \phi_i$, where $T$ is the test task. This can be seen as an offline method, where suboptimal policies from $T-1$ tasks are stored and averaged to get the initialization for the test task.

The pre-trained baseline uses suboptimal policy from an already trained task to initialize the policy on the test task, i.e., $\phi_t = \hat{\pi}_{t'}$, where $\phi_t$ is the policy initialization for the task $t$, and $\hat{\pi}_{t'}$ is the suboptimal policy returned some pre-trained task $t'$. This baseline could be thought of as the Strawman initialization strategy, where the policy for the next task is initialized using the suboptimal policy from the previous task, i.e., $\phi_{t+1} = \hat{\pi}_t$.

For the Meta-SRL in the discrete state action space (i.e., Frozen lake), we take the weighted average of the previous suboptimal policies weighted by the stationary distributions induced by each suboptimal policy over all the states. We also adapt the learning rates $\alpha_t$ for the CRPO algorithm for each task $t$. The difference between FAL and Meta-SRL is that Meta-SRL weights the suboptimal policies higher in the states which were encountered more frequently.

Random initialization baseline is using random iniitalization for the within-task algorithm CRPO.

**Experimental setup:** We run all the algorithms online where tasks are encountered sequentially and present the results for the test task after the policy initialization suggested by respective baselines. We do 10 runs for each baseline on the test task and present the variance plots. On the test task, we train for 8 steps on the Frozen lake and for 5 steps on the Acrobot. In Frozen lake, each step corresponds to 5 episodes, and the average rewards/costs are reported for each step in the performance and constraint violation plots.

## H.1 FROZEN LAKE

**Frozen lake:** For the Frozen lake, we randomly generate $T = 10$ different orientations as tasks over the probability of a state being frozen or a hole and evaluate the performance for the scenarios with high task-similarity (low variance for the latent CMDP distribution) or low task-similarity (high variance for the latent CMDP distribution). The agent gets rewarded $+2$ when it reaches the goal state and incurs a cost $-1$ when it falls into a hole. We choose the constraint threshold $d_{t,i} = 0.3$.

**High task-similarity:** To generate tasks with high task-similarity, we start with a random frozen lake grid of $4 \times 4$, where the probability of each grid being frozen is $0.7$. Then, we generate 9 different grids which differ from the first one by only one of the grids. This means the agent always encounters the new grid, which is very similar to the previous task. From Figure 4, we can observe that baselines pre-trained and FAL are competitive with the Meta-SRL in terms of reward maximization and almost zero constraint violations. The good performance of the pre-trained baseline can be explained using the fact that the the test task and the training task are very similar; the policy initialization will be close to the optimal policy of the test task.

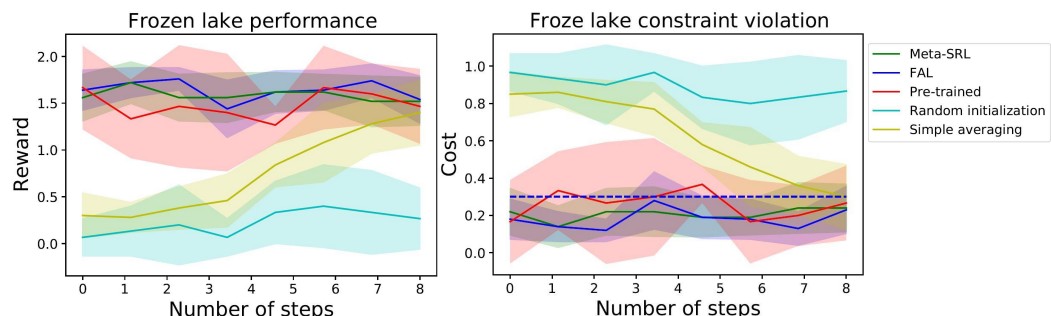

Figure 4: Frozen lake results for reward maximization and constraint violations when the task-relatedness is high. The blue dashed line represents the averaged thresholds for the constraint violations.

**Low task-similarity:** In this case, random tasks are generated where the probability of a tile being frozen is kept between $0.3$ and $0.7$. The tasks are less similar due to the high uncertainty associated with the changing orientations.

## H.2 ACROBOT

**Acrobot:** Acrobot is a 2 link robot OpenAI gym environment with continuous state space. The agent is rewarded when it achieves a certain height of the end link. Two constraints are introduced for two links, where a $-1$ cost is incurred if any link swings in the prohibited direction. We randomly generate $T = 50$ different tasks with different mass links and centers of gravity.

**High task-similarity:** To generate tasks with high similarity for the acrobot, we considered changing the mass of the links, the center of gravity (COG), and constraint threshold for each link. The changes in these quantities were done by adding noise to the default quantities. We considered a Gaussian noise with a low variance of $0.1$ to change the tasks only slightly. From Figure 5, we can observe that only pre-trained and FAL baselines are competitive with the Meta-SRL in terms of reward maximization and almost zero constraint violations.

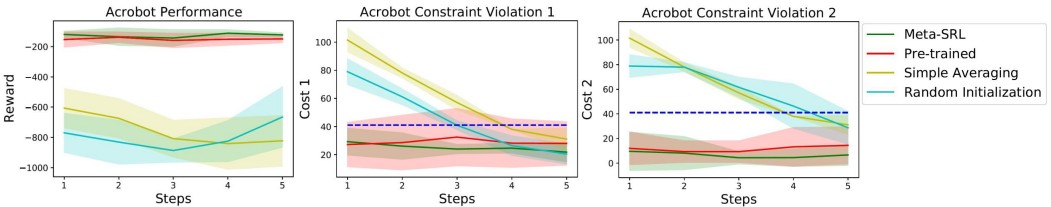

Figure 5: Acrobot results for reward maximization and constraint violations when the task-relatedness is high. The blue dashed line represents the averaged thresholds for the constraint violations.

**Low task-similarity:** To generate low similar tasks, we increased the variance of the Gaussian noise to $0.3$. We can observe from Figure 2 that the performance of the baselines was poor for constraint satisfaction, while Meta-SRL could converge quickly for reward maximization and constraint satisfaction.

Note that, in real-world settings, tasks are likely to have low similarity in terms of how close their optimal policies are. The good performance of Meta-SRL under these settings highlights its potential to be extended to real-world settings where safety constraints are present.

### H.3 HALF CHEETAH

Half-cheetah is a simulation environment for a 2-dimensional robot. It consists of 9 links and eight joints, where the goal for the cheetah is to run at a certain velocity. It has a 17-dimensional state space and a 6-dimensional action space. The reward is calculated as the negative of the absolute difference between the current cheetah velocity and the desired velocity. The original HalfCheetah environment does not have any constraints. We introduce a constraint that penalizes the deviation of the cheetah's head from some desired height:

$$h_{cheetah} - h_{target} \leq \epsilon,$$

where we specify the cumulative absolute difference between the cheetah head height and the desired height to be less than a tolerance $\epsilon$. The cheetah is trained on $T = 100$ tasks for both high and low task-similarity settings.

**High task-similarity:** To generate tasks with high similarity, the goal velocity for each training task is uniformly sampled from a range of $[0.35, 0.65]m/s$. From Figure 6, we can observe that under high task similarity settings, Meta-SRL is able to achieve high rewards and also able to keep the constraint violation of the cheetah's head height below the threshold. Under high task-similarity settings, pre-trained and Meta-SRL perform well, as expected. However, it can be observed that both simple averaging and random initialization perform poorly in this setting.

**Low task-similarity:** To generate tasks with low similarity for the half-cheetah, the goal velocity for each training task is uniformly sampled from a range of $[0.0, 1.0]m/s$. Tasks are less similar due to the high variance of goal velocities that the cheetah is trained on. It can be observed from Figure 7 that Meta-SRL is able to achieve higher rewards and zero constraint violations quickly compared to other baseline initializations under low task-relatedness settings. The pre-trained baseline can achieve higher rewards, but similar to other baselines, it cannot achieve constraint satisfaction within 10 steps. It can also be observed that both simple averaging and random initialization perform poorly in reward maximization in this setting. Close inspection indicates that there is a high variance among the policy parameters learned from each task, which may result in interference among different tasks in the relatively high dimensional state space.

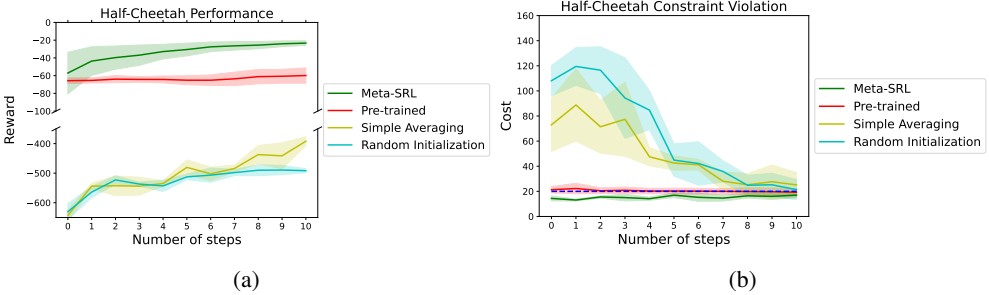

Figure 6: Half-cheetah results for reward maximization and constraint violations when the task-relatedness is high. The blue dashed line represents the averaged thresholds for the constraint violations. Here, the simple averaging and random initialization perform the worst on the test task.

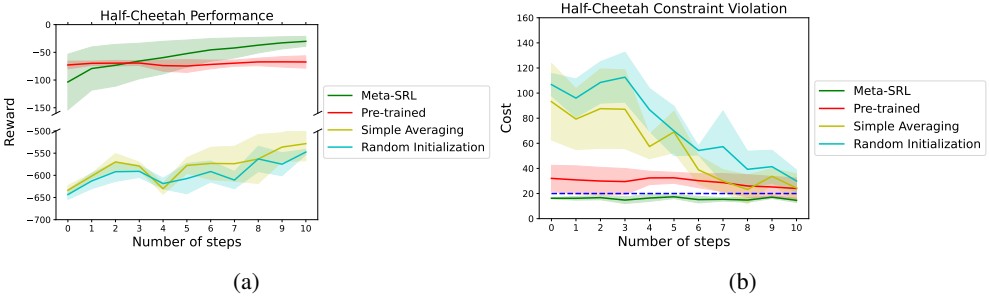

Figure 7: Half-cheetah results for reward maximization and constraint violations when the task-relatedness is low. The blue dashed line represents the averaged thresholds for the constraint violations.

## H.4 HUMANOID

Humanoid is a simulation environment of a 3D bipedal robot, which consists of a torso (abdomen) with two arms and legs. Each leg and arm has two links (representing the knees and elbows, respectively). It has a 376-dimensional observation space and a 17-dimensional action space. The goal of the humanoid is to walk forward as fast as possible without falling over.

The original humanoid environment does not have any constraints. We introduce a constraint that penalizes the deviation of the angles between the torso and the upper arm and the angle between the upper arm and the lower arm, such that humanoid motions are smooth and graceful. The cumulative constraint is given as:

$$\left|\theta_{tr} - \frac{\pi}{4}\right| + \left|\theta_{tl} - \frac{\pi}{4}\right| + \left|\theta_r - \frac{\pi}{4}\right| + \left|\theta_l - \frac{\pi}{4}\right| \le \epsilon,$$

where $\theta_{tr}, \theta_{tl}, \theta_r$, and $\theta_l$ are the angles between the torso and right arm, the angle between the torso and the left arm, the angle between both the upper and lower right arms, and the angle between both the upper and lower left arms, respectively. We specify the cumulative absolute difference between all these angles and the desired angle to be less than a tolerance $\epsilon$. The reward is calculated on the basis of the dot product between the direction and the velocity vector of the Center of Gravity of the humanoid, multiplied by the default scaling value in the humanoid environment $W_f$. the humanoid as follows:

$$r = W_f(v_y \sin\theta + v_x \cos\theta),$$

where $r$ is the instantaneous reward that accounts for the amount of forward movement by the humanoid, $v_x$, and $v_y$ are the horizontal and lateral components of the velocity, and $\theta$ is the walking direction of the humanoid. The default value of $W_f$ is 1.25. The humanoid is trained on $T = 250$ tasks for both high and low task-similarity settings.

**High task-similarity:** We generate different tasks by changing the direction of motion of the humanoid. Possible direction angles in the humanoid environment range from $-\pi/2$ to $\pi/2$ (which varies from left to right). To generate tasks with high similarity, the goal direction of the humanoid for each training task is uniformly sampled from a range of $[-\pi/4, \pi/4]$. From Figure 8, we can observe that under high task similarity settings, Meta-SRL is able to achieve high rewards and maintain the constraint violation of the humanoid's hand and torso angles below the threshold of $\epsilon = 4$. Under high task-similarity settings, both pre-trained and Meta-SRL perform well, as expected. However, it can be observed that both simple averaging and random initialization perform poorly in this setting. Moreover, the pre-trained baseline also fails to learn reward-maximizing behaviors.

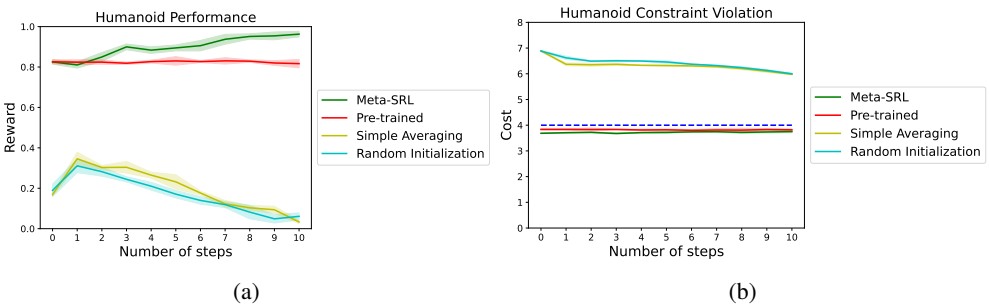

Figure 8: Humanoid results for reward maximization and constraint violations when the task-relatedness is high. The blue dashed line represents the averaged thresholds for the constraint violations.

**Low task-similarity:** To generate tasks with low similarity for the humanoid, the goal direction for each training task is uniformly sampled from a range of $[-\pi/4, \pi/4]$. Tasks are less similar due to the high variance of goal direction that the humanoid is trained on. Figure 9 shows that Meta-SRL is able to quickly achieve higher rewards and zero constraint violations compared to other baseline initializations under low task-relatedness settings. The pre-trained baseline also achieves constraint satisfaction in this case but fails to learn behaviors to maximize the rewards within 10 steps. It can also be observed that both simple averaging and random initialization perform poorly in reward maximization in this setting. This can be attributed to a high variance among the policy parameters learned from each task, which may result in interference among different tasks in the relatively high-dimensional state space.

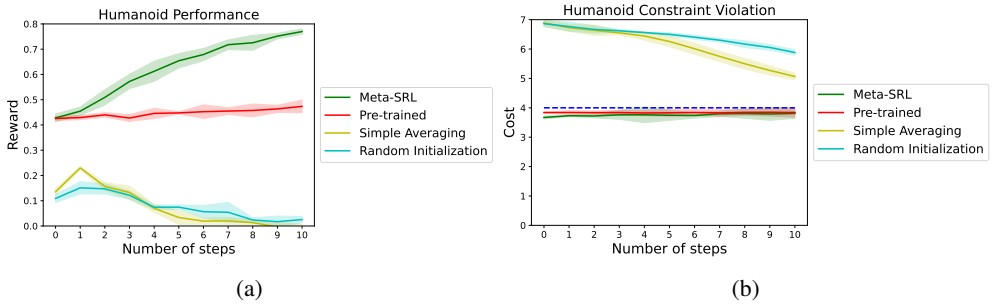

Figure 9: Humanoid results for reward maximization and constraint violations when the task-relatedness is low. The blue dashed line represents the averaged thresholds for the constraint violations.

# I  RELATION OF META-SRL TO HARDNESS RESULTS IN (KWON ET AL., 2021)

There is a key difference in the problem setting of meta-learning in our study and the latent MDP setting in (Kwon et al., 2021). The latent MDP setting is more challenging in the sense that there is no clear boundary between tasks. In the latent MDP setting, each episode may come from an unknown MDP drawn from a distribution (as a special case of POMDP); in the meta-learning setting, the agent

knows when a new task has arrived and is allowed to interact with the MDP over a set of episodes (the number is linear with respect to $M$ in our paper). Due to the above difference, the worst-case lower bound of requiring an exponential number of episodes to learn an $\epsilon$-optimal policy in (Kwon et al., 2021) does not hold in our case. Indeed, if the identity (referred to as "context" in latent MDP) is revealed or can be inferred, (Kwon et al., 2021) is able to achieve a regret that is polynomial in the number of episodes (Thms. 3.3 and 3.4 from (Kwon et al., 2021)).

Furthermore, a close examination of the bounds provided by (Kwon et al., 2021) also reveals some differences from our result. In particular, let $K$ be the number of contexts in a latent MDP and $N$ be the total number of episodes ($N$ is on the order of $TM$ in our case as we encounter $T$ tasks, each with $M$ episodes). Then (Kwon et al., 2021) is able to bound the regret (without dividing by the number of episodes $N$) as $\mathcal{O}(\sqrt{KN})$. To compare their bound with ours, we consider each task in the meta-learning setting as a context, so $T = \mathcal{O}(K)$. Therefore, their upper bound (after dividing by the number of episodes $N = TM$) becomes $\mathcal{O}(1/\sqrt{M})$, which does not diminish with the number of tasks $T$. Note that our bound (see the comment after Corollary 1) is $\mathcal{O}\left(\frac{\hat{V}_\psi}{M^{3/4}\sqrt{T}}\right)$ (after dividing by the number of episodes $N = TM$), where for simplicity we have assumed $\mathcal{E}_T = 0$, i.e., exact access to the loss function. Note that $\hat{V}_\psi$ is a measure of task-relatedness (a smaller value indicates more relatedness among tasks). It can be seen that while we have a worse order dependence on $M$, our bound scales with task relatedness $\hat{V}_\psi$ and diminishes with respect to the increasing number of tasks $T$. This is expected as we leverage the relatedness among contexts (in fact, the result of (Kwon et al., 2021) would hold when tasks are sufficiently different from each other to infer the contexts with spectral methods). In summary, we refer to (Kwon et al., 2021) as an example that achieving regret diminishing in the number of tasks $T$ is hard, even with the assumption of observing the task identities (contexts).

## J    NOTATIONS AND CONSTANTS

| Notation | Definition |
|---|---|
| $t$ | index of task |
| $k$ | index of OGD steps |
| $\phi_t, \pi_{t,0}$ | policy initialization for task $t$ |
| $\alpha_t$ | learning rate of within-task algorithm |
| $t \in [T]$ | set of all tasks, where $[T] = \{1, \ldots, T\}$ |
| $\mathcal{M}_t$ | CMDP for task $t$ |
| $\mathcal{S}$ | state space of CMDP $\mathcal{M}_t$ |
| $\mathcal{A} \in \mathbb{R}^{n_a}$ | action space of CMDP $\mathcal{M}_t$ |
| $\rho_t$ | initial state distribution of task $t$ |
| $P_t(\cdot|s,a)$ | transition kernel for task $t$ |
| $c_{t,0} : \mathcal{S} \times \mathcal{A} \to [0,1]$ | reward function |
| $c_{t,i} : \mathcal{S} \times \mathcal{A} \to [0,1]$ | cost function $i$ for task $t$ |
| $p$ | total number of constraints |
| $m \in [M]$ | set of all timesteps for within-task algorithm |
| $\Delta(\mathcal{A})^{|\mathcal{S}|}$ | simplex over all state-action pairs |
| $\pi_t : \mathcal{S} \to \Delta(\mathcal{A})$ | stochastic policy for task $t$ |
| $\nu_t^\pi$ | state visitation distribution of policy $\pi$ at task $t$ |
| $\theta$ | softmax policy parameters |
| $V_{t,\pi}^i(s)$ | state-value function for reward ($i=0$) or cost $i$ in task $t$ with policy $\pi$ |
| $Q_{t,\pi}^i(s,a)$ | action-value function for reward ($i=0$) or cost $i$ in task $t$ with policy $\pi$ |
| $J_{t,i}(\pi)$ | expected total reward ($i=0$) or cost $i$ for task $t$ and policy $\pi$ |
| $d_{t,i}$ | bound on the expected total cost $i$ for task $t$ |
| $\Pi_t^*$ | set of optimal solutions for task $t$ |
| $\pi_t^*$ | optimal policy for task $t$ |
| $c_{max}$ | maximum value of reward/cost functions |
| $D_{KL}(\cdot|\cdot)$ | KL divergence |
| $\bar{R}_0, \bar{R}_i$ | TAOG and TACV |
| $\hat{\pi}_t$ | suboptimal policy returned by within-task algorithm in task $t$ |
| $D^*, \hat{D}^*$ | true and empirical task-similarity |
| $V_\psi, \hat{V}_\psi$ | true and empirical task-relatedness |
| $\Delta\mathcal{A}_\varrho$ | shrinkage simplex set inside $\Delta\mathcal{A}$ |
| $L_g, L_\pi$ | Lipschitzness and smoothness parameter for KL divergence of policy $\pi$ w.r.t. initial policy |
| $C_\pi$ | maximum value of KL divergence of policy $\pi$ w.r.t. initial policy |
| $\omega_{\pi/D_t}(s,a)$ | stationary distribution correction for task $t$ at state $s$ action $a$ |
| $\mathcal{D}_t$ | off policy dataset for task $t$ |
| $\mu_\pi$ | strong convexity parameter for KL divergence of policy $\pi$ w.r.t. initial policy |
| $\{\psi_t^*\}_{t=1}^T$ | dynamically varying comparator sequence |
| $\epsilon_t$ | inexactness in the KL divergence estimation using DualDICE |
| $\hat{\nu}_t$ | state distribution induced by $\hat{\pi}_t$ |
| $\epsilon_{opt}$ | optimization error in DualDICE |
| $\epsilon_{approx}$ | approximation error in DualDICE |
| $\mathcal{F}, \mathcal{H}$ | hypothesis class used in DualDICE |
| $\mathcal{E}_T$ | cumulative inexactness for KL divergence estimation given by $\sum_{t=1}^T \epsilon_t$ |
| $\tilde{\mathcal{E}}_T$ | cumulative square root of inexactness for KL divergence estimation given by $\sum_{t=1}^T \sqrt{\epsilon_t}$ |
| $h : [0,\rho] \to (0,\infty)$ | strictly increasing continuous definable function used in Theorem 3.1 |
| $\nabla_t, \hat{\nabla}_t$ | Exact and inexact gradient |
| $\beta$ | learning rate for inexact OGD |
| $P_X(\cdot)$ | Projection operator |
| $\mathcal{P}_T$ | path-length of the comparator $\psi_t^*$ |
| $K_{in}$ | number of iterations in the critic estimation of CRPO |
| $\eta_t$ | tolerance for the constraint violation $d_{t,i}$ for task $t$ |
| $\mathcal{P}_T, \mathcal{S}_T$ | path-length and squared path-length of the comparator $\psi_t^*$ |
| $\partial_\epsilon f(\cdot)$ | $\epsilon$-subgradient of function $f$ |
| $Dom(f)$ | Domain of the function $f$ |
| $\hat{f}_t(\cdot)$ | loss functions for suboptimal policy $\mathbb{E}_{\hat{\nu}_t}[D_{KL}(\hat{\pi}_t|\pi_{t,0})]$ |

Table 1: Table of notations

Table 2: Table of constants.

| Notation | Definition | Notation | Definition |
|---|---|---|---|
| $c_1^t$ | $2$ | $c_2^t$ | $\frac{4c_{max}^2|\mathcal{S}||\mathcal{A}|}{(1-\gamma)^3}$ |
| $c_3^t$ | $\frac{3+(1-\gamma)^2}{(1-\gamma)^2}$ | $c_4^t$ | $\frac{3c_{max}}{(1-\gamma)^2}$ |
| $c_5^t$ | $\frac{2\sqrt{(1-\gamma)}|\mathcal{S}||\mathcal{A}|}{1-2\kappa}$ | $c$ | $c \in \left(0, \frac{2}{L_\pi}\right)$ |
| $C_1$ | $2(L_\pi + \beta)$ | $C_2$ | $(L_\pi + \beta)\frac{3c\alpha+6\alpha L_\pi}{2\mu_\pi\alpha L_\pi}$ |
| $C_3$ | $3(L_\pi + \beta)$ | $C_4$ | $\frac{2L_g}{2-\sqrt{2}}$ |
| $C_5$ | $\frac{2L_g}{2-\sqrt{2}}\sqrt{\frac{c\alpha+2L_\pi\alpha}{2\alpha\mu_\pi L_\pi}}$ | | |

