# OpenReview forum: "A CMDP-within-online framework for Meta-Safe Reinforcement Learning"
_ICLR.cc/2023/Conference — ICLR 2023 notable top 25%_

### Official Review · Reviewer_GvYx · 2022-10-23

**Confidence:** 3
**Clarity, Quality, Novelty And Reproducibility:** The writing of the paper is somewhat …
**Correctness:** 3
**Technical Novelty And Significance:** 2
**Empirical Novelty And Significance:** 2
**Recommendation:** 3

**Strength And Weaknesses:**

Strength: Propose the a novel CMDP-within-online framework where the within-task is CMDP and the meta learner aims to learn the meta-initialization and learning rate. It shows that the task-averaged regrets for optimality gap (TAOG) and constraint violations (TACV) diminish with respect to both the number of steps for the within-task algorithm M and the number of tasks T.  It adapts the learning rates for each task
to a dynamic environment.

Weakness:

1. the notation $\nu^*_t$ in page 2 is unclear. What is $\pi^*$? It makes harder to understand this paper.

2. The primal approach explained in Section 2.1 is confusing. Why the sub-optimality gap in (2) holds by running CRPO? What is CRPO? Is CRPO an online or offline algorithm? There is no explanation about CRPO and no comments about the "carefully chosen parameters". It is not helpful for people who is not familiar with CRPO and makes the paper not self-contained. Those observations make the paper hard to read.

3. Lemma 1 relies on $D^*$ is known and Theorem 3.1 relies on approximation error of DualDIce. The whole algorithm/analysis seems to be a composition of existing analysis (DualDice+CRPO), which might lack generality. It is hard to understand what is the key contribution of this paper.

4. Theorem 3.3 replies on the assumptions of two regret upper bound $U^{init}_T$ and $U^{sim}_T$. There is no explanation regrading why those assumptions are reasonable. With those assumptions, the hardness of analysis is largely diminished.

5. In the experiment, only frozen lake and acrobot are considered. How about the your algorithm for Mujoco environments?

**Summary Of The Paper:**

This paper studies the problem of meta-safe reinforcement learning (Meta-SRL) through the CMDP-within-online framework to establish the first provable guarantees in this important setting. It obtains task-averaged regret bounds for the reward maximization (optimality gap) and constraint violations using gradient-based meta-learning and show that the task-averaged optimality gap and constraint satisfaction improve with task-similarity in a static environment, or task-relatedness in a dynamic environment.

Furthermore, it enables the learning rates to be adapted for every task and extends our approach to settings with a competing dynamically changing oracle. Finally, experiments are conducted to demonstrate the effectiveness of the approach.


**Summary Of The Review:**

Please address the questions I mentioned above.

---

> ### Author Response · Authors · 2022-11-13
> **Extra experiments on Mujoco**
>
> **W5: In the experiment, only frozen lake and acrobot are considered. How about your algorithm for Mujoco environments?**
>
>
> A: Thank you for the suggestion. Please see the revised paper where we have included additional experiments on the HalfCheetah and Humanoid from Mujoco. We discuss the results and some design aspects of the HalfCheetah experiments below.
>
> **Safety constraint.** The original HalfCheetah environment does not have any constraints. We introduce a constraint that penalizes the deviation of the cheetah’s head from some desired height:
>
> $$h_{cheetah} - h_{target} \leq \epsilon,$$
>
> where we specify the cumulative absolute difference between the current cheetah head height and the desired height to be less than a tolerance $\epsilon$.
>
> **Experimental setup.** We trained the cheetah on a sequence of tasks with high or low similarity. To generate tasks with high similarity, the goal velocity for each training task is uniformly sampled from a range of $[0.35,0.65]m/s$, making each task relatively similar. To generate tasks with low similarity, the goal velocity for each training task is uniformly sampled from a wider range of $[0.0,1.5]m/s$. In the latter case, tasks are less similar due to the high variance of goal velocities that the cheetah is trained to achieve. The total number of tasks is $T=100$ for both settings.
>
> **Results and discussions.** The plots in Appendix H.3 of the revised paper show the performance of the Meta-SRL and baseline methods under the high-and-low task-similarity settings. It can be observed that Meta-SRL is able to achieve higher rewards and zero constraint violations quickly compared to other baseline initializations under low task-relatedness settings. The pre-trained baseline can achieve higher rewards, but similar to other baselines, it cannot achieve constraint satisfaction within 10 steps. Under high task-similarity settings, both pre-trained and Meta-SRL perform well, as expected. However, it can be observed that both simple averaging and random initialization perform poorly in this setting. Close inspection indicates that there is a high variance among the policy parameters learned from each task, which may result in interference among different tasks in the relatively high dimensional state space. For more details, please see the experimental section and Appendix H.3 in the updated paper.

---

> ### Author Response · Authors · 2022-11-13
> **Details regarding CRPO**
>
> **W2: The primal approach explained in Section 2.1 is confusing. Why does the sub-optimality gap in (2) hold when running CRPO? What is CRPO? There is no explanation about CRPO and no comments about the "carefully chosen parameters". It is not helpful for people who are not familiar with CRPO and makes the paper hard to read.**
>
> A: Due to space restrictions, we were not able to include all the details for the CRPO algorithm in the main text. To address your concern, we have now included an additional sentence explaining the high-level idea of the CRPO algorithm, and another sentence that refers the reader to Appendices B and D for further details on CRPO:
>
> Appendix B: CRPO algorithm and notations used in the discussions.
>
> Appendix D: auxiliary results to support the main results of the paper, where each result also includes specifications on the choice of parameters. We have added comments on the chosen parameters after each result.
>
> **Why suboptimality in (2) holds?**
>
> The suboptimality gap in eq.(2) directly follows from Thm. 1 of (Xu et.al. 2021), where we make the simplifying assumption that the Q-estimation error is zero. For completeness, we have included a simplified proof in Lemma 4 in Appendix D.
>
> **What is CRPO? Comment on the choice of parameters.**
>
> (Added to the main paper:) “CRPO is a primal-based online CMDP algorithm, which performs either policy optimization (natural gradient ascent on the reward) when constraints are not violated, or constraint minimization (natural gradient descent on the constraint function) for one of the violated constraints. For more details about the CRPO algorithm, choice of parameters, and the convergence analysis for a single task, we refer the reader to Appendices B and D.”
>
> More specifically, Appendix B introduces the details of the working of CRPO and its parameters. In Appendix D, Lemma 4 provides a simplified proof of how the parameters can be carefully chosen to achieve a convergence rate sublinear in the number of timesteps $M$. A more comprehensive result where we consider the effect of Q-estimation error and adaptive learning rates for CRPO can be found in Appendix G (Theorem G.1).

---

> ### Author Response · Authors · 2022-11-13
> **Unclear notation on page 2**
>
> **W1: The notation** $\nu_t^*$ **on page 2 is unclear. What is** $\pi^*$? **It makes it harder to understand the paper.**
>
> A: We apologize for the typo in the notation of $\pi^*$, which should be $\pi_t^*$. Specifically, $\pi_t^*$ is an optimal policy for **task** $t$. Hence, $\nu_t^*(s) := \mathbb{E}_{s_0\sim \rho_t} \left[\nu^{\pi_t^*}\_{t,s_0}(s)\right]$ is the corresponding state visitation distribution induced by policy $\pi_t^*$ when the initial state $s_0$ is sampled from initial state distribution $\rho_t$ at task $t$. We have addressed this in the updated version of the paper, where changes are highlighted in blue.

---

> ### Author Response · Authors · 2022-11-13
> **Clarification regarding Theorem 3.3 assumptions**
>
> **W4: Theorem 3.3 relies on the assumptions of two regret upper bounds** $U_T^{init}$ and $U_T^{sim}$.**There is no explanation regarding why those assumptions are reasonable. With these assumptions, the hardness of the analysis is largely diminished.**
>
> A: We would like to clarify this major misunderstanding of our key results. Terms $U_T^{init}$ and $U_T^{sim}$ are simply placeholders for upper bounds on the regret for some inexact online optimization algorithm and are introduced to state our results in the most general way. In our Algorithm 1, INIT and SIM can be any inexact online optimization algorithms and the results of Thm. 3.3 can be instantiated by plugging in the respective regret upper bounds $U_T^{init}$ and $U_T^{sim}$.
>
> In the online optimization literature, the regret bounds are well-established for many algorithms such as follow the (regularized) leader and online gradient descent (OGD), when the online learners have exact or bandit access to the loss functions. In our setting, we have access to a loss function that may differ from the true loss functions, i.e., inexact access. This is because we do not have access to the optimal policies and state distributions, but only a suboptimal policy and the estimated state distributions. Hence, it requires us to derive static and dynamic regret bounds for the inexact versions of the loss. To this end, we also provide the **first results** on bounding the dynamic and static regrets (i.e., $U_T^{init}$ and $U_T^{sim}$) for inexact OGD.  Please refer to Appendix E for these results.
>
> To specialize the general result of Thm. 3.3 to the case of inexact OGD, we present the upper bounds on TAOG and TACV, in Corollary 1. Some highlights of this result are as follows:
>
> * The bounds are improved in terms of $M$ and $T$ due to the adaptation of the learning rates. Specifically, the bounds diminish at a rate of $\mathcal{O}\left(\frac{1}{M^{3/4}\sqrt{T}}\left(\mathcal{E}_T+ \sqrt{\frac{\mathcal{E}_T}{T} +\hat{V}_\psi^2 } \right) \right)$ as compared to the previous rate $\mathcal{O} \left(\frac{1}{\sqrt{M}}\sqrt{\frac{\mathcal{E}_T}{\sqrt{T}}+  \hat{D}^{*2} } \right)$.
>
> * The above result also makes explicit the dependence on task-relatedness and theoretically validates the benefits of meta-safe RL.
>
> * A practical aspect of our algorithm is that it does not require the knowledge of quantities such as $\mathcal{S}_T$, $\mathcal{P}_T$ and $\mathcal{E}_T$ to decide the value of learning rate $\alpha_t$.

---

> ### Author Response · Authors · 2022-11-13
> **Composition of existing analysis (DualDICE+CRPO) - Clarification**
>
> **W3: Lemma 1 relies on known** $D^*$**, Thm 3.1 relies on the approximation error of DualDICE. The whole algorithm/analysis seems to be a composition of existing analysis (DualDICE+CRPO), which might lack generality. It is hard to understand what is the key contribution of this paper.**
>
> A: We respectfully disagree with the comment: “seems to be a composition of existing analysis (DualDICE + CRPO) which might lack generality”.
>
> *Firstly*, to obtain the error bound in Thm. 3.1, we propose a decomposition of three terms in the following equation (eq. 4 in the paper):
> $$\mathbb{E}\_{\nu_t^*}[D\_{KL}(\pi_t^*|\pi)] - \mathbb{E}\_{ \hat{\nu}_t}[D\_{KL}(\hat{\pi}\_{t}|\pi)]=\underbrace{\mathbb{E}\_{\nu_t^*}[D\_{KL}(\pi_t^*|\pi)] - \mathbb{E}\_{ \tilde{\nu}\_t}[D\_{KL}(\pi_t^*|\pi)]}\_{(A)}+ \underbrace{\mathbb{E}\_{ \tilde{\nu}\_t}[D\_{KL}(\pi_t^*|\pi)] - \mathbb{E}\_{ \hat{\nu}\_t}[D\_{KL}(\pi_t^*|\pi)]}\_{(B)}+\underbrace{\mathbb{E}\_{\hat{\nu}\_t}[D\_{KL}(\pi_t^*|\pi)] - \mathbb{E}\_{ \hat{\nu}\_t}[D\_{KL}(\hat{\pi}\_{t}|\pi)]}\_{(C)}.$$
> This decomposition is **general** in the sense that it provides a guideline to bound each term with strategies potentially different from the paper. Note that only the second error term B uses the analysis of DualDICE, which can be replaced by a different analysis should we choose a different way to estimate the stationary distribution; this is of no loss of generality.
>
> To bound the other 2 terms (A and C), we develop **new techniques** based on tame geometry and subgradient flow systems, which help us control the terms    $|\hat{\pi}_t - \pi_t^*|$ and $|\nu_t^* - \tilde{\nu}_t|$  in terms of policy parameters, i.e., $|\theta_t^* - \hat{\theta}_t|$. This is the **first result** in this general setting, and extends our current understanding of the analysis of RL. As a reference, the only relevant result that characterizes the distance of a suboptimal policy to the set of global policies has been limited to some restricted settings, i.e., (natural) policy gradients (Mei et al., (2020), Lemmas 3 and 15, non-uniform Lojasiewicz for policy gradient). Our result is **algorithm-agnostic** as we directly analyze the landscape of the optimization. All details are presented in Appendix F, and the comparison with the related methods in the literature is discussed in the paragraph before Assumption 2.

---

> > ### Author Response · Authors · 2022-11-13
> > **Further points: the analysis with full CRPO and guides on instantiation to other cases**
> >
> >
> > *Second*, Lemma 1 and Theorem 3.1 are preliminary results presented to aid the understanding of the general methodology, although each of them is novel and requires nontrivial derivations. The full results--our key contributions are Theorems 3.2 and 3.3, where we provide a rigorous analysis of the task-averaged static and dynamic regret bounds of the proposed CMDP-within-online methods under the **most practical settings** (critic estimation errors, adaptive learning rates).
> >
> > *Last*, the overall framework is applicable for the design and analysis of meta-safe RL and is not limited to CRPO. To *instantiate our framework*, we only need the following ingredient:
> >
> > *A convergence/regret analysis for a single-task CMDP, which makes *explicit* the dependence on convergence/regret of the algorithm on meta parameters (initialization, learning rates, etc.)*
> >
> > In the case of CRPO, the original paper provides a convergence analysis but does not make the dependence on meta-parameters explicit, so it entails some derivations. For a potentially different algorithm, such derivations can be inspired by studying how it performed in our paper (see Lemma 4 in Appendix B and Lemma 21 in Appendix G). In addition, the decomposition proposed in eq. 4 in the main text applies in general to Bregman divergence, which includes the KL divergence. The analysis in Appendix F that bounds each of the three terms can be adapted to the general settings.
> >
> > **References:**
> >
> > Mei et al. (2020) "On the global convergence rates of softmax policy gradient methods." ICML

---

> ### Author Response · Authors · 2022-11-13
> **Response to reviewer GvYx**
>
> Dear reviewer GvYx,
>
> We value your feedback and your recognition of the strength of the proposed CMDP-within-online framework. First, to address your concern regarding our **key contributions**, we list them as below:
>
> * We provide a CMDP-within-online framework for meta-safe reinforcement learning that is **practical** in the following aspects:
>   1) to perform meta-learning updates, only the policy $\hat{\pi}_t$ learned from each task is needed (instead of the optimal policy $\pi_t^*$),
>   2) similarly, we do not need to have access to the optimal policy $\pi_t^*$,
>   3) the learning rates are adapted along with the initial policies to alleviate the need to access the task-similarity/task-relatedness.
>
> * We provide a rigorous analysis of the regret bounds for the **task-averaged optimality gap (TAOG)** and **task-averaged constraint violation (TACV)**, and establish that they scale with task-similarity or task-relatedness (Thm. 3.2, and 3.3, Cor. 1). **Our analysis is sound** in the following sense:
>     1) we bound the TAOG and TACV, which are dynamic regret with respect to a sequence of optimal policies for each task,
>     2) we analyzed the regret bounds on TAOG and TACV by examining two important cases, where we compare with a **static initial policy** (in a static environment), and a sequence of **dynamic initial policies** (in a shifting environment) (please refer to our response to Reviewer fmH2 (https://openreview.net/forum?id=mbxz9Cjehr&noteId=uMYBqKOia9), for further explanation of this important aspect),
>     3) we consider the most practical settings (critic estimation errors, adaptive learning rates, no access to optimal policies) with the original CRPO algorithm, and 4) our result establishes the **theoretical benefits** of meta-learning that can leverage task-similarity and task-relatedness.
>
> Furthermore, we remark on our **key technical contributions** that support the above developments, which may be of independent interest:
> * We study the *optimization landscape* of CMDP that is *algorithmic-agnostic* (unlike the existing work of (Mei et al. (2020), Lemmas 3 and 15, non-uniform Lojasiewicz for policy gradient) that is restricted to the setting of policy gradient (see Thm 3.1). This is achieved by developing *new techniques* based on tame geometry and subgradient flow systems. (This result is crucial to remove the assumption that requires access to the global optimal policy in regret analysis.)
> * We provide static and dynamic regret bounds for **inexact online gradient descent** (see Theorems E.1, E.2, E.3, and Lemma 11 in Appendix E), which we leverage to obtain our final theoretical results (Thm. 3.2 and 3.3, Cor. 1).
> * Our theoretical analysis provides a **template** to analyze similar scenarios of meta-safe RL, where the within-task algorithm may be different from CRPO. To *instantiate our framework*, we only need the following ingredient: *a convergence/regret analysis for a single-task CMDP, which makes explicit the dependence on convergence/regret of the algorithm on meta parameters (initialization, learning rates, etc.)* Please see our response to https://openreview.net/forum?id=mbxz9Cjehr&noteId=-oy1yPAGubD
>
> Below, we provide further details regarding your concerns in a point-by-point manner. If you have any additional clarification questions, we would appreciate a follow-up discussion.

---

> ### Author Response · Authors · 2022-11-15
> **A friendly reminder of the rebuttal's conclusion**
>
> Respected reviewer GvYx,
>
> We'd like to express our gratitude once more for your constructive feedback, which lead to several updates that improve the understanding of our work. We've responded to each of your questions. Hopefully, you'll find that they adequately address your concerns.
>
> In addition, we would like to know if you have any additional questions or require clarification before the rebuttal phase concludes. We would be happy to address them in the rebuttal's revision.
>
> Best wishes,
>
> Authors of Paper3277

---

> ### Author Response · Authors · 2022-11-17
> **We hope that our responses addressed your concerns**
>
> Dear reviewer GvYx,
>
> Thank you once again for taking the time to review our paper, and giving us insightful comments to improve the paper. We'd appreciate it if you could let us know whether our responses addressed your concerns (especially on the contributions of our work). As the end of the rebuttal phase approaches, we look forward to hearing from you and remain available for any additional clarification you may require.
>
> With gratitude,
>
> Authors of paper3277

---

> ### Author Response · Authors · 2022-11-21
> **Start of phase 2 discussion: we anticipate your feedback!**
>
> Dear Reviewer GvYx,
>
> As the phase 2 discussion period has started, we sincerely look forward to your feedback. The authors deeply appreciate your time and efforts spent reviewing the paper and helping improve its quality.
>
> We would highly appreciate it if you could once again help review our responses and let us know if we were able to address your concerns.
>
> Please also let us know if there are further questions or suggestions about this paper. We strive to keep improving the quality of the paper, and we are highly grateful for your feedback and time.
>
> With gratitude,
>
> Authors of paper 3277

---

> ### Author Response · Authors · 2022-11-26
> **Looking forward to your feedback!**
>
> Dear Reviewer GvYx,
>
> The conclusion of discussion period is closing, and we eagerly await your response. The authors greatly appreciate your time and effort in reviewing this paper and helping us improve it. Please help us to review our responses once again and kindly let us know if they fully or partially address your concerns.
>
> Kind regards,
>
> Authors of Paper3277

---

### Official Review · Reviewer_fmH2 · 2022-10-24

**Confidence:** 4
**Correctness:** 3
**Technical Novelty And Significance:** 3
**Empirical Novelty And Significance:** 2
**Recommendation:** 6

**Clarity, Quality, Novelty And Reproducibility:**

Overall, the paper has a clear delivery except for some ambiguity. Since the paper builds on several existing algorithms, mixing notation from different sources make it hard to follow the main results. More efforts need to be made to explain the proposed algorithm well. Some assumptions or techniques are introduced in the middle of the paper, while it is less discussed how do they address the key challenge. In addition, many references are mixed in the content, which is a distraction to readers.

The paper presents a new meta learning framework for safe RL, which is useful to generalize across different safe learning tasks. To achieve this, the proposed method combines several existing techniques in a clever way. However, the analysis and assumption mostly build on existing results, the novelty in which requires justification. The experiment utilizes a simple RL environment which might be too simple for meta-learning to generalize.

Here are some other questions for consideration:

- It is useful to summarize an abstract meta algorithm at the beginning instead of delaying Algorithm 1 to the end of the paper.

- Whey we can't use the original CRPO results? Assuming exact action-value function makes the proposed algorithm and theory impractical.

- How does Assumption 1 address the hardness results from Kwon et al. (2021)? How are they related to your constrained learning problems?

- Assumption 2 need to be clarified: 'o-minimal structure'.

- Please check your citation formats. They are not standard.

- It is useful to verify the effectiveness of the proposed method in other environments, e.g., humanoid?

**Details Of Ethics Concerns:**

The paper concerns the meta-learning problem with multiple constrained learning tasks. It is important to discuss the pros/cons of the proposed framework when engineers apply the proposed method to problems with fairness constraints on different tasks.

**Strength And Weaknesses:**

Strengths:

- The studied meta learning problem is motivated by applications of learning a safe policy. This is a useful generalization of meta-learning to  constrained Markov decision processes, which allows to adapt the learnt policy to new constraints quickly.

- The propsoed CMDP embedded online learning framework is practical in sense that it only relies on the inexact solution to sub-CMDP  tasks and data trajectories from previous tasks.

- When the task similarity is bounded, the authors provide theoretical guarantees on both task-within optimality gap and constraint violation in static regret bound, under some assumptions on policy initializations. For practical use, the authors generalize this result to a dynamic regret bound, which is more useful since it does not assume boundedness of the task-similarity.

Weaknesses:

- The provided CMDP embedded online learning framework builds on a simplified version of the existing CRPO algorithm. Although this simplification permits convenience in analysis, it brings some restriction to the applicability of the proposed algorithm.

- In the meta-learning problem with multiple tasks, the static regret seems to be not very useful since the bounded ask similarity is assumed and environments associated with different tasks are different. Otherwise, this becomes a single task problem which doesn't have to necessitate meta-learning.

- The authors mentioned the hardness of bounding regrets in meta-learning by citing Kwon et al. (2021). This seems to be the key challenge to establish the theoretical guarantees. Although a series of assumptions are made, it is not very clear if they are necessary to address this challenge. In addition, the challenge from constraint seems to be unrevealed yet.

- The proposed CMDP embedded framework relies on DualDICE to estimate state-action distributions. However, DualDICE is an existing offline RL algorithm for the standard MDPs. It is more natural to think of some off-line CMDP algorithms that respect the constraints more effectively.


**Summary Of The Paper:**

The paper studies the meta-learning problem regarding various safe RL tasks that are modeled by constrained Markov decision processes (CMDPs).  The authors proposed a CMDP embedded online learning framework to generalize learnt safe policies to other unseen CMDP environments. In theory, the authors proved regret bounds in terms of within-task optimality gap and constraint violations in both static and dynamic regret metrics. Finally, some comparison experiments are provided to show the effectiveness of the proposed method.

**Summary Of The Review:**

All claims of the paper are supported by proofs or references. Most of them are well-supported, but some cited results are vague or modified in an ambiguous way. The proposed meta learning method combines existing RL algorithms for safe RL and theory builds on existing results and assumptions. The experiments are based on some basic RL environments.


============================

POST-REBUTTAL. Thank you for your response. Since most of my concerns are addressed, the score has been updated.

---

> ### Author Response · Authors · 2022-11-13
> **Discussions on the ethical aspects of the proposed work**
>
> **Details of Ethics Concerns: The paper concerns the meta-learning problem with multiple constrained learning tasks. It is important to discuss the pros/cons of the proposed framework when engineers apply the proposed method to problems with fairness constraints on different tasks.**
>
> A:  Thank you for pointing out this important issue. Please see the following paragraph of discussions.
>
> There is an increasing need to address fairness as a constraint in learning settings. Existing works that aim to achieve zero-shot generalization without any task-specific adaptation have limited capability to adapt to shifting environments. While online meta-learning is a principled technique to learn good priors over model parameters for fast adaptation in a sequential setting, existing methods often do not address constraints and thus have limited applications in fairness-aware learning.
>
> The proposed framework can potentially be adapted to reinforcement learning tasks with fairness constraints in a **nonstationary environment**. In practice, this can provide a strategy that learns priors over policy parameters not only to master the current fairness-aware task but also to become proficient with quick adaptation at learning newly arrived tasks. Our theoretical analysis can be leveraged to provide sublinear bound on the “task-averaged fairness violation” regret. Similar ideas have been explored by Zhao et al. (2021) in the supervised learning setting, while we are not aware of any work on the reinforcement learning counterpart. Thus, it can be an extension for future work to explore the extent to which our method can address this important problem.
>
> Nevertheless, fairness constraints present a unique challenge for meta-safe RL settings, as fairness constraints should rarely be violated in a real-world setting due to the implicated discrimination or bias. Additional efforts, such as incorporating pessimism or developing offline methods, may be entailed to reduce fairness violations during initial deployment.
>
> **References**
>
> Bai et al (2022). "Achieving Zero Constraint Violation for Concave Utility Constrained Reinforcement Learning via Primal-Dual Approach.".
>
> Zhao et al. (2021). "Fairness-aware online meta-learning." Proceedings of the 27th ACM SIGKDD Conference on Knowledge Discovery & Data Mining. 2021.

---

> ### Author Response · Authors · 2022-11-13
> **Extra experiments on Mujoco**
>
> **Q6: It is useful to verify the effectiveness of the proposed method in other environments, e.g., humanoid?**
>
> A: Thank you for the suggestion. Please see the revised paper, where we have included additional experiments on the HalfCheetah and Humanoid environment in Appendix H.
>
> We discuss the results and some design aspects for the HalfCheetah experiments below.
>
> **Safety constraint.** The original HalfCheetah environment does not have any constraints. We introduce a constraint that penalizes the deviation of the cheetah’s head from some desired height:
>
> $$h_{cheetah} - h_{target} \leq \epsilon,$$
>
> where we specify the cumulative absolute difference between the current cheetah head height and the desired height to be less than a tolerance $\epsilon$.
>
> **Experimental setup.** We trained the cheetah on a sequence of tasks with high or low similarity. To generate tasks with high similarity, the goal velocity for each training task is uniformly sampled from a range of $[0.35,0.65]m/s$, making each task relatively similar. To generate tasks with low similarity, the goal velocity for each training task is uniformly sampled from a wider range of $[0.0,1.5]m/s$. In the latter case, tasks are less similar due to the high variance of goal velocities that the cheetah is trained to achieve. The total number of tasks is $T=100$ for both settings.
>
> **Results and discussions.** The plots in Appendix H.3 of the revised paper show the performance of the Meta-SRL and baseline methods under the high-and-low task-similarity settings. It can be observed that Meta-SRL is able to achieve higher rewards and zero constraint violations quickly compared to other baseline initializations under low task-relatedness settings. The pre-trained baseline can achieve higher rewards, but similar to other baselines, it cannot achieve constraint satisfaction within 10 steps. Under high task-similarity settings, both pre-trained and Meta-SRL perform well, as expected. However, it can be observed that both simple averaging and random initialization perform poorly in this setting. Close inspection indicates that there is a high variance among the policy parameters learned from each task, which may result in interference among different tasks in the relatively high dimensional state space. For more details, please see the experimental section and Appendix H.3 in the updated paper.
>
>
> We would like to remark that it is not a trivial task to adapt an environment designed for a single task, such as Frozen lake and Acrobot, to the meta-learning settings, since we need to implement the most physically meaningful ways to simulate “nonstationarity” between tasks. Furthermore, for environments like Mujoco, we also need to design proper utility functions to incorporate constraints. Fundamentally, this is due to a lack of benchmark problems in this important setting (meta-learning with safety constraints). On this note, we believe another contribution of our study is to initiate the development of some potential benchmark problems to facilitate future developments.

---

> ### Author Response · Authors · 2022-11-13
> **Clarification on Assumption 2: 'o-minimal structure'**
>
> **Q4: Assumption 2 needs to be clarified: 'o-minimal structure'.**
>
> A: The definition of  "o-minimal structure” is provided in Appendix F.1. Assumption 2 is not a strong assumption as practically all functions from real-world applications, including deep neural networks, are definable in some o-minimal structures; also, the composition of mappings, along with the sum, inf-convolution, and several other classical operations of analysis involving a finite number of definable objects in some o-minimal structure remains in the same structure. For Assumption 2 to hold, a sufficient condition is to require that the reward and utility functions belong to the same o-minimal structure.
>
> **Next, we clarify the places where we use Assumption 2**. Recall the following decomposition (eq.4 in the main text):
>
> $$\mathbb{E}\_{\nu_t^*}[D\_{KL}(\pi_t^*|\pi)] - \mathbb{E}\_{ \hat{\nu}_t}[D\_{KL}(\hat{\pi}\_{t}|\pi)]=\underbrace{\mathbb{E}\_{\nu_t^*}[D\_{KL}(\pi_t^*|\pi)] - \mathbb{E}\_{ \tilde{\nu}\_t}[D\_{KL}(\pi_t^*|\pi)]}\_{(A)}+ \underbrace{\mathbb{E}\_{ \tilde{\nu}\_t}[D\_{KL}(\pi_t^*|\pi)] - \mathbb{E}\_{ \hat{\nu}\_t}[D\_{KL}(\pi_t^*|\pi)]}\_{(B)}+\underbrace{\mathbb{E}\_{\hat{\nu}\_t}[D\_{KL}(\pi_t^*|\pi)] - \mathbb{E}\_{ \hat{\nu}\_t}[D\_{KL}(\hat{\pi}\_{t}|\pi)]}\_{(C)}.$$
>
> where we bound the difference between $\mathbb{E}\_{\nu_t^*}[D\_{KL}(\pi_t^*|\pi)]$ and $\mathbb{E}\_{ \hat{\nu}_t}[D\_{KL}(\hat{\pi}\_{t}|\pi)]$. Assumption 2 is required to bound terms (A) and (C), as established in Thm. 3.1. In particular, to bound the terms (A) and (C), we resort to bound the distance between the suboptimal policy $\hat{\pi}_t$ and optimal policy $\pi_t^*$. Controlling the distance between any policy and an optimal policy based on the suboptimality gap requires the optimization to have some curvatures around the corresponding optima. By assuming the definability of policy parametrization and objective/constraints in some o-minimal structure allows the optimization to have some curvature around the optima by recalling the nonsmooth Kurdyka-Lojasiewicz inequality. The actual proof is very technical and the details can be found in Appendix F.
>
> **Last**, we remark that the present study is the first to analyze the optimization landscape of the CMDP in these general settings and the result extends our current understanding of the analysis of RL. As a reference, the only relevant result that characterizes the distance of a suboptimal policy to the set of global policies has been limited to some restricted settings, i.e., (natural) policy gradients (Mei et al. (2020), Lemmas 3 and 15, non-uniform Lojasiewicz for policy gradient). Our result is algorithm-agnostic as we directly analyze the landscape of the optimization.
>
> **References**
>
> Mei et al. (2020) "On the global convergence rates of softmax policy gradient methods." ICML

---

> ### Author Response · Authors · 2022-11-13
> **Relation of Assumption 1 to hardness result in Kwon et al. (2021)**
>
> **Q3: How does Assumption 1 address the hardness results from Kwon et al. (2021)? How are they related to your constrained learning problems?**
>
> A: As discussed in our response previously to the question titled "Relation of Meta-SRL to hardness results in (Kwon et al. (2021))", the setting considered in Kwon et.al is a special case of a partially observable MDP, which is the fundamental reason for the hardness result.  In our case, we assume the agent knows when it encounters a new task, so their hardness result does not apply. In other words, Assumption 1 is not used to address the hardness results from Kwon et al. (2021).
>
> Assumption 1 allows some initial explorations for the meta-initialization policy at the beginning of a task. Technically, Assumption 1 implies the following three conditions on the policy initialization (as discussed in the paragraph after Assumption 1): the following holds for the meta-initialization policy $\pi_{t,0}$ for any state $s \in \mathcal{S}$ with positive constants $C_\pi$, $L_g$, $L_\pi$ and $\mu_\pi$:
>
> 1) $|D_{KL}(\pi_t^*(\cdot|s)|\pi_{t,0}(\cdot|s))|$, $|D\_{KL}(\hat{\pi}\_t(\cdot|s)|\pi\_{t,0}(\cdot|s))|\leq C\_{\pi}$;
> 2) $D_{KL}(\pi_t^*(\cdot|s)|\pi_{t,0}(\cdot|s))$ is $L_g$-Lipschitz and $L_\pi$-smooth in $\pi_{t,0}(\cdot|s)$;
> 3) $D_{KL}(\pi_t^*(\cdot|s)|\pi_{t,0}(\cdot|s))$ is $\mu_\pi$-strongly convex in $\pi_{t,0}(\cdot|s)$.
>
> Essentially, we make use of these conditions in the proof of Lemma 1, Lemma 3, and Thm. 3.2, where we exploit the strong convexity, Lipschitzness, and boundedness of the KL divergence w.r.t., the meta-initialization policy. We also make use of boundedness and Lipschitzness of the KL divergence in bounding the terms (A) and (C) in eq. 4, and eventually obtain Thms 3.1 and 3.3.
>
> We expect that Assumption 1 is also needed in unconstrained meta-learning by adapting our method, i.e., the MDP-within-online framework. Technically, Assumption 1 is a minimal requirement even for single-task CRPO to provide provable guarantees. This can be seen in the convergence guarantee of the original CRPO method (Lemma 4 and Lemma 21 in our paper, or Thm. 3 in (Xu et al. 2021)) last line of their proof before the term $D_{KL}(\pi_t^*|\pi_{t,0})$ is hidden in the big-O notation). For example, as shown in our eq. 2,
>
> $$R\_0 = J_{t,0}(\pi\_t^*) - \mathbb{E}[J\_{t,0}(\hat{\pi}\_t)]\leq \frac{2}{\alpha_t M}\mathbb{E}_{s \sim \nu\_t^*}[D\_{KL}(\pi\_t^*|\pi\_{t,0})]+\frac{4 \\alpha_t c\_{max}^2|\mathcal{S}| |\mathcal{A}|}{(1-\gamma)^3}.$$
>
> To ensure that the bound is nontrivial, we need to bound the term $D_{KL}(\pi_t^*|\pi_{t,0}).$ However, if $\pi_{t,0}$ does not have full support over the state/action space, then there may be a state $s$ where $\pi_t^*(s) > 0$ but $\pi_{t,0}(s) = 0$, which would make the KL divergence infinite.

---

> ### Author Response · Authors · 2022-11-13
> **Suggestion of moving the algorithm to the beginning**
>
> **Q1: It is useful to summarize an abstract meta-algorithm at the beginning instead of delaying Algorithm 1 to the end of the paper.**
>
> A: Thank you for the suggestion. Please see the revised paper.

---

> ### Author Response · Authors · 2022-11-13
> **Why not use some offline CMDP algorithm to estimate state distributions, instead of DualDICE?**
>
> **W4: The proposed CMDP-embedded framework relies on DualDICE to estimate state-action distributions. However, DualDICE is an existing offline RL algorithm for the standard MDPs. It is more natural to think of some offline CMDP algorithms that respect the constraints more effectively.**
>
> A:  In our algorithm, we only need an algorithm to **estimate the stationary distribution** of the learned policy $\hat{\pi}_t$ with access to offline trajectory data, so DualDICE (an offline algorithm for estimation) exactly fits the requirement. Since the safety aspect is addressed by the within-task algorithm CRPO and Algorithm 1 uses the state distribution estimation from DualDICE to meta-initialize CRPO, an offline RL algorithm for potential policy improvement is **not** required for our method to work.
>
> However, we agree that it seems plausible that an algorithm that can take advantage of the information on rewards and constraints in estimating the stationary distribution *may help* the constraint violation go to zero faster (either empirically or theoretically). This may be especially beneficial when we aim to achieve "zero violation” conditions by initializing a policy with safety guarantees. We leave it as future work.

---

> ### Author Response · Authors · 2022-11-13
> **Relation of Meta-SRL to hardness results in (Kwon et al. (2021))**
>
> **W3: The authors mentioned the hardness of bounding regrets in meta-learning by citing Kwon et al. (2021). Although a series of assumptions are made, it is not very clear if they are necessary to address this challenge. In addition, the challenge from constraint seems to be unrevealed yet.**
>
> A: We believe there is a misunderstanding on the relation of our work to Kwon et al. (2021) and would like to clarify as follows.
>
> 1) There is a key difference in the problem setting of meta-learning in our study and the latent MDP setting in (Kwon et.al. (2021)). The latent MDP setting is more challenging in the sense that there is no clear boundary between tasks. In the latent MDP setting, each episode may come from an unknown MDP drawn from a distribution (as a special case of POMDP); in the meta-learning setting, the agent knows when a new task has arrived and is allowed to interact with the MDP over a set of episodes (the number is linear with respect to $M$ in our paper). Due to the above difference, the worst-case lower bound of requiring an exponential number of episodes to learn an $\epsilon$-optimal policy in (Kwon et.al. (2021)) does not hold in our case.
>
> 2) Indeed, if the identity (referred to as ``context” in latent MDP) is revealed or can be inferred, (Kwon et.al. (2021)) is able to achieve a regret that is polynomial in the number of episodes (Thms. 3.3 and 3.4 from Kwon et.al. (2021)).
>
> 3) Close examination of the bounds provided by (Kwon et.al. (2021)) also reveals some differences from our result. In particular, let $K$ be the number of contexts in a latent MDP and $N$ be the total number of episodes ($N$ is on the order of $TM$ in our case as we encounter $T$ tasks, each with $M$ episodes). Then (Kwon et.al. (2021)) is able to bound the regret (without dividing by the number of episodes $N$) as $\mathcal{O}(\sqrt{KN})$. To compare their bound with ours, we consider each task in the meta-learning setting as a context, so $T = \mathcal{O}(K)$. Therefore, their upper bound (after dividing by the number of episodes $N=TM$) becomes $\mathcal{O}(1/\sqrt{M})$, which does not diminish with the number of tasks $T$. Note that our bound (see the comment after Cor. 1) is $\mathcal{O}\left( \frac{\hat{V}_\psi}{M^{3/4}\sqrt{T}} \right)$ (after dividing by the number of episodes $N = TM$), where for simplicity we have assumed $\mathcal{E}_T = 0$, i.e., exact access to the loss function. Note that $\hat{V}_\psi$ is a measure of task-relatedness (a smaller value indicates more relatedness among tasks). It can be seen that while we have a worse order dependence on $M$, our bound scales with task relatedness $\hat{V}_\psi$ and diminishes with respect to the increasing number of tasks $T$. This is expected, as we leverage the relatedness among contexts (in fact, the result  “of Kwon et al. (2021)” would hold when tasks are sufficiently different from each other to infer the contexts with spectral methods).
>
> In summary, we refer to (Kwon et.al. (2021)) as an example that achieving regret diminishing in the number of tasks $T$ is hard, even with the assumption of observing the task identities (contexts). We have added the above discussions to clarify the relation to (Kwon et.al. (2021)) and make comparisons with their results in Appendix I of the updated paper.

---

> > ### Author Response · Authors · 2022-11-13
> > **Challenges from constraints**
> >
> > We remark that constraints may pose challenges when the upper bounds on their violation have different (and even conflicting) forms than the upper bound on the suboptimality of reward maximization. However, in the case of CRPO, they happen to be aligned with each other (see eq.2, restated below):
> >
> > $$R\_0 = J_{t,0}(\pi\_t^*) - \mathbb{E}[J\_{t,0}(\hat{\pi}\_t)]\leq \frac{2}{\alpha_t M}\mathbb{E}\_{s \sim \nu\_t^*}[D\_{KL}(\pi_t^*|\pi\_{t,0})]+\frac{4 \alpha_t c_{max}^2|\mathcal{S}| |\mathcal{A}|}{(1-\gamma)^3},$$
> >
> > $$R\_{i} = \mathbb{E}[J\_{t,i}(\hat{\pi}\_t)]- d\_{t,i} \leq  \frac{2}{\alpha_t M}\mathbb{E}\_{s \sim \nu_t^*}[D\_{KL}(\pi\_t^*|\pi\_{t,0})]+\frac{4 \alpha_t c_{max}^2 |\mathcal{S}| |\mathcal{A}|}{(1-\gamma)^3},   \forall  i = 1,...,p.$$
> >
> > Should the upper bounds on $R_0$ and $R_i$ differ, a potential strategy is to adopt a multi-objective online learning algorithm. We believe that this is a direction for future work.
> >
> > In addition, CRPO is a primal-based algorithm, which, unlike most primal-dual-based safe RL algorithms, avoids the introduction of the dual variables; thus, there is no additional need to search for meta-initialization of the dual variables. However, adapting our framework for primal-dual algorithms is also a potential extension.

---

> ### Author Response · Authors · 2022-11-13
> **Clarification regarding the comment: "static regret seems to be not very useful for meta-learning"**
>
> **W2: In the meta-learning problem with multiple tasks, the static regret seems to be not very useful since the bounded task similarity is assumed and the environments associated with different tasks are different. Otherwise, this becomes a single-task problem that does not have to necessitate meta-learning.**
>
> A: We believe there is a misunderstanding of the notion of static regret in the context of meta-learning and would like to make the following clarifications.
>
> 1)  In our study, the "static comparator’’ used in the static regret refers to the **static policy initialization** $\phi$, **not** the static optimal policy.
>
> 2)  The goal of our theoretical results (Lem. 1, Thm. 3.2, Thm. 3.3, Cor. 1) is to bound the **dynamic regret**, namely TAOG and TACV defined in eq. 3 and restated as follows:
> $$\bar{R}\_0  = \frac{1}{T}\sum_{t=1}^T\bigg[ J\_{t,0}(\pi_t^*) - \mathbb{E}[J\_{t,0}(\hat{\pi}\_t)] \bigg], \hspace{0.3cm} \bar{R}\_{i} =  \frac{1}{T} \sum_{t=1}^T\bigg[ \mathbb{E}[J\_{t,i}(\hat{\pi}\_t)] - d_{t,i} \bigg], \ \forall i = 1,...,p.$$
> Note that the comparators above are a sequence of optimal policies $\\{\pi_t^*\\}_{t=1}^T$.
>
> 3) One of the key **novelties of our work** lies in the *proposed mechanism to bound the dynamic regrets (TAOG and TACV), which are measured against a dynamic sequence of optimal policies* $\{\pi_t^*\}_{t=1}^T$, *via the static regret, which is measured against a fixed initial policy* $\phi$. The crux of our idea is **to relax the problem of bounding the dynamic regret into the corresponding but "easier” problem of bounding the static regret of the upper bounds of the dynamic regret.** This is based on the key observation that we can bound each term of the dynamic regret (left-hand side of the inequality below) by a loss term based on the initial policy (right-hand side, denoted by $f_t(\pi\_{t,0})$):
>
> $$J\_{t,0}(\pi_t^*) - \mathbb{E}[J\_{t,0}(\hat{\pi}\_t)] \leq \underbrace{\frac{2}{\alpha_t M}\mathbb{E}\_{s \sim \nu_t^*}[D\_{KL}(\pi\_t^*|\pi_{t,0})]+\frac{4 \alpha_t c_{max}^2|\mathcal{S}| |\mathcal{A}|}{(1-\gamma)^3}}_{f_t(\pi\_{t,0})}.$$
>
> Thus, summing both sides from $t=1$ to $T$ and dividing by $T$, we have that
>
> $\bar{R}\_0 \leq \frac{1}{T} \sum_{t=1}^T f\_t(\pi\_{t,0}).$
>
> Hence, we have transformed the original problem to a relaxed problem to bound the right-hand side of the inequality above, and all our analyses (Lemma 1, Thms. 3.2, 3.3, Cor. 1) are conducted under this principle. For example, in Lemma 1, under some simplifying assumptions (see our response to your first question https://openreview.net/forum?id=mbxz9Cjehr&noteId=YmyBV6_irnK), we resort to a **``static comparator for initialization”** “bounding the regret with respect to a static comparator for initialization” $\phi$ defined as follows:
>
> $$\frac{1}{T} \sum_{t=1}^T f_t(\pi_{t,0}) - f_t(\phi).$$
>
> Under some common assumptions on $f_t$, we can readily apply existing results from online learning literature to provide such bounds (e.g., $\mathcal{O}(1/\sqrt{T})$ for OGD/FTRL with convex loss functions). Importantly, the bound obtained directly translates to a bound on $\bar{R}_0$, our final target.

---

> > ### Author Response · Authors · 2022-11-13
> > **Practical consideration of defining the static regret with respect to the initial policy**
> >
> > There is a key practical consideration by defining the static regret with respect to the *initial policy*, not the final learned policy. A static regret with respect to a static final learned policy has exactly the flaw pointed out by the reviewer, which reduces the meta-learning problem to a single-task problem. However, a static regret with respect to an initial policy provides freedom for the safe RL algorithm to adapt the initial policy based on observations within the task, which is exactly **the meta-learning scenario analyzed in this paper**.
> >
> > While the static regret with respect to a static initial policy is already sufficient to bounding the “dynamic regret with respect to the dynamic sequence of optimal policies (as is performed above), we further analyze the” **dynamic regret with respect to the dynamic sequence of initial policies** in our key results (Thm. 3.3 and Cor. 1) to address the scenario where the *best meta-initial policies in hindsight may also drift over time*.
> >
> > We hope through the series of clarifications above that we are able to address the reviewer’s concern and convince the reviewer that our analysis with respect to static/dynamic regret of initial policy is a strength, not a weakness (or design flaw). We are happy to address follow-up questions and we have updated the manuscript accordingly to address this important point. (See the highlighted paragraph after Lemma 2 in the updated paper).

---

> ### Author Response · Authors · 2022-11-13
> **Clarification regarding: "Proposed framework builds upon the simplified version of CRPO"**
>
> **W1: The provided CMDP-embedded online learning framework builds on a simplified version of the existing CRPO algorithm. Although this simplification permits convenience in analysis, it brings some restrictions to the applicability of the proposed algorithm.**
>
> A: While we agree that a simplified version of CRPO is used to illustrate the main idea, we would also like to point out that **a full analysis with the original CRPO algorithm is also provided in the main text** to ensure practical applicability. In particular, the results of Thm. 3.1, Thm. 3.3, and Corollary 1 are based on the analysis with the **non-simplified** CRPO algorithm. More details are as follows.
>
> 1) We confirm that we use the simplified version of CRPO to obtain the result in Lemma 1. For quick reference, we use the following (eq. 2 in the paper) as the upper bound for CRPO regret that assumes zero critic-estimation error:
>
> $$J\_{t,0}(\pi\_t^*) - \mathbb{E}[J\_{t,0}(\hat{\pi}\_t)]\leq\frac{2}{\alpha_t M}\mathbb{E}\_{s \sim \nu\_t^*}[D\_{KL}(\pi\_t^*|\pi\_{t,0})]+\frac{4 \alpha_t c_{max}^2|\mathcal{S}| |\mathcal{A}|}{(1-\gamma)^3},$$
>
> where $\hat{\pi}\_t$ is the policy returned by running CRPO for $M$ steps with learning rate $\alpha_t$ in task $t$, $\pi^*\_t$ is the optimal policy, $c_{max}$ is the upper bound on reward/cost function, and $D_{KL}(\cdot|\cdot)$ is the KL divergence. Lemma 1 uses the simplified result above to provide a succinct presentation of the key idea of the CMDP-within-online framework.
>
> 2) As we point out in the paragraph after Lemma 1, this simplified version is not practical for several reasons:
>    * the assumption of access to the optimal policy $\pi_t^*$ when we perform online learning on the upper bound;
>    * the assumption of access to the optimal state distributions $\nu_t^*$ when we perform online learning on the upper bound;
>    * the assumption of access to task-similarity $D^*$ when we set the learning rate $\alpha$.
>
> In addition to the above, we have added a paragraph before Section 2.2, related to the reviewer’s concern that the assumption of the use of a simplified version of CRPO results in the simplified form of the upper bound.
>
> **We address the above limitations with a series of developments in Section 3** to make the proposed framework practical.

---

> > ### Author Response · Authors · 2022-11-13
> > **Further elaborations of the analysis of the original CRPO algorithm**
> >
> > To be more specific, the final bound of our Meta-SRL algorithm presented in Theorem 3.3 (along with Corollary 1) makes use of the *non-simplified CRPO* regrets after incorporating the critic-estimation error as given in eq. 5, restated as follows:
> >
> > $$\bigg[ \frac{c_1^t}{\alpha_{t}M}\mathbb{E}\_{s\sim \nu_t^*}[D\_{KL}(\pi_t^*|\pi\_{t,0})] +  c_2^t\alpha_t    + \sum_{i=0}^p \frac{c_3^t\alpha_{t}+c_4^t\alpha_{t}^2}{\alpha_{t}\sqrt{M} }+\frac{c_5^t}{\sqrt{M}} \bigg] ,$$
> > where the constants are provided in Theorem G.1 in Appendix G.
> >
> > In summary, in our key results (Thm. 3.3 and Cor. 1), we provide a full analysis with the original CRPO (no simplifications). We also remove the unrealistic assumptions from Lemma 1 by designing and analyzing the proposed meta-SRL algorithm without access to the optimal within-task policy $\pi_t^*$, the optimal state distributions $\nu_t^*$, or the task-similarity $D^*$ (when setting the learning rates).
> >
> > We hope this also clarifies the reviewer's second question: **Why we cannot use the original CRPO results? Assuming an exact action-value function makes the proposed algorithm and theory impractical.**

---

> ### Author Response · Authors · 2022-11-13
> **Response to reviewer fmH2**
>
> Dear reviewer fmH2,
>
> Thank you for your feedback and for recognizing our work as a **practical generalization of meta-learning to CMDPs with rigorous guarantees**. We address your comments below.
> If you have any additional clarification questions, we would appreciate a follow-up discussion.

---

> ### Author Response · Authors · 2022-11-15
> **A friendly reminder of the rebuttal's conclusion**
>
> Respected reviewer fmH2,
>
> We'd like to express our gratitude once more for your constructive feedback, and for raising an important issue regarding ethical considerations. We've made several updates to the paper that improve the understanding of our work and responded to each of your questions. Hopefully, you'll find that they adequately address your concerns.
>
> In addition, we would like to know if you have any additional questions or require clarification before the rebuttal phase concludes. We would be happy to address them in the rebuttal's revision.
>
> Best wishes,
>
> Authors of Paper3277

---

> ### Author Response · Authors · 2022-11-17
> **We hope that our responses addressed your concerns**
>
> Dear reviewer fmH2,
>
> We want to thank you again for taking the time to review our paper and giving us constructive feedback to improve the paper. We'd appreciate it if you could let us know whether our responses addressed your concerns. As the end of the rebuttal phase approaches, we look forward to hearing from you and remain available for any additional clarification you may require.
>
> Thank you in advance,
>
> Authors of paper3277

---

> ### Author Response · Authors · 2022-11-21
> **Start of phase 2 discussion: we anticipate your feedback!**
>
> Dear Reviewer fmH2,
>
> As the phase 2 discussion period has started, we sincerely look forward to your feedback. The authors deeply appreciate your time and efforts spent reviewing the paper and helping improve its quality.
>
> We would highly appreciate it if you could once again help review our responses and let us know if we were able to address your concerns.
>
> Please also let us know if there are further questions or suggestions about this paper. We strive to keep improving the quality of the paper, and we are highly grateful for your feedback and time.
>
> With gratitude,
>
> Authors of paper 3277

---

> ### Author Response · Authors · 2022-12-01
> **Thank you for your response and the updated score**
>
> Dear Reviewer fmH2,
>
> Thank you for updating your score after taking into account our revisions.
> As a follow-up to your response---that we have addressed most (*but not all*) of your concerns---we can provide further clarifications for anything that has not been fully addressed.
>
> We appreciate your active engagement during this phase and thank you once again for your efforts in reviewing our paper.
>
> Authors of Paper 3277

---

### Official Review · Reviewer_DCQm · 2022-10-27

**Confidence:** 4
**Correctness:** 4
**Technical Novelty And Significance:** 4
**Empirical Novelty And Significance:** 3
**Recommendation:** 8

**Clarity, Quality, Novelty And Reproducibility:**

I think this is a very good paper, rather clear and of high quality. I vote for acceptance, but would encourage the authors to expand on the experimental evaluation, for instance adding one environment, and making the distributional shift in environments clearer. I list a few questions to the authors below.

Perhaps relevant to add to the literature is the recent paper listed below that deals with robust MDP learning against shifting environments

Suilen et al. Robust anytime learning of Markov Decision Processes. NeurIPS 2022.


Questions

— how is it justified that the ‘meta-initialization’ policy is ‘fair’ that is, has full support over the state/action space? I will necessarily not be safe, if I’m not mistaken. Please discuss this requirement.

— with long sequences of learning tasks, how does the approach perform in sparse reward settings which usually require a lot of data?

— should the safety constraints also be adapted with shifting environments, or what is the motivation to have them static?

— how do the individual tasks compare to a safe policy improvement problem (Thomas et al), where one aims to (safely, or better, reliably) find a policy that outperforms the behavior policy?


**Strength And Weaknesses:**

++Important problem, clearly fitting into ICLR
++Strong technical contribution

—Experimental evaluation is very sparse.


**Summary Of The Paper:**

The proposed method concerns safe reinforcement learning as a constrained MDP problem. In particular, the authors consider safe meta RL, a method to deal with dynamically shifting environments. Technically, the method is realized using a so-called ‘CMDP-within-online’ framework, where a constrained learning task is performed ‘within’, and the resulting policies are input for the meta learner. Theoretical regret bounds and estimation errors (regarding sub-optimality) are given, and the expected theoretical learning rates are derived.


**Summary Of The Review:**

Strong paper, experimental evaluation should slightly be improved.

---

> ### Author Response · Authors · 2022-11-13
> **Relation of individual tasks to the safe policy improvement problem**
>
> **Q5: How do the individual tasks compare to a safe policy improvement problem (Thomas et al), where one aims to (safely, or better, reliably) find a policy that outperforms the behavior policy?**
>
> There are some key differences between the individual tasks (modeled as CMDP) and the safe policy improvement problem:
>
> 1) In the CMDP setup, the utility functions imposed as constraints are often different from the reward function. In the safe policy improvement problem, safety is typically defined by deploying a policy with a return not lower than the behavior policy (with high probability), where the return is often measured with respect to a single reward function.
>
> 2) In the CDMP setup, while the learned policy is required to satisfy constraints (with high probability), each policy update may not monotonically improve the constraint satisfaction. In contrast, in the safe policy improvement problem, it is expected that each policy update can improve (if not deteriorate) the performance of the current policy (with high probability).
>
> However, there are some close connections:
>
> In the problem of meta-safe RL, the goal is to learn an initialization that can quickly adapt to the new task and improve both return and constraint satisfaction. This goal is shared in the problem of safe policy improvement in nonstationary environments (e.g., (Chandak et al. 2020)), where a policy is expected to quickly adapt to the new environment. The technique in (Chandak et al. 2020) is different from ours, as they update the policy based on predictions of its performance in the future. Such a method can also be synergized with our method, where we can perform online meta-learning with a predicted sequence (currently being developed by the authors).
>
> Technically, there is a blurred boundary between the meta-safe RL problem and the safe policy improvement problem in a nonstationary setting: if we treat the behavior policy *as the learned policy from the current task*, and would like to learn a policy that has good performance in the next task, then techniques developed in these two areas can often benefit each other.
>
> **References:**
>
> Thomas, Philip S. (2015) "Safe reinforcement learning.".
>
> Thomas, Philip S., et al.  (2019) "Preventing undesirable behavior of intelligent machines.
>
> "Chandak, Yash, et al (2020). "Towards safe policy improvement for non-stationary MDPs." NeurIPS
>
> **Suggestion: Perhaps relevant to add to the literature is the recent paper Suilen et al.**
>
> A: Thank you for bringing our attention to this very interesting and relevant work. We have added the above reference to the related work section.

---

> > ### Comment · Reviewer_DCQm · 2022-11-17
> > **Thanks for the clarifications!**
> >
> > The comments by the authors clarified all my questions.

---

> ### Author Response · Authors · 2022-11-13
> **Should the safety constraints also be adapted to shifting environments?**
>
> **Q4: Should the safety constraints also be adapted to shifting environments, or what is the motivation to have them static?**
>
> A: Indeed, safety constraints should be adapted to shifting environments--we do not consider them to be static.
>
> **On the algorithm side,** we only work with one loss function for online meta-learning updates because it is an upper bound shared between the objective and constraints. Specifically, in the case of CRPO, the upper bounds for reward maximization and constraint satisfaction happen to be aligned with each other (see eq. 2, restated below):
>
> $$R\_0 = J_{t,0}(\pi\_t^*) - \mathbb{E}[J_{t,0}(\hat{\pi}\_t)]\leq \frac{2}{\alpha_t M}\mathbb{E}\_{s \sim \nu_t^*}[D\_{KL}(\pi\_t^*|\pi\_{t,0})]+\frac{4 \alpha_t c_{max}^2|\mathcal{S}| |\mathcal{A}|}{(1-\gamma)^3},$$
>
> $$R\_{i} = \mathbb{E}[J_{t,i}(\hat{\pi}\_t)]- d_{t,i} \leq  \frac{2}{\alpha_t M}\mathbb{E}\_{s \sim \nu\_t^*}[D\_{KL}(\pi\_t^*|\pi\_{t,0})]+\frac{4 \alpha_t c_{max}^2 |\mathcal{S}| |\mathcal{A}|}{(1-\gamma)^3},   \forall  i = 1,...,p.$$
>
> Thus, by adapting the meta-initialization with respect to the loss function listed on the right-hand side of the above inequalities, we are able to simultaneously adapt both the reward and constraints to the shifting environments. Intuitively, this is because the learned policy has already encoded strategies for these separate objectives into one single policy.
>
> On the other hand, constraints may entail separate treatments when the upper bounds on their violation have different (and even conflicting forms) than the upper bound on the suboptimality of reward maximization. If the upper bounds on $R_0$ and $R_i$ differ, a potential strategy is to adopt a multi-objective online learning algorithm. We believe that this is an interesting direction for future work.
>
> **On the experiment side,** we allow the constraints to vary from task to task. In the frozen lake experiments, the constraints (i.e., the locations of holes) change in every task. In the acrobot experiments, we consider different constraint violation thresholds $d_{t,i}$ for each task and each constraint. Our experiments show that the Meta-SRL is able to learn the meta-initialization which leads to fast convergence in terms of both reward maximization and constraint satisfaction.

---

> ### Author Response · Authors · 2022-11-13
> **Justification regarding meta-initialization policy having full support over the state/action space, and safety considerations?**
>
> **Q2: How is it justified that the ‘meta-initialization’ policy is ‘fair’ i.e., has full support over the state/action space? It will necessarily not be safe if I’m not mistaken. Please discuss this requirement.**
>
> A: Technically, Assumption 1 (full support over the shrinkage simplex set $\Delta \mathcal{A}\_\varrho$) is a minimal requirement even for single-task CRPO to provide provable guarantees. This can be seen in the convergence guarantee of the original CRPO method (Lemma 4 and Lemma 21 in our paper, or Thm. 3 in (Xu et al. 2021)) last line of their proof before the term $D_{KL}(\pi_t^*|\pi_{t,0})$ is hidden in the big-O notation). For example, as shown in our eq. 2,
>
> $$R\_0 = J_{t,0}(\pi\_t^*) - \mathbb{E}[J_{t,0}(\hat{\pi}\_t)]\leq \frac{2}{\alpha_t M}\mathbb{E}\_{s \sim \nu\_t^*}[D\_{KL}(\pi_t^*|\pi\_{t,0})]+\frac{4 \alpha_t c_{max}^2|\mathcal{S}| |\mathcal{A}|}{(1-\gamma)^3}.$$
>
> To ensure that the bound is nontrivial, we need to bound the term $D_{KL}(\pi_t^*|\pi_{t,0}).$ However, if $\pi_{t,0}$ does not have full support over the state/action space, then there may be a state $s$ where $\pi_t^*(s) > 0$ but $\pi_{t,0}(s) = 0$, which would make the KL divergence infinite.
>
> We share the concern of the reviewer that the initial policy may not be safe to start with. This is one of the **key motivations** for us to design a meta-initialization framework that helps *adapt to each task’s constraints and drives constraint violation quickly below the required threshold*. In the experiments, we can see that all the methods start with some constraint violations on the new task, but Meta-SRL with the learned meta-initialization leads to fast improvement in terms of both return and constraint satisfaction.
>
> We believe it is potentially a future direction to learn a meta-initialization that is safe from the first step. It is promising to explore the connection of our technique with existing works on “zero-constraint violations”, such as by incorporating pessimism in learning:
>
> **References:**
>
> Liu, Tao, et al. (2021) "Learning policies with zero or bounded constraint violation for constrained mdps." NeurIPS
>
> Bai et al (2022). "Achieving Zero Constraint Violation for Concave Utility Constrained Reinforcement Learning via Primal-Dual Approach.".

---

> ### Author Response · Authors · 2022-11-13
> **Details on experimental evaluations and extra experiments.**
>
> **Q1: I encourage the authors to expand on the experimental evaluation, for instance adding one environment, and making the distributional shift in environments clearer.**
>
> A: Thank you for the suggestion. Please see the revised paper where we have included additional experiments on the HalfCheetah and Humanoid environment in Appendix H. We will also provide further details to clarify the distributional shift in environments.
>
> Details regarding the distribution shift in the environments are presented in Appendix H. We will make them clearer. Please see below for further information.
>
> **Frozen lake:** We consider a sequence of $T=10$ tasks, where each task differs in the orientations and the probability of a state being frozen or not. We consider two settings when generating the sequence of tasks: high task-similarity and low task-similarity. To generate tasks with high task-similarity, we start with a random frozen lake (modeled as a grid), where the probability of each tile being frozen is $0.7$. For the subsequent environments, we generate a different grid that differs from the first one by only a few tiles. This puts the agent in a setting where the agent always encounters a new grid that is very similar to the previous task. For low task-similarity, random tasks are generated independently, where the probability of a tile being frozen is kept between $0.3$ and $0.7$. The tasks are less similar due to the high uncertainty associated with the changing orientations.
>
> **Acrobot:** Similarly, consider a sequence of $T=50$ tasks, where each task has different mass links and centers of gravity. To generate tasks with high similarity for the acrobot, we considered changing the mass of the links, center of gravity (COG), and constraint threshold for each link. The changes in these quantities were done by adding noise to the default quantities. We considered a Gaussian noise (mean 0) with a low variance of $0.1$ to change the tasks only slightly. To generate low similar tasks, we increased the variance of the Gaussian noise to $0.3$.

---

> ### Author Response · Authors · 2022-11-13
> **Response to the reviewer DCQm**
>
> Dear reviewer DCQm,
>
> Thank you for your feedback and for recognizing our work to address **an important problem with strong technical contributions**. We address your comments below. If you have any additional clarification questions, we would be happy to follow up.

---

### Author Response · Authors · 2022-11-18
**General response and summary of updates in the paper**

Dear Reviewers,

We sincerely thank you all for your insightful comments and constructive feedback, which help us improve the paper. We have revised the paper to address the comments and questions of the reviewers (revision is highlighted in blue). Some of the key updates include:

1) Clarification of key contributions: a) the **first** framework for meta-safe RL with **provable guarantees** under the most **practical settings** (obviate the access to optimal policy or its stationary distribution or the task-similarity/task-relatedness measures), b) rigorous analysis of the regret bounds for task-averaged optimality gap (TAOG) and task-averaged constraint violation (TACV), which crucially establishes the relation with task-similarity or task-relatedness and **theoretical benefits** of meta-learning. (Reviewer GvYx, fmH2).


2) Discussions of our key **technical** contributions on top of and in support of our main theoretical results, which can be of independent interest: a) the first study in the literature on the optimization landscape of CMDP that is **algorithmic-agnostic** (obtained with new techniques based on tame geometry and subgradient flow system for constrained optimization), b) the first result on the static and dynamic regret bounds for **inexact online (multiple) gradient descent**, and c) discussions on the generality of our **analysis template**. (Reviewers GvYx, fmH2).


3) Improved presentation of some key ideas: a) after the definitions of TAOG/TACV, we discussed the technique of reducing the problem of bounding the dynamic regrets (TAOG and TACV) into the corresponding but easier problem of bounding the static regret measured by the upper bounds of TAOG/TACV, and b) after our main result (Thm. 3.3, we clarified that the terms $U_T^{init}$ and $U_T^{sim}$ are just the placeholders for the respective regret bounds for some inexact online algorithms to state our result in the most general way, and we further instantiate it in Cor. 1 to show the theoretical benefits. (Reviewer fmH2, GvYx).

4) Discussion of Assumption 1: why it is needed, how it is used, and its implications/relation to the hardness results presented in (Kwon et al.,(2021)) (see Appendix I). Discussion of Assumption 2 (o-minimal structures): when it can be satisfied and how it is used to obtain the KL-divergence estimation error bound in Thm. 3.1. (Reviewer fmH2).

5) Discussions of the ethical aspect of the proposed method on the incorporation of fairness constraints in Appendix J. (Reviewer fmH2).


Above all, we have conducted **additional experiments** on the HalfCheetah and humanoid environment from Mujoco to enhance the experimental section. In particular, we have adapted the original environments designed for *single-task* learning to the *meta-learning setting* by implementing the most physically meaningful constraint functions and ways to simulate *shifting of tasks*; we believe the set of environments developed in this study (Acrobot, Frozen Lake, Half Cheetah, Humanoid) can serve as potential benchmarks or guidelines in this important but underexplored setting (meta-learning with safety constraints). (Reviewers DCQm, fmH2, GvYx)

Please notify us if you have any further comments/concerns; we would be happy to address them.

With gratitude,

Authors of Paper 3277

---

### Author Response · Authors · 2022-12-07
**Summary of revisions in the paper**

We would again like to thank all the reviewers for their constructive and insightful feedback, which helped us improve the quality of the paper.

Below, we summarize the revisions made to the paper (highlighted in blue):

1. **[Section 1]** Added discussion of our key technical contributions along with our main theoretical results, which may be of independent interest, following the comments of the Reviewer fmH2 and GvYx.
2. **[Section 2.1]** Fixed the typo $\pi_t^*$, added an explanation of the CRPO algorithm, and provided a pointer to Appendices B and D for further algorithmic details as suggested by Reviewer GvYx.
3. **[Section 2.2. Section 3.1]** Added clarification on the rationale behind using a simplified version of the CRPO to obtain preliminary results (see footnote 1), improved presentation of key ideas for bounding dynamic regrets TAOG and TACV in terms of static regret, and provided the intuition behind using fixed initial policy to formulate static regret, as noted by the Reviewer fmH2.
4. **[Section 2.3, Appendix I]** Further reasoning behind the use of Assumption 1 and its implications/relationship to the hardness results presented in (Kwon et al., (2021)), as commented by Reviewer fmH2.
5. **[Section 3.1]** Further discussion on Assumption 2 (o-minimal structure), how we use it to obtain the result in Thm. 3.1 (added a pointer to Appendix F) for further technical details, as pointed out by Reviewer fmH2.
6. **[Section 3.3]** Provided clarifications for the terms used in Thm. 3.3 ($U_T^{init}$ and $U_T^{sim}$), as noted by the Reviewer GvYx.
7. **[Section 4, Appendix H]** Added additional experiments on HalfCheetah and Humanoid environments from Mujoco to enhance empirical evaluations; provided details on the distribution shift used to conduct experiments, as suggested by all reviewers.
8. **[Conclusion, Appendix J]** Added broader impact statement and discussion on the inclusion of fairness constraints for socially responsible systems in Appendix J, as suggested by Reviewer fmH2.

We hope that our responses and revisions to the paper address the reviewer's concerns, and we will be more than happy to discuss further in the event of any unanswered questions. Thank you!

With gratitude,

Authors of Paper 3277

---

### Decision · Program_Chairs · 2023-01-20

**Decision:**

Accept: notable-top-25%

**Justification For Why Not Higher Score:**

* Sparse experiments (although a few environments were added in the rebuttal)
* Theoretical analysis is not always clear

**Justification For Why Not Lower Score:**

* highly innovative work
* safe RL is very important
* The proposed CMDP embedded online learning framework is practical in the sense that it only relies on the inexact solution of sub-CMDP tasks and data trajectories from previous tasks.

**Metareview: Summary, Strengths And Weaknesses:**

The paper introduces the CMDP-within-online framework for meta-safe RL.

Strengths:
* highly innovative work
* safe RL is very important
* The proposed CMDP embedded online learning framework is practical in the sense that it only relies on the inexact solution of sub-CMDP tasks and data trajectories from previous tasks.

Weaknesses:
* Sparse experiments (although a few environments were added in the rebuttal)
* Theoretical analysis is not always clear

**Note From Pc:**

if the above contains the word "oral" or "spotlight" please see: "oral" presentation means -> notable-top-5% and "spotlight" means -> notable-top-25%. As stated in our emails, we are disassociating presentation type from AC recommendations